

**Title**

*A comprehensive characterization of ice nucleation by three different types of cellulose particles immersed in water: lessons learned and future research directions*

**Authors**

Naruki Hiranuma[1,*], Kouji Adachi[2], David Bell[3,**], Franco Belosi[4], Hassan Beydoun[5], Bhaskar Bhaduri[6,***], Heinz Bingemer[7], Carsten Budke[8], Hans-Christian Clemen[9], Franz Conen[10], Kimberly Cory[1], Joachim Curtius[7], Paul DeMott[11], Oliver Eppers[12], Sarah Grawe[13], Susan Hartmann[13], Nadine Hoffmann[14], Kristina Höhler[14], Evelyn Jantsch[8], Alexei Kiselev[14], Thomas Koop[8], Gourihar Kulkarni[3], Amelie Mayer[12], Masataka Murakami[2,****], Benjamin Murray[15], Alessia Nicosia[4,*****], Markus Petters[16], Matteo Piazza[4], Michael Polen[5], Naama Reicher[6], Yinon Rudich[6], Atsushi Saito[2], Gianni Santachiara[4], Thea Schiebel[14], Gregg Schill[11], Johannes Schneider[9], Lior Segev[6], Emiliano Stopelli[10,******], Ryan Sullivan[5], Kaitlyn Suski[3,11], Miklós Szakáll[12], Takuya Tajiri[2], Hans Taylor[16], Yutaka Tobo[17,18], Daniel Weber[7], Heike Wex[13], Thomas Whale[15], Craig Whiteside[1], Katsuya Yamashita[2,*******], Alla Zelenyuk[3], and Ottmar Möhler[14,*]

**Affiliations**

1. Department of Life, Earth and Environmental Sciences, West Texas A&M University, Canyon, TX, USA
2. Meteorological Research Institute, Tsukuba, Japan
3. Pacific Northwest National Laboratory, Richland, WA, USA
4. Institute of Atmospheric Sciences and Climate, National Research Council, Bologna, Italy
5. Center for Atmospheric Particle Studies, Carnegie Mellon University, Pittsburgh, PA, USA
6. Department of Earth and Planetary Sciences, Weizmann Institute, Rehovot, Israel
7. Institute for Atmospheric and Environmental Science, Goethe University of Frankfurt, Frankfurt/M., Germany
8. Faculty of Chemistry, Bielefeld University, Bielefeld, Germany
9. Max-Planck-Institut für Chemie, Mainz, Germany
10. Environmental Geosciences, University of Basel, Basel, Switzerland
11. Department of Atmospheric Science, Colorado State University, Fort Collins, CO, USA
12. Institute for Atmospheric Physics, University of Mainz, Mainz, Germany
13. Leibniz Institute for Tropospheric Research, Leipzig, Germany
14. Institute for Meteorology and Climate Research – Atmospheric Aerosol Research, Karlsruhe Institute of Technology, Karlsruhe, Germany
15. Institute for Climate and Atmospheric Science, School of Earth and Environment, University of Leeds, Leeds, UK
16. Department of Marine, Earth, and Atmospheric Sciences, North Carolina State University Raleigh, NC, USA
17. National Institute of Polar Research, Tachikawa, Tokyo, Japan
18. Department of Polar Science, School of Multidisciplinary Sciences, SOKENDAI (The Graduate University for Advanced Studies), Tachikawa, Tokyo, Japan
*corresponding authors: Naruki Hiranuma (nhiranuma@wtamu.edu) and Ottmar Möhler (ottmar.moehler@kit.edu)
**Now at, Laboratory of Atmospheric Chemistry, Paul Scherrer Institute, Villigen, Switzerland
***Now at, Department of Soil and Water Sciences, Hebrew University of Jerusalem, Israel
****Now at, Institute for Space-Earth Environmental Research, Nagoya University, Nagoya, Japan
*****Now at, Laboratoire de Météorologie Physique (Lamp-CNRS) Aubiere, France
******Now at, Water Resources and Drinking Water Department, Eawag, Dübendorf, Switzerland
*******Now at, Snow and Ice Research Center, Nagaoka, Japan



**Abstract**

We present the laboratory results of immersion freezing efficiencies of cellulose particles at supercooled temperature ($T$) conditions. Three types of chemically homogeneous cellulose samples are used as surrogates that represent supermicron and submicron ice nucleating plant structural polymers. These samples include micro-crystalline cellulose (MCC), fibrous cellulose (FC) and nano-crystalline cellulose (NCC). Our experimental data show that particles resembling the MCC lab particle occur also in the atmosphere. Our immersion freezing dataset includes data from various ice nucleation measurement techniques available at seventeen different institutions, including nine dry dispersion and eleven aqueous suspension techniques. With a total of twenty methods, we performed systematic accuracy and precision analysis of measurements from all twenty measurement techniques by evaluating $T$-binned (1 °C) data over a wide $T$ range (-36 °C < $T$ < -4 °C). Specifically, we inter-compared the geometric surface area-based ice nucleation active surface-site (INAS) density data derived from our measurements as a function of $T$, $n_{s,geo}(T)$. Additionally, we also compared the $n_{s,geo}(T)$ values and the freezing spectral slope parameter ($\Delta\log(n_{s,geo})/\Delta T$) from our measurements to previous literature results. Results show that freezing efficiencies of NCC samples agree reasonably well, whereas the diversity for the other two samples spans for ~10 °C. Despite given uncertainties within each instrument technique, the overall trend of the $n_{s,geo}(T)$ spectrum traced by the $T$-binned average of measurements suggest that predominantly supermicron-sized (giant hereafter) cellulose particles (MCC and FC) generally act as more efficient ice-nucleating particles than NCC with about one order of magnitude higher $n_{s,geo}(T)$. Further, our results indicate significant diversity between dry and aqueous suspension measurement techniques. The ratios of the individual measurements ($n_{s,ind}$) to the log average of $n_{s,geo}(T)$ range 0.6-1.4 across the examined $T$ range. In general, the ratios of the log average of dry dispersion measurements are higher than those of aqueous suspension measurements. The observed discrepancy may be due to non-uniform active site density for different sizes and/or the alteration in physico-chemical properties of cellulose by liquid-suspending it. Unless otherwise defined, the cellulose system may not be an ideal calibrant. Given such a distinct difference between two subgroups of immersion freezing techniques, standardization of our methods, especially INP sampling and treatment, may be one approach to reduce the measurement diversity and valiability when we deal with a complex material like cellulose. A community-wide effort to identify specimen-specific limitations and characteristics of each technique, as well as consolidating the $n_{s,geo}(T)$ parameterization, is an alternative approach to achieve overall precise and accurate ice-nucleating particle measurements.



## 1. Introduction

### 1.1 Background

Glaciation of supercooled clouds through immersion freezing induced by ice-nucleating particles (INPs) is an important atmospheric process affecting the formation of precipitation

and the Earth's energy budget (*Boucher et al.*, 2013; *Vergara-Temprado et al.*, 2018). Currently, the climatic impact of INPs is, however, uncertain due to our insufficient knowledge regarding their diversity and abundance in the atmosphere (e.g., *Hoose and Möhler*, 2012; *Murray et al.*, 2012; *Kanji et al.*, 2017; *Knopf et al.*, 2018). Recently, micro-crystalline cellulose (MCC) particles of <16 µm in diameter, extracted from natural wood pulps (Aldrich, 435236),

have been identified as an efficient INP (*Hiranuma et al*. 2015a, H15a hereafter). Experiments with this surrogate may provide useful information to understand the role of biological INPs in the troposphere as presented in H15a. Conspicuously, the H15a modeling results suggest that the tropospheric concentration of ice-nucleating cellulose becomes substantial (>0.1 $L^{-1}$) below about -21 °C.

15        In general, airborne cellulose particles are prevalent (>0.05 µg $m^{-3}$) throughout the year even at remote and elevated locations as reported in *Sánchez-Ochoa et al.* (2007). Their water insoluble, hydrolysis resistant and heat resistive features (*Fernández et al.*, 1997; *Quiroz-Castañeda & Folch-Mallol*, 2013) may in part explain the long-range transport and high concentrations of cellulose even at geographically dispersed sites. Another unique

characteristic of ambient cellulose is its wide range of physical size available for freezing. For example, the size distribution measurements of ambient cellulose particles by *Puxbaum and Tenze-Kunit* (2003) indicate the presence of particulate cellulose in the range from 10 nm to >20 µm. The presence of supermicron particles, possessing larger surfaces as compared to submicron ones, is remarkable since they can potentially act as giant INPs since large surfaces

may promote efficient formation of ice embryos (*Pruppacher and Klett*, 2010; *Schnell and Vali*, 1972 and 1973). Nevertheless, more comprehensive characterization of ice-nucleating properties of various cellulose-containing particles is indeed necessary to examine if the ice-nucleating activity is specific to MCC or generally relevant to all cellulose materials in the atmosphere.

### 1.2 Previous INUIT Inter-comparison Activities

In 2012, the German research consortium-led INUIT (Ice Nuclei research UnIT) project was commenced to comprehensively study the heterogeneous ice nucleation processes in the atmosphere. Throughout the period since, this project has provided a trans-national platform



to bolster collaborative research activities between various yet meticulous groups who study atmospheric INPs. In turn, INUIT has accelerated ice nucleation research in a wide range of study scales from nanoscopic microphysics (e.g., *Kiselev et al*., 2017) to cloud scale modeling (e.g., *Diehl and Mitra*, 2015 ; *Paukert and Hoose*, 2014) in cross- and inter-disciplinary

manners.

Formerly, several INUIT studies addressed quantitative validations of ice nucleation (IN) instruments using test proxies of atmospheric particles (*Wex et al*., 2015; *Hiranuma et al*., 2015b; *Burkert-Kohn et al*., 2017). Some studies focused on identifying potential reasons of the data diversity (e.g., different experimental methods and sample preparation methods).

For example, *Burkert-Kohn et al*. (2017) remarked the importance of the inter-comparison workshop by co-deploying instruments with a uniform aerosol dispersion procedure and size segregation method to minimize the diversity in ice nucleation results. *Hiranuma et al*. (2015b), H15b henceforth, took a different approach to perform an inter-comparison of INP measurement techniques. The authors demonstrated the collaborative multi-institutional

laboratory work with a total of fourteen institutions (seven from Germany, four from U.S., one from U.K., one from Switzerland and one from Japan) by distributing a test particulate sample to partners and allowing measurements at their home laboratories. The authors discussed the potential effect of sampling of the dust, agglomeration, flocculation, surface estimation methods, multiple nucleation modes and chemical aging on the observed data deviation

amongst seventeen different IN instruments. This study suggested that a combination of above-listed factors may be responsible for ~8 °C diversity in terms of temperature and up to three orders of magnitude difference with respect to the ice nucleation active surface-site (INAS) density, $n_s(T)$, parameters. Further, two follow-up studies on potential effects of aggregation upon IN were performed in *Emersic et al*. (2015) and *Beydoun et al*. (2016). The

former study presented the potential role of aggregation and sedimentation of mineral particles, altering their IN efficiency in aqueous suspension, by combining experimental and modeling approaches. The latter study presented a subset of cellulose data used in the concurrent study, and the authors postulated that the widening of the frozen fractions and enhanced ice activity towards high $T$ was attributable to increased diversity in ice nucleating

activity for lower concentrations and particle surfaces. In other words, there is a distribution of active sites between individual droplets depending on the total surface area. Nevertheless, our understanding of overall consistency of current INP measurement techniques and dominant mechanisms that may be responsible for diversity among measurements is still insufficient.



### 1.3 Goals

The measurement strategy for this study was formulated in year 2015 to further augment our understanding of sensitivity of various ice nucleation instrument techniques towards immersion freezing efficiency. Beyond official INUIT-participating institutes, including

Bielefeld University (BU), Goethe University Frankfurt (GUF), Johannes Gutenberg University of Mainz (JGU), Karlsruhe Institute of Technology (KIT), Max Planck Institute for Chemistry (MPIC), Leibniz Institute for Tropospheric Research (TROPOS), Technical University of Darmstadt (TUD) and Weizmann Institute of Science (WIS, alphabetical order according to the abbreviations), ten associated institutes (five from U.S., three from E.U. and two from Japan)

are involved in this study. These associated partners include Carnegie Melon University (CMU), Colorado State University (CSU), North Carolina State University (NC State), Pacific Northwest National Laboratory (PNNL), West Texas A&M University (WTAMU), Institute of Atmospheric Sciences and Climate-National Research Council (ISAC-CNR), University of Basel, University of Leeds, Meteorological Research Institute (MRI) and National Institute of Polar Research (NIPR).

We have shared three cellulose samples: micro-crystalline cellulose (MCC, Aldrich, 435236), fibrous cellulose (FC, Sigma, C6288) and nano-crystaline cellulose (NCC, Melodea, WS1) as atmospheric surrogates for non-proteinaceous biological particles to perform immersion freezing experiments with the collaborators involved in this study to obtain immersion freezing data as a function of multi-experimental parameters (see **Sect. 3.1**). The motivation

of using multiple types of cellulose was to examine the immersion freezing abilities of both predominantly supermicron (MCC and FC) and submicron (NCC) cellulose particles towards assessing a wide size range of chemically uniform biological particles.

      A total of twenty measurement techniques are used in this inter-comparison study to compile a comprehensive dataset for evaluating immersion freezing properties of cellulose

samples. The dataset is analyzed to understand functional dependence of various experimental parameters and of cellulose particle characteristics. In this work, eleven instruments test samples using aqueous suspensions, while nine examine aerosolized powders dispersed in synthetic air with a low RH or atomized/nebulized-suspensions containing cellulose samples followed by diffusion drying process, referred to as dry dispersion methods

henceforth. The basic experimental methods and parameterization approaches used to interpret the data are discussed in **Sects. 3.1 and 3.2**.

      This work extends a previous proof-of-principle experiment that demonstrated the importance of cellulose-containing particles in the atmosphere (H15a). The main objective of this study is to examine how different ice nucleation instrument techniques compare when

using chemically homogeneous biological material rather than multi mineral systems, such as



illite NX (e.g., *Broadley et al.*, 2012) and understand if cellulose can be used as a standard reference material in INP research. Besides, the comprehensive ice nucleation data of cellulose materials presented in this work can be used to elucidate the role of biological ice-nucleating aerosols (e.g., *Després et al.*, 2012).


## 2. Sample Preparation and Characterization

### 2.1 Sample Specifications

All of our samples are linear polymers of glucosyl derivatives, mechanically extracted through <200 °C heat application and catalytic oxidation (e.g., *Battista et al.*, 1962; *Brinchi et al.*, 2013).

In particular, MCC is extracted from hardwoods (e.g., oak, personal communication with the manufacturer, Aldrich). A summary of major properties of three samples is provided in **Table 1**. Briefly, these highly stable biopolymers, whose bulk density ranges between 1.0-1.5 g cm$^{-3}$, exhibit different physical dimensions depending on sample processing and treatments. As seen in **Table 1**, the geometric size of dispersed particles are more than ten-fold smaller than

the electron micrograph-assessed size of bulk materials without any exception, suggesting the presence of super aggregates in non-dispersed bulk samples. We note that the powder size of MCC reported by the manufacturer (~50 μm) is in good agreement with our Scanning Electron Microscope (SEM)-measured size. In contrast, the particle size of NCC reported on the manufacturer's material data sheet (TEM-based data) is more comparable to the dispersed

particle diameter of ~0.2 μm than the SEM-based size. In this manuscript, the NCC size by SEM represents the size of NCC residuals (i.e., leftover particles after evaporating water content) from 5 μL suspension droplet of 0.03 wt%. Due to the high viscosity of the gelatinous form of NCC (4,665 ± 200 cP at 25 °C), aggregation may have occurred while evaporating water. Even after the 15 minute ultrasonic bath treatment of the suspension, aggregates seem to remain

unelucidated, which is reflected in its SEM-based diameter of >2.5 μm. A more detailed discussion of particle and residual size distributions are available in the **Supplemental Information**.

The average aspect ratios (ARs) of each cellulose material in **Table 1** were estimated with an identical procedure employed in our previous H15a study. We evaluated a total of

4,976 MCC, 371 FC and 764 NCC particles. The Everhart-Thornley Detector (ETD) of a scanning electron microscope (SEM, FEI, Quanta 650 FEG) was used to acquire the below-the-lens micrograph image and measure two dimensional axis length of particles deposited on membrane filters. The degree of elongation appears to be higher for NCC (average AR up to 2.93) when compared to MCC and FC (average AR of <2.30). Nonetheless, all sample types

show that particles are elongated with an aspect ratio varying from ~2 to 3, which is similar to our previous measurement on MCC particles (i.e., 2.1).

Three different measurements of the unit surface area per unit mass (specific surface area, *SSA*), namely geometric *SSA*, SEM-based *SSA* and BET-*SSA*, for each system are also shown in **Table 1**. These measurements correspond to *SSA* of mechanically aerosolized



particles (<10 μm in diameter) in the Aerosol Interaction and Dynamics in the Atmosphere (AIDA) chamber, droplet residuals obtained after evaporating water content of 5 μL droplet of 0.03 wt% aqueous suspension and bulk samples, respectively. Our intention of using different *SSA* metrics is to provide the most adequate parameter for the $n_{s,geo}(T)$ estimation of individual

techniques based on their characteristics (e.g., geometric *SSA* for dry dispersion techniques and SEM-based *SSA* for aqueous suspension techniques). As demonstrated in our previous H15b comparison effort, when a reduced *SSA* value is observed for a same sample, it indicates the presence of agglomeration. Hence, the degree of aggregation of cellulose fibers is presumably responsible for the observed differences in SEM-based *SSA* values for residuals

obtained from suspensions from geometric *SSA* of the mechanically aerosolized particles (**Table 1**). Alternatively, a loss of larger particles from the sample which may happen in airborne aerosols due to settling or impaction in the particle generation set-up may also lead to different SSA values if the surface properties of the cellulose particles differ with the particle size.

### *2.2 Chemical Composition*

Single particle mass spectra of dry dispersed FC and MCC particles in the size range between 200 and 3500 nm were measured in the laboratory using the Aircraft-based Laser ABlation Aerosol Mass spectrometer (ALABAMA, *Brands et al*., 2011). The averaged mass spectra of both cellulose types are shown in **Fig. 1**. The mass spectra of the dry dispersed

particles show high signals of anions at mass-to-charge ration, m/z, of -45 ($HCO_2$), -59 ($CH_3COO$) and -71 ($C_3H_3O_2$). These are typical markers for biomass burning particles, especially levoglucosan $C_6H_{10}O_5$, 1,6-anhydro-*β*-D-glucopyranose) (*Silva et al*., 1999). Levoglucosan is an anhydrous sugar formed from the pyrolysis of carbohydrates, such as naturally occurring starch and cellulose (*Madorsky et al*., 1959; *Lakshmanan et al*. 1969). Thus, it is not surprising

that the mass spectrum of cellulose particles resembles that of levoglucosan. The above mentioned marker ions should therefore be regarded as general markers for plant-related material and are not unique to levoglucosan or cellulose. Now for the cations, the prominent ions are found on the peaks at m/z 19 ($H_3O^+$), 27 ($Al^+$ or $C_2H_3^+$), 39 ($K^+$), 43 ($AlO^+$, $C_2H_3O^+$, or $C_3H_7^+$) and 56 ($Fe^+$). The presence of some ions, such as Al, K and Fe, may indicate

contamination of the sample.

A more detailed analysis of the individual mass spectra revealed several distinct particle types. Using a combination of fuzzy clustering (*Hinz et al*., 1999) and the marker peak search method based on the above mentioned and further characteristic ions, we found that ~75% of FC particles contained the characteristic marker peaks. The average mass spectrum



of these FC particles is shown in **Fig. 1a**. The remaining 25% of the particle mass spectra showed similar cation spectra but the anions were dominated by signals of elemental carbon ($C_n^-$). This may be due to a stronger fragmentation of the cellulose molecules or due to other effects. Previous studies have identified at least 37 different compounds in products of

cellulose pyrolysis (*Schwenker and Beck*, 1963). Further, those ions in the remaining 25% of the spectra may indicate aluminosilicates that could be a contamination of the sample. The source of these impurities is not known. Two potential sources include the manufacturing process (e.g., controlled acid hydrolysis during the mechanical extraction of natural fibers) and/or contamination from ambient lab air. Similar results were obtained for dry dispersed

MCC cellulose particle (See **Fig 1b**). Briefly, approximately 60% of the mass spectra were clearly identified by means of the above mentioned marker peaks. The remaining mass spectra show again the $C_n$ pattern, possibly indicating higher fragmentation, as well as the aluminosilicate contamination.

To compare properties of MCC particles generated by nebulization and dry dispersion,

a single particle mass spectrometer (miniSPLAT), a Centrifugal Particle Mass Analyser (CPMA), and a Scanning Mobility Particle Sizer (SMPS) (*Zelenyuk et al.*, 2015; *Alexander et al.*, 2016) were used to measure the aerosol particles vacuum aerodynamic and mobility diameters ($d_{va}$ and $d_m$ respectively) of mass-selected MCC particles, their mass spectra and effective densities. The "nebulized" cellulose particles were generated by nebulizing a 0.06 wt%

suspension using PELCO all-glass nebulizer (14606, Ted Pella, Inc.) and dried through a diffusion dryer prior to characterization. The "powder" particles were generated by powder dispersion using the TOPAS Solid Aerosol Generator (SAG 410) with the spoon method, where small volumes of dry cellulose sample are dispersed by placing it on a spoon and holding it under the ejector.

The results of these measurements are shown in **Fig. 2**. As shown in **Fig. 2a**, for a given mass and, thus, for a given volume equivalent diameter ($d_{ve}$), the nebulizer-generated MCC particles have smaller mobility diameters when compared to the dry powder population. In contrast, the nebulized MCC particles have larger $d_{va}$ than the dry powder ones (**Fig. 2b**). Such behavior indicates that MCC particle generated by dry dispersion are more aspherical and have

larger dynamic shape factors than nebulizer-generated particles (*Alexander et al.*, 2016; *Beranek et al.*, 2012). Consistently, we find that the full width at half maximum (FWHM) of the $d_{va}$ distributions for mass-selected MCC particles generated by dry powder dispersion are broader than those observed for nebulizer-generated particles with the same mass, signifying the presence of more aspherical particles and particles with distribution of shapes as discussed

in detail in separate publications (*Alexander et al.*, 2016; *Beranek et al.*, 2012). As an example,



data shown in **Fig. 2b** and the material density of 1.5 g cm$^{-3}$ yield average free-molecular regime dynamic shape factors of 2.20 and 1.96 for dry powder dispersion and nebulizer-generated MCC particles, respectively. The $d_{va}$ measurements of size-selected particles can also be used to calculate the average effective densities of the nebulizer- and dry powder-generated particles, shown in **Fig. 2c**. The figure shows that at least across the examined size range ($d_{va}$ and $d_m$ <450 nm) the calculated effective densities appear to be independent on the particle size (**Fig. 2c**), implying homogeneous physical properties. The average effective density of the nebulizer-generated MCC particles (1.16 ± 0.05 g cm$^{-3}$) is higher than the average effective density of dry powder-generated particles (0.96 ± 0.03 g cm$^{-3}$), pointing to the relative abundance of compacted, less aspherical and/or less porous particles in the nebulized population. However, both effective densities are lower than the bulk material density (1.5 g cm$^{-3}$), indicating that both types of particles are aspherical or/and have voids. Clearly, the micrographs of cellulose particles indicate their aspherical elongated appearance with substantial amount of surface structures (Figs. S1 and S3 of H15a).

Finally, **Fig. 2d** presents the comparison of the average mass spectra of nebulizer- and dry-generated MCC particles, acquired by miniSPLAT. The mass spectra of the MCC particles generated by dry dispersion were dominated by C$^+$, CO$^+$, CO$_2$$^+$, C$_2$O$_2$H$^+$, C$_2$O$_3$H$^+$, O$^-$, C$_2$H$^-$. The mass spectra of the MCC particles generated by nebulization of aqueous cellulose suspension exhibited additional peaks (i.e., Na$^+$, K$^+$), most likely from the trace-level metal impurities in the water. Note that the high relative intensity of these peaks in *all* mass spectra of individual nebulizer-generated MCC particles are due to high ionization efficiencies of the alkali metals in single-particle mass spectrometers like miniSPLAT and ALABAMA. While the presence of these trace metals in nebulizer-generated MCC particles, presumably will have negligible effects on IN measurements, the significant differences in shape and morphology of nebulizer- and dry powder-generated MCC particles may affect their IN activity.

### 2.3 Tests to Investigate Impurities

We characterized the samples in addition to what the manufacturers reported. One of the weaknesses of the indirect technique validation at multiple venues is the difficulty to ensure sample purity and stability during distribution and measurement at each institute. Impurity inclusions are often uncontrollable partly because each team treats the samples differently for necessity and known reasons (**Sect. 3.1**). Potential sources of contaminants include organic gases covering the substrate's surface or the interaction of volatile organic compounds (VOCs) at the vapor-liquid interface (*Whale et al.*, 2015). Besides, several previous studies have reported the dissolution behavior of contaminants (e.g., siloxane and sodium containing



materials) from the standard apparatus, such as conductive tube and glassware in water, and even ultra-pure water itself (e.g., *Yu et al.*, 2009; *Timco et al.*, 2009; *Bilde and Svenningsson*, 2004).

Though it is hard to identify the source of any potential contaminations and isolate the possibility of sample impurity from other sources and artifacts, such as apparatus and procedures used for solution preparation or sample dispersion, the INUIT group has made an effort to ensure the quality and purity of the samples. The laboratory test results from two electron microscopy groups (KIT and MRI) are discussed in the following sections.

In the Laboratory for Electron Microscopy at the Karlsruhe Institute of Technology, we tested the purity of MCC and FC powders (>0.4 μm), transported back and forth between U.S. and Europe, using a SEM (FEI, Quanta 650 FEG). In this test, we placed bulk cellulose powders on 47 mm membrane filters (Whatman® Nuclepore™ Track-Etched Membranes, 0.2 μm pore size) followed by the sputter coating process to cover cellulose particles with a conductive carbon layer. Subsequently, the coated-membranes were placed in a SEM chamber and exposed to an electron beam to assess the brightness of individual particles with a backscattered electron detector (contrast/brightness = 88.8/74.2) and their elemental compositions with an energy dispersive X-ray (EDX) detector. At the end, this assessment allows for isolation of non-carbonaceous materials (e.g., dusts and metals) from the other materials according to the brightness contrast (if there are any). With this methodology, we analyzed a total of 5637 particles (3898 MCC and 1739 FC particles) and found impurity inclusions of only <0.25%. This number is nearly equal to the impurity fraction in MCC of 0.28%, which is reported in *Ohwoavworhua and Adelakun* (2010). A few contaminants identified in our cellulose samples are copper/aluminum oxide, quartz, chromium sulfate/sulfide, sodium chloride, non-aluminosilicate salt, pure chromium and lead. Note that no aluminosilicates were found. Except lead (*Cziczo et al.*, 2009), all other compounds are known for negligible ice nucleation activities at $T$ > -25 °C and for at least an order magnitude lower $n_s(T)$ as compared to H15a-MCC as suggested in our previous AIDA tests and other studies (e.g., *Archuleta et al.* 2005; *Steinke*, 2013; *Hiranuma et al.*, 2014; *Atkinson et al.*, 2013).

A complementary impurity analysis was carried out using another SEM-EDX (SU-3500, Hitachi) and a transmission electron microscope (TEM, JEM-1400, JEOL) at MRI, Japan. A total of 123 SEM images of MCC and FC powders (<10 μm) as well as a few TEM images of NCC that has the geometry of several tens nanometer with 500-800 nm length were analyzed. There were no notable contaminants except some expected elements, such as sulfur and sodium, possibly stemmed from the manufacturing process of NCC [i.e., $(C_6H_9O_5)_n (SO_3Na)_x$].





In some cases, bulk particles may break up and apart into fragments, and those fragments may appear in an analytical instrument (e.g., single particle mass spectrometer) with a high detection sensitivity and efficiency. For MCC, the total fraction of contaminants, which may cumulatively derive from any experimental procedures (e.g., sample transport,

treatment and impurity), is 3%, as formerly reported in H15a. Ostensibly, these contaminants may have emanated from the brush generator or the AIDA chamber wall. Nonetheless, our blank reference expansion AIDA experiments (i.e., background expansion cooling measurements without aerosol) suggest that impurities are quantitatively negligible to impact overall ice nucleation activity of cellulose itself at heterogeneous freezing temperatures of $T >$

-33 °C. In brief, we examined the immersion mode IN activity of 'sample blanks' injected through running a blank brush generator for >60 min in the chamber. Our SMPS/APS measurements showed that the blank injection provided >10 $cm^{-3}$ of particle concentration (equivalent to >1 $\mu m^2$ $cm^{-3}$ surface), and >80% of background particles are smaller than 250 nm. Our experimental results (2 independent expansions; INUIT03_2 and _3) indicated no ice

observed at $T >$ -33 °C. Further discussion regarding impurity is beyond the scope of the concurrent study.

### 2.4 Atmospheric Relevance

To examine if ambient particles resemble our test cellulose particles, we compared the laboratory spectra of dry dispersed FC and MCC to the ambient particle spectra measured by

a single particle mass spectrometer, ALABAMA. For the ambient measurement, ALABAMA was utilized on board of the Gulfstream G-550 High Altitude and Long-Range Research Aircraft (HALO) during the Midlatitude Cirrus (ML-CIRRUS) aircraft campaign to study aerosol-cloud-climate interactions focused on natural cirrus clouds in 2014 over Central Europe (*Voigt et al.*, 2017). We chose to assess the ALABAMA data from the ML-CIRRUS campaign because this

aircraft measurement was conducted at mid-latitudes, where abundant cellulose aerosols might be expected.

We searched the data set of 24,388 atmospheric particle mass spectra for the occurrence of the characteristic marker peaks found from the reference mass spectra (i.e., **Fig. 1**). For this search we focused on cations because the data quality of the anions during ML-

CIRRUS was not sufficient. Depending on the exact search criteria and signal intensity thresholds, we found that between 0.5 and 1.0% of the particles (between about 120 and 240 particles) matched the search criteria. For the comparison between the ambient mass spectra and the reference mass spectra, we restricted the size range of the reference mass spectra to vacuum aerodynamic diameters below 900 nm because the inlet system of ALABAMA



transmitted only particles up to 900 nm during the aircraft measurements. The mean mass spectra of the ambient particles were compared with the laboratory spectra (< 900 nm only) by means of the correlation coefficient ($r^2$). The correlation coefficient ranged between 0.5 and 0.6 ($r^2$), indicating that the atmospheric particles are not identical to the laboratory

5    spectra of cellulose, but show a certain resemblance in the abundance of ions. The best match (averaged mass spectrum of 238 atmospheric particles and averaged mass spectrum of 22 MCC spectra of particles < 900 nm) is shown in **Fig. 3**. The correlation coefficient $r^2$ of the two spectra is 0.58. The atmospheric particles were found in all altitudes in the troposphere and even in the lowermost stratosphere during ML-CIRRUS ranged between 10 and 14 km.



## 3. Methods

### 3.1. Ice Nucleation Measurements

Twenty techniques were used to investigate the ice-nucleating properties, in particular immersion freezing (*Vali et al.*, 2015), of cellulose particles (**Table 2**). In this study, nine

techniques employed dry dispersion methods that refer to experiments employing water vapor condensation onto dry dispersed particles followed by droplet freezing, and another set of eleven techniques used aqueous suspension methods that denote the experiments started with the test sample pre-suspended in water before cooling. Detailed information of individual methods and their applications to study atmospherically relevant INPs are provided in

references given in **Table 2** and elsewhere (e.g., *DeMott et al.*, 2017). The summary tables containing quantitative and nominal descriptions of both dry dispersion and aqueous suspension methods are available in **Tables 3-6**.

A summary of quantifiable parameters involved in dry dispersion experiments is given in **Table 3**. For dry dispersion measurements, both monodisperse and polydisperse aerosol

populations were used to examine ice nucleation abilities. Monodisperse particles were size-selected by a differential mobility analyzer (DMA, manufacturer information are given in **Table 1**), and selected sizes ranged from 320 to 800 nm in mobility diameter depending on the aerosol and ice detection sensitivity of the technique. For MCC and FC, polydisperse particles were predominantly in the supermicron size range, but the particle size distributions varied

between techniques as the mode diameters ranged from ~1 to 2 μm. The measured geometric *SSA* values correspondingly deviated for up to an order of magnitude for all cellulose sample types, indicating various size distributions. Similarly, the size of supercooled droplets ranged from 2.6 to 90 μm, and the ratio of the aerosol size (i.e., mode diameter) to the droplet size also ranged over two orders of magnitude (0.0036-0.5). Furthermore, a total number of

droplets examined per experiment varied over two orders of magnitude (100-10,000) depending on the technique. Above all, the temperature uncertainty of the dry dispersion techniques was fairly small (within ± 1 °C) despite of variation in cooling rate (0.9-2.8 °C min$^{-1}$), ice nucleation time (0.2 s – 15 min) and a difference in the way of determining the fraction of frozen droplets. Concerning the latter, most of the dry-dispersion methods measure the

concentration of ice crystals and separately determine the particle concentration, assuming that for immersion freezing measurements the conditions chosen in the instrument cause all particles to be activated to droplets. This yields a value called "activated fraction"(*AF*) in e.g., *Burkert-Kohn et al.* (2017). Others look at the entirety of all droplets and check how many of these are frozen, determining a "frozen fraction" (*FF*), the latter being done e.g., for LACIS



(*Burkert-Kohn et al.*, 2017), but generally also for all aqueous suspension methods. Likewise, the uncertainties in $RH_w$ and $S_w$ are also small (<5%). However, it should be pointed out that recently systematic differences were described when comparing CFDC (continuous flow diffusion chamber) methods with other immersion freezing methods (AIDA and LACIS), (*DeMott et al.*, 2015; *Burkert-Kohn et al.*, 2017). In these studies, simultaneous measurements at the same measurement location were done, and CFDCs yielded lower results by roughly a factor of 3 for conditions where all particles should activate to droplets in the instruments.

**Table 4** provides a summary of quantifiable experimental parameters of the aqueous suspension techniques. A majority of the techniques used the bulk cellulose samples, containing larger particle sizes as compared to dry dispersed ones. In association with their large grain size, bulk samples exhibited smaller $SSA$ than dry dispersed ones (**Table 1**). Note that the SEM-based $SSA$ values from **Table 1** were used for the $n_{s,geo}(T)$ estimation of most bulk-based measurements. Two exceptions were the <10 μm particles examined with NIPR-CRAFT and dispersed particles collected on filters and scrubbed with deionized water for FRIDGE-CS. The results of these unique size-segregated measurements were compared to the bulk results (see **Sect. 4.3**).

The volume of water used each aliquot in aqueous suspension techniques was in many cases much larger than in the volume of the droplets generated in dry dispersed techniques. The ratio of the aerosol mass (i.e., mass equivalent diameter) to the droplet mass of this subset was on average much smaller (for less than an order of magnitude) as compared to that of the dry dispersion subgroup. Therefore, the solute concentration per drop in the wet suspension experiments was greater than in the dry suspension experiments. This might be important since solutes have been shown to both enhance and suppress ice nucleation even in very dilute solutions (Kumar et al., 2018; Whale et al. 2018). An exception was WISDOM, which used <100 μm droplet diameters (<0.5 nL volume). A total number of droplets examined per experiment was several hundred at the most and typically smaller than that of dry dispersion techniques. The total surface area probed was, however, much larger in aqueous suspension methods, resolving much warmer temperatures. Temperature was well-controlled in these methods. For example, similar to the dry dispersion measurements, the temperature uncertainty was fairly small (within ± 1 °C) regardless of variations in cooling rate (0.4-2.0 °C min$^{-1}$). As seen in **Table 4**, the weight percent of particle suspensions varied over five orders of magnitude ($10^{-5}$ to 1 wt%) to access a wider freezing temperature range. On the other hand, the resulting $n_{s,geo}(T)$ uncertainty of >20% and slope parameter of $n_{s,geo}(T)$ spectrum (0.2 < $\Delta\log(n_{s,geo})/\Delta T$ < 0.47) exhibited large deviations as can be seen in **Table 4**. $\Delta\log(n_{s,geo})/\Delta T$ of this subgroup





(~0.34) was on average larger than the dry dispersion subgroup (~0.18). More detailed discussion of quantifiable parameters in **Tables 3 and 4** are provided in **Sect. 4.5.2**.

Nominal method descriptions of dry dispersion and wet suspension techniques are listed in **Tables 5** and **6**. Information given in these tables include the impactor type used while

dispersing cellulose materials (if employed), background correction method, ice detection method, valid data range, sample pre-treatment, water type and status of the suspension solution while generating droplets/vials.

Background correction methods vary amongst the dry dispersion methods (**Table 5**). For CFDCs (CSU-CFDC, INKA and PNNL-CIC), background INP concentrations estimated by

taking measurements through a filter for before and after the sample period were accounted. For cloud simulation chambers (AIDA and MRI-DCECC), an expansion without aerosols in the vessel, namely blank expansion (*Hiranuma et al*., 2014), was conducted to confirm negligible background non-IN active particle concentrations prior to the experiment. For diffusion cells (DFPC-ISAC and FRIDGE-default), background INP concentrations on blank filters/wafer were

subtracted from the actual ice crystal concentrations of loaded filter/wafer.

Note that only non-mandatory guidelines were provided as an experimental protocol by INUIT to those who employed aqueous suspension techniques, and the experimental protocol for the wet suspension techniques was decided by each investigator. The intention was not to introduce limitations and constrains to participants. For MCC and FC, the INUIT

protocol recommended the following procedures:

1. Measurements with <0.05 wt% suspension,
2. Idle time of ~30 min without stirring for large particles to settle out,
3. Prepare droplets out of the quasi-steady state suspension (i.e., the upper layer of the suspension),

4. Storage of the sample in the chemically inert container at ambient temperature.

In a similar way, for NCC, the INUIT protocol suggested:

1. One minute sonication of the original sample for initial homogenization,
2. Dilution to the desired final concentration using deionized water (18.2 MΩ cm$^{-1}$),
3. Mixing the suspension vigorously for 3 minutes using high shear mechanical stirrer,

30       homogenizer or probe sonicator to get homogenous suspension; alternatively, using an ultrasonic bath for 30 minutes in the case of sample volume <10 ml,
4. Measurements with <0.03wt% in order to diminish particle aggregation,
5. Storage of the sample in dry and cool (4 °C) environment.



The background levels of the aqueous methods are discussed in detail in **Sect. 4.5.1**. More detailed discussion regarding nominal parameters is given in **Sect. 4.5.3**.

### 3.2. Ice Nucleation Parameterization

In this section, we describe a procedure to parameterize immersion freezing abilities for both dry dispersion methods and aqueous suspension techniques. The immersion freezing data of cellulose particles in a wide range of temperatures is then discussed by comparing $n_{s,geo}(T)$ spectra from all twenty instruments. Please note that using the scaled metrics for the validation (e.g., $n_{s,geo}(T)$ scaling with the technique specific *SSA* value) is indispensable in this study because the changes or uncertainties in surface area amongst groups are an issue as described in **Sect. 3.1**. The INP concentration per volume of air ($n_{INP}(T)$, e.g., *DeMott et al.*, 2017; *Vali*, 1971) is a useful parameter for instrumental evaluation when utilizing identical samples at a single location with known sampling flows, but is not applicable in this work.

The majority of dry dispersion methods employs the approximation of *Niemand et al*. (2012). If the activated ice fraction is small (< 0.1), the Taylor series approximation can be applied, and we can estimate $n_{s,geo}(T)$:

$$n_{s,geo}(T) = -\ln\left(1 - \frac{N_{ice}(T)}{N_{total}}\right)\left(\frac{1}{S_{ve}}\right) \approx \frac{N_{ice}(T)}{N_{total}S_{ve}} = \frac{N_{ice}(T)}{S_{total}} \ , \tag{1}$$

in which $N_{ice}(T)$ is the cumulative number concentration of formed ice crystals at $T$ (cm$^{-3}$), $N_{total}$ is the total number concentration of particles prior to any freezing event (cm$^{-3}$), $S_{ve}$ is the volume equivalent surface area of an individual particle (m$^2$), and $S_{total}$ is the total surface area (m$^2$). For the LACIS data, the left part of Eqn. (1) was used without any approximation.

One distinct exception is the electrodynamic balance (EDB) method, in which the probability of contact freezing on a single collision, $e_c$, is first inverted from *FF* to take into account the rate of collision and, then, scaled to surface area of a single INP to estimate $n_{s,geo}(T)$ (*Hoffmann et al*., 2013a; 2013b):

$$n_{s,geo}(T) = \frac{e_c(T)}{k_{imm} \cdot S_{ve}} \tag{2}$$

Note that the INP particle colliding with the supercooled droplet is only partially submersed in water, and therefore the surface available for nucleation is corrected by a dimensionless factor $k_{imm}$. The value of this factor depends on the wettability of the particle surface and is generally unknown. In this work, $k_{imm} = 1$ has been assumed. The effective surface area of MCC particles has been derived from the scanning electron microscope images of the particles collected on the Nuclepore® membrane filters placed inline to the EDB, as described in **Supplemental Information**.





The results of eleven aqueous suspension methods are interpreted in terms of the frozen fraction ($FF$), INP concentration per volume of liquid ($c_{INP}$, *Vali*, 1971) and geometric size-based ice nucleation active surface-site density ($n_{s,geo}(T)$, *Connolly et al.*, 2009; H15b). The cumulative $FF$ at $T$ is:

$$FF(T) = 1 - \frac{N_u}{N},$$ (3)

where $N_u$ is the number of unfrozen droplets and $N$ is the total number of originally liquid entities. Following Eqn. 1 in *DeMott et al.* (2017), conversion to $c_{INP}$ at $T$ is expressed by

$$c_{INP}(T) = -\frac{1}{V_d} ln\left(\frac{N_u(T)}{N}\right),$$ (4)

where $V_d$ represents the individual droplet volume. Finally, the $n_{s,geo}(T)$ value as a function of

$T$ can be estimated by

$$n_{s,geo}(T) = \frac{c_{INP}(T)}{\rho_w \omega \theta},$$ (5)

where $\rho_w$ is the water density (= 997.1 g L$^{-1}$), $\omega$ is the mass ratio of analyte and water (unit-less) and $\theta$ is the $SSA$ value (m$^2$ g$^{-1}$), provided in **Tables 2 and 4**.

Accordingly, we compare the $n_{s,geo}(T)$ and $\Delta log(n_{s,geo})/\Delta T$ (i.e., the freezing spectral

slope parameter, H15b) data from our measurements to five literature results. These reference results include previously reported $n_{s,geo}(T)$ curves of illite NX particles from H15b (hereafter H15NX), MCC particles from H15a (hereafter H15MCC), Snomax (*Wex et al.*, 2015, hereafter W15), desert dusts (*Ullrich et al.*, 2017, hereafter U17) and K-feldspar (*Atkinson et al.*, 2013, hereafter A13). The $n_{s,geo}(T)$ (m$^{-2}$ as a function of °C) fits from the reference literature

are:

$$n_{s,geo}^{H15NX,dry} = \exp((27.92 \times \exp(-\exp(0.05 \times (T + 13.25)))) + 6.32),$$

$$T \in [-37, -18]; \Delta log(n_{s,geo})/\Delta T = 0.18,$$ (6)

$$n_{s,geo}^{H15NX,wet} = \exp((22.64 \times \exp(-\exp(0.16 \times (T + 20.93)))) + 5.92),$$

$$T \in [-34, -11]; \Delta log(n_{s,geo})/\Delta T = 0.37,$$ (7)

$$n_{s,geo}^{H15MCC,dry} = \exp(-0.56 \times T + 7.50),$$

$$T \in [-30, -15]; \Delta log(n_{s,geo})/\Delta T = 0.24,$$ (8)

$$n_{s,geo}^{H15MCC,wet} = \frac{2.57 \times 10^7 + \frac{-2.84 \times 10^7}{1 + \exp(\frac{-25.19 - T}{1.45})}}{SEM - based\ SSA_{MCC}},$$

$$T \in [-28, -22]; \Delta log(n_{s,geo})/\Delta T = 0.35,$$ (9)

$$n_{s,geo}^{W15} = \frac{(1.40 \times 10^{12}) \times (1 - (\exp((-2.00 \times 10^{-10}) \exp(-2.34 \times T))))}{geometric\ SSA_{Snomax}},$$

$$T \in [-38, -2]; \Delta log(n_{s,geo})/\Delta T = 0.88\ (-2\ °C < T < -10.7\ °C),$$ (10)

$$n_{s,geo}^{U17} = \exp(150.577 - (0.517 \times (T + 273.150))),$$

$$T \in [-30, -14]; \Delta log(n_{s,geo})/\Delta T = 0.22,$$ (11)





$$n_{s,\text{geo}}^{\text{A13}} = 10^4 \times \exp(-1.038(T + 273.150) + 275.260) \times \frac{BET-SSA_{K-feldspar}}{geo-SSA_{K-feldspar}},$$

$T \in [-25, -5]; \Delta\log(n_{s,\text{geo}})/\Delta T = 0.45.$ (12)

For H15MCC (wet), the $n_m(T)$ to $n_{s,\text{geo}}(T)$ conversion was performed using SEM-based $SSA$ constants of 0.068 m$^2$ g$^{-1}$. The geometric $SSA$ value of 7.99 m$^2$ g$^{-1}$ was used for W15. This $SSA$

value was derived from the polydisperse particle size distribution measurements of Snomax obtained during AIDA studies, whose IN data are included to compute immersion freezing results reported in *Wex et al.* (2015). For microcline (K-feldspar), the $n_{s,\text{geo}}(T)$ to $n_{s,\text{BET}}(T)$ conversion was performed using a laser diffraction-based geometric $SSA$ of 0.89 m$^2$ g$^{-1}$ and an N$_2$ BET-SSA of 3.2 m$^2$ g$^{-1}$ reported in *Atkinson et al.* (2013). Please note that laser diffraction

tends to be sensitive to the larger particles in a distribution, so it may miss the smaller particles and underestimate surface area.

### 3.3. Temperature Binning

A consistent data interpolation method is important to systematically compare different ice nucleation measurement methodologies as demonstrated in H15b. In this study, we present

*T*-binned average ice nucleation data (i.e., 1 °C bins for -36 °C < T < -4 °C). Unless the data were originally provided in 1 °C binned-data (i.e., weighted-average or cumulative counts) [i.e., BINARY, DFPC-ISAC, FRIDGE-CS (MCC portion), LINDA, NC State-CS, NIPR-CRAFT, WISDOM and WT-CRAFT], all data are binned in a consistent manner using either a moving average (where original data points are finer than 1 °C) or a Piecewise Cubic Hermite Interpolating Polynomial

function (where original data points are equivalent or coarser than 1 °C). For the former case, the default span for the moving average is 3. If the total number of original data points is less than 6 and the ratio of interpolated data points to original data points is larger than 0.5 (i.e., M-WT, EDB, AIDA for FC), we used the given ratio – which is specific to the technique – for the moving average span to implement the interpolation without obvious errors. The comparison

of *T*-binned immersion freezing spectra from particle dispersion methods and aqueous suspension methods is discussed in **Sect. 4.1**.

### 3.4. Surface Structure Analyses

Cellulose particles consist of a complex porous morphology with capillary spaces between the nanoscale fibrils (H15a). These surface structures may make the surface accessible to water

and induce a varying sensitivity to heterogeneous ice formation (*Page and Sear*, 2006; *Subramanyam et al.*, 2016; *Kiselev et al.*, 2016). To better understand the nanoscale surface morphology of cellulose materials, surface structures of all three cellulose materials were characterized using a scanning electron microscope (SEM, SU-3500, Hitachi). To minimize the



deformation of a specimens' surface by the intense electron beam bombardment, we purposely used an acceleration voltage of 5 keV and a working distance of 5 mm in a low vacuum mode (50 Pa). Dry MCC and FC particles from the batches were sprinkled over a carbon tape substrate. A number of SEM images (61 MCC and 62 FC particles) were afterwards

taken for randomly selected <10 μm particles with an Ultra Variable Pressure (UVD) detector at 2560 ×1920 pixel resolution. After the micrograph image acquisition, our images were analyzed to estimate the line structure density and size distribution of defects on the surface of all 123 particles. For the image processing, background signals from the carbon tape substrate in the proximity of target particles were first removed by subtracting threshold

intensities between particles and the background. Thus, particles were distinguished from the carbon tape by choosing an appropriate threshold value of image intensity to yield binary images (*Adachi et al.*, 2007 and 2018). Followed by the background correction, line structures on the particle surfaces were clipped. These line structures were typically brighter than the other areas because of their edge effects on the UVD images. Line structures with >0.25 μm

were chosen to characterize the particle surface, i.e., surface features with <0.25μm were ignored as noise because of a lack of SEM image resolution.  Afterwards, the length of individual line structures extracted from the original SEM image was measured over the entire grid along both X and Y axes. No major image distortion was observed and, hence, no corrections for curvature were applied. Lastly, the distributions of the length were integrated

for particle type (i.e., MCC and FC) to assess the overall size distributions of these surface linear peaks. Consequently, surface areas of all 123 particles were also measured from SEM images, and the abundances of the line structures were scaled to their surface area measured by SEM.

        Our attempt to facilitate SEM for NCC surface characterization was unsuccessful since our NCC sample contained fibers smaller than its spatial detection limit (~0.25 μm).

Complementarily, we employed a transmission electron microscope (TEM, JEM-1400, JEOL) to analyze the NCC surface. The NCC sample was diluted with water (0.03wt% NCC) and pipetted onto TEM grids with both formvar and lacey carbon substrates (U-1007 and U-1001, respectively; EM-Japan, Tokyo, Japan). The results of both our SEM and TEM analyses are available in **Sect. 4.4**.





### 4. Results and Discussions

#### 4.1. Dry dispersion vs. aqueous suspension methods

Temperature-binned ensemble $n_{s,geo}(T)$ spectra of MCC, FC and NCC in a temperature range between -4 and -38 °C are presented in **Fig. 4.** Different columns (**a-c**) correspond to different sample types: (a) MCC, (b) FC and (c) NCC. The top panels show a comparison between dry dispersion type measurements and aqueous suspension measurements of cellulose samples with previous parameterizations of other reference samples (**panels i**). The $n_{s,geo}(T)$ spectra from each subgroup of techniques are independently summarized in **panels ii and iii**. More detailed representations of $n_{s,geo}(T)$ spectra from individual techniques are available in **Figs. 6-8** and are discussed in **Sect. 4.3**. Lastly, the bottom panels **(panels iv)** show the overall deviation between maxima and minima of $n_{s,geo}(T)$ as pink shaded areas. As inferred from the first three panels (**i, ii and iii**), dry particle-dispersed measurements generally show higher $n_{s,geo}(T)$ values than aqueous suspension measurements above -24 °C regardless of sample types. Furthermore, as apparent in **panels iv**, the $n_{s,geo}(T)$ differences among measurements can extend up to three orders of magnitude at -20 °C (for MCC and FC) and -15 °C (for NCC), where the results from particle dispersion measurements and a majority of suspension measurements coexist.

 The observed divergence in $n_{s,geo}(T)$ is most significant at temperature higher than -24 °C, where the slope in the aqueous suspension spectra is steeper (i.e., $\Delta log(n_s)/\Delta T > 0.34$). Most aqueous suspension methods capture the abruptly increasing segment of the $n_{s,geo}(T)$ spectral slopes at -20 °C > $T$ > -25 °C. In this $T$ region, the slope is virtually identical to the slopes of wet H15NX and H15MCC spectra (0.35-0.37, **Eqns. 7 and 9**) and is also closely parallel to the A13 parameterization (0.45, **Eqn. 12**), suggesting the number of active sites are different. Likewise, our $T$-binned data from dry dispersion methods exhibit similar $n_{s,geo}(T)$ values when compared to the previous parameterizations. For instance, our dry dispersed cellulose spectra (i.e., $\Delta log(n_s)/\Delta T$ of 0.20, 0.28 and 0.22 for MCC, FC and NCC) present comparable trends to the dry H15 curves (0.18-0.24, **Eqns. 7 and 9**) and U17 parameterization (0.22, **Eqn. 11**).

 It is interesting that a similar difference between dry dispersion and aqueous suspension results (i.e., $n_{s,geo}(T)$ of dry dispersed particle > $n_{s,geo}(T)$ of suspension results) is made by previous inter-comparison activities with mineralogically heterogeneous dust particles (*Emersic et al.*, 2015; H15b). In brief, *Emersic et al.* (2015) reports the dry dispersion chamber-measured $n_{s,geo}(T)$ can be up to a factor of 1000 larger than the cold stage results for multiple mineral dust samples, including illite NX, Kaolinite and K-feldspar. Our previous study also shows that $n_{s,geo}(T)$ of illite NX increases sharply at colder temperatures in the $T$ range





from -18 °C to -27 °C, followed by the leveling off segment at the low temperature region. It is certainly common for the $n_{s,geo}(T)$ spectrum to level off at the $n_{s,geo}(T)$ maxima. As mentioned in **Sect. 1.2.**, several studies (*Emersic et al.*, 2015; *Beydoun et al.*, 2016) reported the mechanism of the observed divergence between two subsets of methods. Nonetheless, the

reduction in the slope of $n_{s,geo}(T)$ spectrum may be a plausible contributor to the higher reported $n_{s,geo}(T)$ values in some aqueous suspension measurement results (WISDOM, CMU-CS in **Sect. 4.3**), which are comparable to the dry dispersion results (i.e., data of freezing of individual droplets containing a single aerosol particle) for illite NX and cellulose (*Beydoun et al.*, 2016).

Next, **Fig. 5** depicts the $n_{s,geo}(T)$ diversity in $\log(n_{s,ind.})/\log(n_{s,avg})$, which represents the ratio of the individual measurements ($n_{s,ind}$) to the log average of $n_{s,geo}(T)$ ($n_{s,avg}$) at given temperatures. In other words, this figure provides an overview of the $n_{s,geo}(T)$ deviations across the various techniques employed in this work. These $n_s$ ratios are shown for the temperature range covered by at least two measurement techniques used in the present study. In this

figure, different panels show three different $n_{s,avg}$ values as denominators, including the average based on all bulk data (All, **panels i, ii and iii**), dry dispersion subgroup (Dry, **panels iv**), or aqueous suspension subgroup (Sus, **panel v**). As for numerators ($n_{s,ind}$), the interpolated $T$-binned data (1 °C) from **Fig. 4** are used. A total of five panels are presented. First, a summary comparison of two method categories (dry dispersion and aqueous suspension) in a

temperature range of -33 °C < T < -15 °C is given in the top panels (**panels i**). As shown in these panels, data deviation (i.e., scatter from the average $\log(n_{s,ind.})/\log(n_{s,avg})$ = 1 line) can be seen in both dry dispersion and aqueous suspension measurements. Other panels provide more evidence on the measurement diversity. In short, while the $\log(n_{s,ind.})/\log(n_{s,avg})$ values range within 0.8-1.2 for Dry Dispersion (DD) and Aqueous Suspension (AS) cases (**panels iv and v**),

more prominent scatter of the $\log(n_{s,ind.})/\log(n_{s,avg})$ values (0.6-1.4) is seen when All is used as $n_{s,avg}$ values (**panels i, ii and iii**). Thus, the observed deviation is the largest with $n_{s,avg}$ of All (i.e., both AS and DD). Furthermore, the deviation becomes more apparent towards higher temperatures. This trend persists regardless of sample type. We will discuss potential explanations for the observed diversity of data from different techniques in the following

section.

### 4.2. Inter-comparison of three sample types

The multiple exponential distribution fits (also known as the Gumbel cumulative distribution function) for $T$-binned data of all three cellulose samples are summarized in **Table 7**. Fit parameters as well as $\Delta\log(n_s)/\Delta T$ for each category are given in this table. As can be



inferred from the table, the overall $\Delta\log(n_s)/\Delta T$ value is almost identical for all three sample types (0.31-0.33) in spite of some deviations observed for min-max (0.26-0.40). The observed consistency in the spectral slopes suggests cellulose material contains relatively similar ice nucleation efficiency across the heterogeneous freezing $T$.

For all cellulose types, a reasonable correlation coefficient ($r$) is found for each portion of techniques (i.e., DD and AS), suggesting reasonable agreement and consistency for the results from a similar group of immersion freezing techniques. However, we must reiterate the discrepancy between DD and AS. For instance, our observation of lower values of DD slopes (0.20-0.29) as compared to those of AS slopes (0.29-0.37) in the similar temperature
range suggests distinct differences between two subsets of methods. Moreover, the dry dispersed-MCC shows relatively lower $\Delta\log(n_s)/\Delta T$ of 0.20 than FC and NCC (note not all instruments delivered FC and NCC measurements, see **Table 2**). This exception potentially indicates fundamental difference of dry dispersed-MCC from other sample types.

    **Table 8** provides the log average of $T$-binned $n_{s,geo}(T)$ values for all of the cellulose
samples, representing detailed comparisons of MCC, FC and NCC. **Figure S2** also summarizes the comparison between the averages for each material (see Supplemental Information for details). As seen in the table and figure, there exists a discrepancy between this study and previous work for MCC. At -28 °C, for example, our log average $n_{s,geo}(T)$ of MCC (3.25 x 10$^9$ m$^{-2}$, **Table 8**) is smaller than the previous MCC result at the same $T$ (1.18 x 10$^{10}$ m$^{-2}$, H15a). This
difference possibly reflects the fact that our average $n_{s,geo}(T)$ includes the results from a multitude of aqueous suspension measurements, which typically fall in the lower range of DD measurements (**Sect. 4.1**), while H15MCC (**Eqn. 9**) is derived from a dry dispersion method only. Note that the $n_{s,geo}(T)$ maxima from **Table 8** and **Fig. S2** reasonably overlap with the H15MCC parameterization.

The highest $n_{s,geo}(T)$ value of the FC experiments (3.6 x 10$^{10}$ m$^{-2}$ at -29 °C from AIDA) is somewhat lower than that of MCC. Similarly, the highest $n_{s,geo}(T)$ value of the NCC experiments (1.5 x 10$^{10}$ m$^{-2}$ at -35 °C from WISDOM) is an order magnitude lower than that of MCC as well as W15.

    **Table 8** (and **Fig. S2**) also implies that MCC possesses higher ice nucleation efficiency
relative to the other two types. First, at above -25 °C, the immersion freezing ability of MCC typically exceed that of NCC. Second, at -22 to -24 °C, where more than seven instruments are involved to calculate the average $T$-binned $n_{s,geo}(T)$, MCC's $n_{s,geo}(T)$ is consistently one order magnitude higher than FC and NCC. Third, when compared to FC, MCC generally possess slightly higher $n_{s,geo}(T)$ at $T$ below -16 °C. Likewise, a similar trend holds true when we compare
MCC to NCC at $T$ below -17 °C. The observed difference is up to two orders of magnitude at -



20 °C. Please note that, at the high $T$ region (> -17 °C), dry dispersion techniques are not sensitive enough to detect INPs with their experimental parameters used in this study (**Tables 3 and 5**). In contrast, detecting rare INPs by increasing the concentration of the aqueous particle suspension is advantageous yet also challenging. In other words, the measurement

uncertainties generally propagate towards high temperatures because the confidence interval is relatively wider when there exists only a few frozen droplets. Hence, our observation of less immersion freezing ability of MCC at this $T$ range (up to a factor of ~20 at -16 °C) may not be conclusive. Particle sedimentation, aggregation and the concentrations effects identified by *Beydoun et al*. (2016) are also more prominent at higher concentration, especially for cellulose

samples.

### 4.3. Individual immersion freezing measurements

All individual $n_{s,\text{geo}}(T)$ spectra of MCC, FC and NCC from each technique are shown in **Figs. 6, 7 and 8**, respectively. Since the primary focus of this study is on the methods inter-comparison, only brief remarks regarding each technique are summarized below. Several special

experiments were carried out using seven techniques to complement our understanding of cellulose ice nucleation. The results from these unique experiments are first described (**Sects. 4.3.1-4.3.7**) followed by the other remarks (**Sects. 4.3.8-4.3.19**).

#### 4.3.1. CSU-CFDC

          Immersion freezing ability of both polydisperse and quasi-monodisperse dry dispersed

MCC particles were characterized by CSU-CFDC. In short, ice-nucleating efficiencies of DMA size-selected MCC particles (500 nm mobility diameter) were compared to that of the polydisperse population for immersion freezing experiments.

          As seen in the **Fig. 6b**, the discrepancy between the results from two populations is substantial. Similar to the LACIS result, a weak temperature dependence of $n_{s,\text{geo}}(T)$ of

monodisperse MCC particles is observed within defined experimental uncertainties (see **Table 3**). Observed quasi-flat $\Delta\log(n_{s,\text{geo}})/\Delta T$ of the monodisperse case suggests a week $T$-dependent immersion freezing ability of given specific size of MCC particles for the investigated temperature range. Conversely, a polydisperse spectrum, which represents the result of an ensemble of different MCC particle sizes, shows a stronger trend of the slope towards low $T$

segment, suggesting a non-uniform distribution of active sites over the available $S_{\text{total}}$ of cellulose in this study. Some previous INUIT studies demonstrated the size independence of the $n_{s,\text{geo}}(T)$ value using submicron hematite and illite NX particles based on AIDA ice nucleation


experiments (*Hiranuma et al.*, 2014 and 2015b). Such a characteristic may not remain true for the immersion mode freezing of giant fiber particles.

For all sample types, as seen in **Figs 6b, 7b and 8b**, the CSU-CFDC results do not agree well with H15a (MCC_dry, **Eqn. 8**). Instead, they virtually agree with the wet generation

results. This is especially true for the results with polydisperse population. Note that formerly observed agreement within a factor of three in $n_{s,geo}(T)$ estimation (cloud simulation chamber INAS > CSU-CFDC INAS; *DeMott et al.*, 2015) is seen only at -30 °C. The observed discrepancy may be due to non-uniform active site density for different sizes. Another possible explanation may be due to the alternation of cellulose physico-chemical properties perhaps upon

humidification during shipping, causing behaviour more like aqueous suspended particles. One thing that we need to keep in mind is that the CFDC uses a 2.4 μm particle impactor at its inlet (**Table 5**). Because of the impactor, there is loss of larger particles. Thus, the $n_{s,geo}(T)$ results may vary, possibly due to the difference in the size of cellulose samples examined. At -23 °C, where the data of size-selected measurements exist for all three cellulose samples, CSU-

CFDC show $n_{s,geo,MCC} \approx n_{s,geo,FC} > n_{s,geo,NCC}$ (**Figs. 6b, 7b and 8b**).

### 4.3.2. DFPC-ISAC

The DFPC-ISAC instrument (*Santachiara et al.* 2010) provided data for condensation/immersion freezing. The use of 103% $RH_w$ in this investigation was optimized to count statistically significant amount of INPs in this system for examined cellulose particles

(i.e., MCC and FC). With this system, we assessed the IN efficiencies of different sizes of MCC and FC particles generated by means of different cyclone cut-sizes (0.5, 1.0, 7.0 μm or none). Further, both dry dispersed (Dry) and nebulizer-generated particles (Wet) were systematically assessed for their INP activities. Without an exception, INP concentrations were measured at -22°C for all specimens. For the case of particles (<0.5 μm cyclone-selected), we additionally

measured INP concentrations at -18 °C to assess the general trend of the INP activates as a function of *T*. This particular case was selected for the extended study due to the similarity of their geometric *SSAs* to those of the AIDA cloud parcel simulation measurements. In addition, while collecting the cellulose particles on nitrate membrane filters (Millipore, 0.47 um pore size) used for IN assessment, parallel measurements of particle size distributions using an

optical particle counter (Grimm, 1.108) were carried out. The results of size distributions, represented by the *SSA* values, are summarized in **Table 9**.

For Dry, increasing the cut-size tends to decrease the *SSA* value, implying large particles come through, and the dominance of the mass relative to the surface becomes significant. This observation is valid as the cyclone is used to remove particles larger than the



designated cut-size. Regardless of whether using the cyclone or not, particle sizes out of the nebulizer-generation is somehow comparable to that of Dry dispersion with a cyclone of 1 μm cut-size. The observed difference between Wet and Dry is indicative of the changes in particle size and morphology while drying atomized particles from a suspension of the powder in water

as described in **Sect. 2.2**.

**Figures 6c and 7c** show all the results of INP measurements by DFPC-ISAC. For MCC, the interpolated DFPC results of the immersed particles (<0.5 μm cyclone-selected) falls in the middle of FRIDGE results of two different modes for -22 °C < $T$ < -18 °C. More interestingly, the slope of the DFPC $n_{s,\text{geo}}(T)$ spectrum ($\Delta\log(n_{s,\text{geo}})/\Delta T = 0.24$) represents the median of the slopes

of FRIDGE measurements (i.e., 0.17 for default mode and 0.31 for immersion mode). This observation is consistent with other results of (1) size-selected particles tend to exhibit a gentle slope (similar to the observations from CFDC and LACIS) and (2) nebulizer-generated techniques tend to result in a deteriorated INP activity (H15b).

Another important implication of the DFPC results is the fact that submicron dry

particles show the highest INP efficiencies, practically lie on $n_{s,\text{geo}}(T)$ data points of H15a parameterization at given $T$ for both MCC and FC. Moreover, inclusion of supermicron sizes (no cyclone or 7 μm) seems reducing IN efficiencies of both MCC and FC. Further investigation is required to interpret these results.

Over the temperature range of -18 to -22 °C, the DFPC results of immersed particles

(<0.5 μm cyclone-selected), show $n_{s,\text{geo,FC}} \approx n_{s,\text{geo,MCC}}$ (**Figs. 6c and 7c**). Note that $n_{s,\text{geo,FC}}$ appears to be slightly higher than $n_{s,\text{geo,MCC}}$. This observation is not consistent with the general trend of $n_{s,\text{geo,MCC}} > n_{s,\text{geo,FC}}$ (**Sect. 4.2**). However, the observed difference is only a factor of <2 on average.

### 4.3.3. FRIDGE

The FRIDGE data were derived from both default mode (a combination of deposition, condensation ice nucleation and immersion freezing at $RH_w$ of 101%) and immersion mode operation for MCC. With these two different operational modes, FRIDGE investigated the ice nucleation ability of both dry and droplet suspended particles deposited on a substrate. Particularly, the default mode operation of FRIDGE provided data from -16 to -30 °C (MCC) by

scanning $RH_{\text{ice}}$ and $RH_w$ (low to high) at a constant temperature. Accordingly, ice crystals formed at the highest $RH_w$ of 101% were considered as a measure of immersion $N_{\text{ice}}$ from dry dispersed particle measurements. Likewise, the immersion mode operation of FRIDGE provided data from -19 to -28 °C (MCC) and from -13 to -23 °C (NCC). As demonstrated in H15b, this immersion mode counts immersion freezing of suspended particles in which the particles





are first washed into droplets and then placed on the substrate to be comparable to the dry dispersion method. Hence, this method is advantageous to collect a filter sample of cellulose, prepared the same way as in the dry dispersion experiment, and then run it on a cold-stage.

**Figure 6e** shows the comparison of $n_{s,geo}(T)$ derived from the two different operation modes of FRIDGE. There are a few important implications from the FRIDGE results. First, on average, the measurements with dry particles in the 'default' setting showed more than an order of magnitude higher $n_{s,geo}(T)$ in comparison to the immersed particles in FRIDGE experiments at $T$ > -22 °C. As shown in **Fig. 6e**, the deposition mode data suggest that $n_{s,geo}(T)$ values for -22 °C < $T$ < -19 °C are close (within a factor of two) to those from MRI-DCECC, in which experiments were carried out with a high degree of particle agglomeration. In comparison to the default mode result, FRIDGE experiments in the pure immersion mode showed much lower $n_{s,geo}(T)$ than that with the default setting, but agreed with other immersion datasets. Second, a steeper $\Delta\log(n_{s,geo})/\Delta T$ of 0.31 was found for the measurements with immersed particles at $T$ > -24 °C when compared to the slope of the deposition mode data (i.e., 0.17). As a temperature shift (i.e., shifting the data a few °C) does not offset the discrepancy, other mechanistic interpretations might be plausible causes of this discrepancy. For instance, this difference may be a consequence of the different IN efficiencies of nucleation modes of both experimental approaches (e.g., deposition + condensation + immersion vs. immersion alone) in the examined temperature range, the different sample preparation processes, effects of agglomeration or a combination of the three. The divergence of default-mode and CS-mode becomes notable $T$ > -24 °C, perhaps suggesting the effect of agglomeration. Specifically agglomeration may take place inside the pipetted droplets. While pipetting agglomeration and separation is avoided by shaking the sample, but during cooling it lasts 15-30min until a droplet freezes.

**Figure 8c** presents the summary of FRIDGE-CS measurements for NCC. The $n_{s,geo}(T)$ spectrum nearly overlaps with the H15b (illite NX wet) reference spectrum. It also agrees well with the other droplet freezing instruments CMU-CS, NIPR-CRAFT, NCS-CS, BINARY and WISDOM. Similar $n_{s,geo}(T)$ values were obtained although the methods analysed droplets of different volumes. In particular FRIDGE and WISDOM $n_{s,geo}(T)$ attach to each other better than 0.3 °C. By comparing NCC to MCC at -23 °C < $T$ < -19 °C, the FRIDGE-CS results show $n_{s,geo,MCC}$ > $n_{s,geo,NCC}$ for >one order magnitude throughout this overlapping $T$ range. Note that the $\Delta\log(n_{s,geo})/\Delta T$ value of NCC (0.40) is somewhat higher than the average slope parameters listed in **Table 7**.

### 4.3.4. LACIS

With LACIS, we examined immersion mode freezing of both atomized and dry dispersed MCC particles separately. For atomized particle generation, particles were dried in a diffusion dryer directly after spraying the suspension. Succinctly, LACIS measured immersion ice nucleation

ability of atomizer-generated MCC particles for 700 nm mobility diameters in the temperature range of -35 °C < $T$ < -30 °C. The selection of this relatively large size was necessary to get a signal above the limit of detection in the system. The experiments with dry dispersed MCC were performed with polydisperse MCC particles for -36 °C < $T$ < -27 °C. Note that a cyclone was used in the air stream of LACIS (see **Table 5**).

Generally, LACIS measurements with dry dispersed MCC particles are in agreement with that from H15a as apparent in **Fig. 6g** ($n_{s,geo}$(-30 °C) ~ 1.5 x $10^{10}$ $m^{-2}$). Furthermore, LACIS measurements down to -36 °C with dry polydisperse MCC particles show that $\Delta \log(n_{s,geo})/\Delta T$ (= 0.17, **Table 3**) is identical to MRI-DCECC for -28 °C < $T$ < -16 °C. Contrastively, the slope of the spectrum for 700 nm size-segregated MCC particles (= 0.05) is considerably lower than

that of the polydisperse case. This slope of the LACIS $n_{s,geo}(T)$ spectrum is parallel to that of the CSU-CFDC spectrum (dry dispersed 500 nm case, slope = 0.05 for -30 °C < $T$ < -24 °C; **Fig. 6b**). Thus, though we cannot certainly define the relative importance of the aerosol generation method (e.g., the changes in physico-chemical properties of particles occurred during atomization as prescribed in **Sect. 2.2**), the aerosol size might have a non-negligible impact on

the variation in spectral slopes. Therefore, the immersion freezing efficiency of MCC particles likely is different for differently sized MCC particles, meaning that a single $n_{s,geo}(T)$ curve cannot be reported for MCC. With this, the method of accounting for differences in surface area between different groups/methods becomes questionable for a complex system like cellulose. Furthermore, its complex morphology (**Sect. 4.4**) causes that the determination of the surface

area is quite prone to errors which can be a reason for the observed differences in $n_{s,geo}(T)$. The $n_s$ framework must be rigorously tested with more empirical data. Nevertheless, for LACIS, both polydisperse and quasi-monodisperse MCC particles exhibit similar $n_{s,geo}(T)$ values above -30 °C (e.g., $n_{s,geo}$(-30 °C) ~ 1.5 x $10^{10}$ $m^{-2}$ in **Fig. 6g**), suggesting a negligible size dependency of $n_{s,geo}(T)$ for MCC particles in this temperature range.

### 4.3.5. LINDA

This vial-based immersion freezing assay was utilized to compare the freezing activity of bulk suspension (0.1 wt% cellulose in NaCl solvent) to that of dry powders individually suspended in each vial (sus vs. pow henceforth). Such comparison was carried out to ensure that employing different methods of vial preparation did not impede ice nucleation of cellulose





samples, including MCC and FC. For the latter procedure (pow), pre-weighed cellulose powders (0.2 mg) were directly poured into 200 mg (199.8 µL) of 0.1% NaCl solution to realize the concentration of cellulose in each vial to be equivalent to 0.1 wt%, such that two procedures became comparable. We note that all vials, regardless of the procedure, were

sonicated (46 kHz) for 5 minutes prior to each LINDA measurement. Note that we used non-sterile NCC (NCC01) for the IN characterization with LINDA.

The results of MCC and FC are shown in **Figs. 6m and 7g**. The results suggest similarity of $n_{s,geo}(T)$ within the experimental uncertainties of LINDA (*Stopelli et al.*, 2014) for the range of examined temperatures (-7 °C to -18 °C). Further, the slope of LINDA $n_{s,geo}(T)$ spectra

($\Delta log(n_{s,geo})/\Delta T$) of 0.29 is identical for both scenario cases (i.e., sus and pow). Hence, for given mass concentration of 0.1 wt%, both vial preparation procedures seem valid. Nonetheless, suspended cellulose powders settle rapidly in both cases, implying the necessity of taking a great care when measuring INP activity of giant particles with the ~200 µL vial-based assay.

For -18 °C < T < -12 °C, the LINDA results (bulk suspension) show $n_{s,geo,MCC} > n_{s,geo,FC}$ with

similar $\Delta log(n_{s,geo})/\Delta T$ (0.29-0.30), verifying comparable performance of this vial-based technique to other suspension methods (**Figs. 6m and 7g**).

**Figure 8f** shows the freezing spectrum of NCC01 with the slope parameter ($\Delta log(n_{s,geo})/\Delta T$) of 0.21. The observation of higher activity of NCC01 compared to MCC and FC implies possible inclusion of INA materials in the original 3% solution of NCC01. The source is

not known, and the source identification is beyond the scope of this inter-comparison work. The sample stability of another NCC sample from another batch, NCC02, is discussed in **Sect. 4.3.6**.

### 4.3.6. NIPR-CRAFT

This suite of cold stage instruments offered the immersion freezing measurements of all three

cellulose samples using droplets with volumes of 5 µL. This microliter range volume was the largest amongst all aqueous suspension techniques employed within this work. Such a large drop volume advantageously enables high resolution immersion freezing analysis for a wide range of temperatures (-31 °C < T < -17 °C). The highest freezing temperatures are attained with the largest droplets, which contain the largest surface area of cellulose.

By means of Stokes-law gravity differential settling (*Tobo*, 2016), <10 µm MCC and FC particles of were extracted to generate droplets containing size-segregated cellulose samples. These droplets were subsequently assessed on NIPR-CRAFT, estimating an immersion freezing ability of MCC and FC with *SSA* of 3.35 m² g⁻¹ (The AIDA-derived geometric *SSA* value, accounting for only <10 µm particles). Afterwards, the obtained results of <10 µm were



compared to those of bulk (SEM-based *SSA* of 0.068-0.087 m$^2$ g$^{-1}$). Furthermore, we facilitated NIPR-CRAFT for the quality check of the NCC sample over time. Expressly, we stored NCC02 at 4 °C for 9 months and made follow-up measurements to examine the potential decay of the samples, potentially altering its immersion freezing.

**Figures 6q and 7k** show the NIPR-CRAFT results for MCC and FC. In general, the NIPR-CRAFT data represent the lower boundary of compiled $n_{s,geo}(T)$ spectrum defined by the bulk of the instruments (**Figs. 5.a.iii and 5.b.iii**). Constant offset between NIPR-CRAFT and the log average of AS methods in $n_{s,geo}(T)$ is seen at -28 °C < $T$ < -21 °C for on average a factor of >9 for MCC and >2.7 for FC. Immersion freezing abilities of bulk and size-segregated samples are in

agreement within the measurement uncertainties. The spectral slopes for bulk MCC and FC are 0.41 and 0.39, respectively, and are in agreements with WT-CRAFT (measurements with 3 µL sonicated samples), indicating the presence of systematic error (e.g., temperature shift towards the low end). The spectral slopes for size-segregated MCC and FC are 0.43 and 0.34, respectively, and are in agreement with bulk NIPR-CRAFT.

**Figure 8i** shows time-trials of NCC02 and similarity in IN activity over 9 months. As inferred from the overlapped spectra, the influence of the decay over time is negligible. Over the time, the spectral slopes and $n_{s,geo}(T)$ remain similar, indicating high stability of NCC02.

       For investigated temperatures listed in **Table 2**, the bulk NIPR-CRAFT results show $n_{s,geo,MCC} > n_{s,geo,FC}$ (**Figs. 6q and 7k**). Corresponding $\Delta\log(n_{s,geo})/\Delta T$ values are similar (0.41 for

MCC and 0.39 for FC) but notably higher than any averaged slope parameters listed in **Table 7**. With even higher slope value of 0.50, the $n_{s,geo,NCC}$ values exceed both $n_{s,geo,MCC}$ and $n_{s,geo,FC}$ at $T$ below -20 °C (**Fig. 8i**).

### 4.3.7. WT-CRAFT

The WT-CRAFT system, which is a replica of NIPR-CRAFT (*Tobo*, 2016), measured the freezing

abilities of droplets containing 0.05-0.0005 wt% MCC and FC at $T$ > -26 °C. WT-CRAFT also examined if the pre-treatment of aqueous suspension (i.e., sonication of 50 mL falcon tube for 15 min) has any influences on IN efficiency of MCC and FC. More specifically, we compared the IN efficiency of 49 drops made out of the sonicated-suspension containing given wt% of MCC and FC to those of non-sonicated suspension left idle for at least 60 min.

The results are shown in **Figs. 6s and 7l**. As seen in these figures, early freezer only appears in the case of pre-application of sonication. This trend is especially notable for the MCC case. As a result, the difference of the spectral slope for MCC deviates from 0.36 (sonicated-case) to 0.52 non-sonicated case). Importantly, our results suggest that MCC may suffer more from the particle settling in the suspension when compared to FC for examined



ranges of temperature and wt%. Nevertheless, the difference in $n_{s,geo}(T)$ is within a factor of four at the most, which is well within our experimental uncertainty (see **Table 4**).

Below -22 °C, WT-CRAFT shows $n_{s,geo,MCC} > n_{s,geo,FC}$ (**Figs. 6s and 7l**). The MCC result exhibits sharper increase in $n_{s,geo}(T)$ within the limited temperature range with $\Delta\log(n_{s,geo})/\Delta T$

of 0.36 than FC ($\Delta\log(n_{s,geo})/\Delta T = 0.30$).

### 4.3.8. AIDA

The AIDA facility at KIT represents the world's foremost facility for studying ice clouds in a controlled setting. As shown in **Fig. 5**, for all cellulose types, the AIDA data hover in the upper bound of comprehensive $n_{s,geo}(T)$ spectrum defined by the bulk of the instruments. The

corresponding $\log(n_{s,ind.})/\log(n_{s,avg})$ is within 1.2. The spectral slope for immersion freezing of cellulose from AIDA varies depending on the sample type. For MCC, $\Delta\log(n_{s,geo})/\Delta T$ is 0.24 and equivalent to that of H15a (MCC, dry, **Eqn. 8**). The larger slope value is found for FC (0.47), which is practically parallel to A13 (0.45), and deviating from other DD instrument ($\Delta\log(n_{s,geo})/\Delta T$ of 0.28). But, the $n_{s,geo}(T)$ data of FC form AIDA are in fair agreement with the

log $n_{s,geo}(T)$ average for examined $T$. Finally, the NCC02 results agree well with CSU-CFDC and WISDOM. Observed quasi-flat $\Delta\log(n_{s,geo})/\Delta T$ of NCC02 (0.04) suggests a week $T$-dependent immersion freezing ability for the investigated temperature range. In addition, similar to the observation made by LINDA, higher activity of NCC01 compared to NCC02 is seen in **Fig. 8a**. This difference suggests the inclusion of INA materials in the original 3% solution of NCC01 (the

source is not known). For investigated temperatures listed in **Table 2**, AIDA show $n_{s,geo,MCC} > n_{s,geo,FC}$ and $n_{s,geo,MCC} > n_{s,geo,NCC}$ (**Figs. 6a, 7a and 8a**).

### 4.3.9. EDB
The contact freezing experiments have been performed with MCC particles preselected in DMA at two electrical mobility diameters: 320 nm and 800 nm. Due to the low concentration

(typically less than 30 cm$^{-3}$) of the MCC particles produced by the dry dispersion method (a turbulent flow disperser, see **Table 5**), and relatively low IN efficacy of MCC particles, the measurements of $e_c$ were possible only in a limited temperature range between -29°C and -32 °C. A strong asphericity of the MCC particles contributes to the uncertainty of $n_{s,geo}(T)$ determination, which differs by two orders of magnitude for particles with mobility diameters

of 320 nm and 800 nm. Additional uncertainty factor is the unknown portion of the MCC particle submersed in water upon contact with the supercooled droplet ($k_{imm}$, see **Eqn. 2**). We set $k_{imm} = 1$ thus giving a lower estimate of the possible $n_{s,geo}(T)$ value. On the whole, the contact INAS density falls nicely within the range of $n_{s,geo}(T)$ values measured by other





instruments, but does not exceed H15MCC parametrization for dry NCC particles. This is not very surprising given the experimental uncertainties of the EDB-based method.

### 4.3.10. INKA

INKA (Ice Nucleation Instrument of the Karlsruhe Institute of Technology; *Schiebel*, 2017) is a
cylindrical continuous flow diffusion chamber built after the design of the CSU-CFDC (*Richardson*, 2010), but with a prolonged residence time of the sample (*Chen et al.*, 2000). Using INKA, we studied the condensation / immersion freezing of MCC, which was dry dispersed into a 4 m³ stainless steel tank using the same procedure as for the AIDA experiments. No additional impactor was used at the INKA inlet.

10        The aerosol freezing ability was measured from -32.5 °C to -25 °C for increasing relative humidity from well below liquid water saturation to about 110% RH in a total of eight scans. Data reported in this paper was interpolated at a relative humidity of 107% RH, taking into account that the nominal relative humidity for CFDCs has to be above 100% in order to enable full aerosol activation (*DeMott et al.*, 2015; *Garimella et al.*, 2017). INKA measured ice
nucleation surface site densities which are close to the average of all measured data (see **Fig. 5**). The results match the data measured by the CSU-CDFDC for polydisperse aerosol, with slightly less pronounced temperature dependence.

### 4.3.11. MRI

MRI cloud simulation chamber experiments were conducted to demonstrate that MCC
particles can act as efficient immersion freezing nuclei in simulated supercooled clouds. The evacuation rate was correspondent to the updraft velocity of 5 m s⁻¹. Dry MCC powders were dispersed by a rotating brush generator (PALAS, RBG1000) and injected into the ventilated 1.4 m³ chamber vessel. Using the data from six experiments, we calculated the ice nucleation active surface-site densities of aerosolized cellulose in the temperature range from -15 °C to -
30 °C. The regression line for the experimental data is $n_{s,geo}(T) = \exp(-0.56T + 7.50)$ with a correlation coefficient of 0.84. As shown in **Figs. 5 and 6h**, for dry MCC type, the MRI cloud simulation chamber data exist in the upper bound of comprehensive $n_{s,geo}(T)$ spectrum.

### 4.3.12. PNNL-CIC

Immersion freezing properties of size-selected MCC samples at a temperature ranging from -
20 to -28 °C were investigated. The chamber was operated at $RH_w = 106 \pm 3\%$, and the evaporation section of the chamber was maintained at aerosol lamina temperature. The uncertainty (±0.5 °C) in the aerosol lamina temperature was calculated based on aerosol lamina profile calculations. $n_{s,geo}(T)$ calculations were performed using immersion freezing frozen fraction and surface area of MCC particles. The $n_{s,geo}(T)$ values varied from 1 x 10⁸ to 1



x $10^9$ m$^{-2}$. $\Delta\log(n_{s,geo})/\Delta T$ (= 0.13, **Fig. 6i**) agreed well with that of the U17 dust parameterization in the same temperature range.

### 4.3.13. BINARY

The three different cellulose types were investigated with the BINARY setup (*Budke and Koop*,

2015), and their sample preparation is described in **Table 6**. We note that the MCC and FC original data are those published in H15a, i.e., before the recommended suspension preparation procedure was developed. As described in H15a these bulk suspensions suffered from sedimentation and, hence, are not predestined for a $n_{s,geo}(T)$ inter-comparison. The original raw data from H15a were re-analyzed here in order to have the same 1 °C binning and

averaging as other techniques. Moreover, a different background correction was applied, also to the NCC samples: the first 5% and last 5% of nucleation data points in a given frozen fraction curve (i.e. the data smaller than 0.05 and greater than 0.95 in *FF*) were excluded, in order to account for a concentration variation between individual droplets due to sedimentation and for nucleation events triggered by the glass substrate or impurities in the "pure" water

background.

For -25 °C < T < -22 °C, the bulk BINARY data for the different cellulose samples are in a similar active site range, i.e. the results show $n_{s,geo,MCC} > n_{s,geo,FC} \approx n_{s,geo,NCC}$ (**Figs. 6j, 7d and 8d**). At -25 °C the MCC and FC data show a rapid change in slope and at lower temperature they level off at a $n_{s,geo}(T)$ value of about $10^8$ m$^{-2}$, which may be due to the sedimentation of

cellulose particles with lower ice nucleation activity as discussed above. In contrast, no such change in slope is observed for NCC (which did not suffer from apparent sedimentation), thus being consistent with higher $n_{s,geo,NCC}$ values observed below -25 °C in small-droplet experiments and dry suspension techniques. Moreover, above -25 °C the NCC data agree well with other large-volume droplet experiments such as NIPR-CRAFT and NC-State CS as well as

with small-droplet techniques such as WISDOM. In summary, these observations imply that techniques using large droplets may suffer from sedimentation if the suspended material consists of particles with a wide size distribution. However, if smaller and homogeneous particles are suspended they give results similar to small droplet techniques.

### 4.3.14. CMU-CS

The immersion freezing ability of wide range of aqueous suspension concentrations and immersion freezing temperatures was measured by CMU-CS (*Polen et al., 2016*; *Beydoun et al.*, 2017; *Polen et al.*, 2018). This cold stage device facilitates the sampling of drops within a squalene oil matrix that allows for experiments using varied wt% of the cellulose test samples



(0.001 to 0.15 wt%) for this study. Drops containing MCC, FC and NCC02 were studied at a cooling rate of 1 °C min$^{-1}$ to determine the immersion freezing temperature spectrum.

A total of 10 immersion mode freezing experiments with a droplet volume of 0.1 μL were performed. Using this instrument, a wide range of temperatures was investigated ($T$ > -

30 °C) yielding $n_{s,geo}(T)$ values ranging from $10^5$ to $10^{10}$ m$^{-2}$. The data from the ten individual runs collapsed into a single $n_{s,geo}(T)$ spectrum suggesting that the mass loading of dust in the droplet did not affect the measurements for the wt% values investigated. For MCC, the data are in fair quantitative agreement with the H15a (Dry MCC) parameterization at temperatures below -25 °C. The $n_{s,geo}(T)$ values of both FC and NCC are about one order magnitude lower

than the MCC $n_{s,geo}(T)$ values, agreeing with a general trend and overlapping with the Wet MCC reference curve.

Remarkably, the CMU-CS data show that the value of $\Delta\log(n_{s,geo})/\Delta T$ for MCC (= 0.20, **Table 4**) is the least amongst the aqueous suspension techniques and the closet to the results of the bulk dry techniques (the DD slope = 0.20, **Table 7**), potentially suggesting a similar and

more atmospherically representative experimental condition (less particle inclusion in a single droplet) when compared to other aqueous methods.

At -25 °C, where the immersion freezing abilities of all three cellulose samples were assessed, the CMU-CS result shows $n_{s,geo,MCC} > n_{s,geo,NCC} > n_{s,geo,FC}$ (**Figs. 6k, 7e and 8e**). Note that MCC and FC exhibit broad $n_{s,geo}(T)$ spectra with the $\Delta\log(n_{s,geo})/\Delta T$ values of 0.20 (MCC) and

0.34 (FC), detecting ice nucleation at <-29 °C, whereas the NCC spectrum spans for limited $T$ range (-25 °C < $T$ < -22 °C) with the $\Delta\log(n_{s,geo})/\Delta T$ value of 0.51. The observed widening of the spectra and detection temperature sensitivity suggests that giant particles have increased diversity in immersion freezing as compared to submicron particles.

### 4.3.15. Leeds-NIPI

μL-NIPI is a droplet freezing device which controls the temperature of 1 μL water droplets supported on a hydrophobic glass slide and monitors freezing in those droplets (Whale et al. 2015). For this study, 0.1 wt% suspensions of FC and MCC cellulose were made up in Milli-Q water by stirring for 30 minutes in glass vials. The suspensions were then stirred continuously while 1 μL droplets were pipetted onto a hydrophobic glass slide using an electronic pipette.

Droplets were then cooled from room temperature (~18 °C) at a rate of 1 °C min$^{-1}$ until they froze, freezing being monitored by a digital camera. A gentle flow of dry nitrogen was passed over the droplets to ensure that ice did not grow across the hydrophobic slide and cause unwanted droplet freezing. Temperature error for the instrument has been estimated at ± 0.4 °C and $n_{s,geo}(T)$ error bars were calculated by propagating the uncertainties from droplet





volume and weighing of the cellulose and water. The instrument has a freezing background, likely caused by minor impurities in the Milli-Q water or on the hydrophobic slide. A background subtraction is performed to account for any freezing caused by this background (O'Sullivan et al. 2015) however the freezing reported here occurred at sufficiently warm

temperatures such that they did not overlap with the background freezing. For investigated temperatures listed in **Table 2**, Leeds-NIPI show $n_{s,geo,FC} \approx n_{s,geo,MCC}$, but the $n_{s,geo,FC}$ values are on average a factor of two higher than $n_{s,geo,MCC}$ across the investigated $T$ range (**Figs. 6l and 7f**). The $\Delta\log(n_{s,geo})/\Delta T$ values for MCC and FC are 0.47 and 0.57, respectively.

### 4.3.16. M-AL

For investigating the immersion freezing of droplets containing cellulose particles we have utilized two independent contact-free drop levitation methods in our laboratory at the Johannes Gutenberg University of Mainz, Germany. One of them is the Mainz Acoustic Levitator (M-AL) which was placed inside a walk-in cold room where the ambient temperature was set to be -30 °C. After introducing single drops into M-AL the drops were cooling down (at

a continuously varying cooling rate) adapting their surface temperature to the ambient temperature. The size of the levitated drops was approx. 2 mm which was determined for each drop from the images captured by a digital video camera attached to the M-AL. Such large droplet size enabled the direct measurement of the surface temperature during the experiments with means of an infrared thermometer, therefore reducing the error in

temperature originating from indirect determination of droplet temperature. The onset of freezing was characterized by a sudden increase in the surface temperature caused by the latent heat released during nucleation. The freezing temperatures of 100 drops was measured for each cellulose samples (MCC, FC and NCC) at two distinct concentrations, 1.0 and 0.1 wt%. Due to the relatively large droplet size a wide range of temperatures was covered (-13 to -23

°C) yielding $n_{s,geo}(T)$ values ranging from $10^4$ to $10^7$ m$^{-2}$. The NCC sample we got for investigation was contaminated by mold therefore the $n_{s,geo}(T)$ deviates significantly from other techniques at temperature above -20 °C (see **Fig. 4c. iii**). For investigated temperatures listed in **Table 2**, M-AL shows $n_{s,geo,MCC} > n_{s,geo,FC}$ and $n_{s,geo,MCC} > n_{s,geo,NCC}$ (**Figs. 6n, 7h and 8g**). For example, at -17 °C, the $n_{s,geo}(T)$ values of MCC, FC and NCC are 2.54 x $10^5$, 2.48 x $10^5$ and 8.28 x $10^4$ m$^{-2}$. The

$\Delta\log(n_{s,geo})/\Delta T$ values vary for 0.28 (FC)-0.40 (MCC) with the spectral parameter of NCC (0.31) falling around the middle.

### 4.3.17. M-WT

The main facility of our laboratory at the JGU Mainz is a vertical wind tunnel (M-WT) in which atmospheric hydrometeors can be freely suspended in the updraft of the tunnel at





temperatures down to -30 °C. Since all hydrometeors (from cloud droplets of few tens of µm to large hailstones with sizes of several centimeters) can be freely floated at their terminal falling velocities the relevant physical quantities, as for instance the Reynolds number and the ventilation coefficient, are equal to those in the real atmosphere.

The immersion freezing measurements in the M-WT have been conducted under isothermal conditions. The air was cooled down to a certain temperature between – 20 and -25 °C and at that temperature the frozen fraction of water droplets containing MCC or FC was measured by investigating typically 50 droplets a day. The drop temperatures were determined from the continuously recorded air temperature and humidity (*Diehl et al., 2014;*

*Pruppacher and Klett, 2010*). The size of the droplets was calculated from the vertical air speed which can be measured by high accuracy in the M-WT (*Diehl et al., 2014*). Due to the small droplet size and the applied INP concentration (0.1 wt%) a relatively narrow temperature range could be investigated yielding $n_{s,geo}(T)$ values ranging from $10^6$ to $10^8$ m$^{-2}$. Over -23 °C < $T$ < -22 °C, M-WT shows $n_{s,geo,MCC} > n_{s,geo,FC}$ (**Figs. 6o and 7i**). Corresponding $\Delta \log(n_{s,geo})/\Delta T$ values

are 0.26 for MCC and 0.48 for FC.

### 4.3.18. NC State-CS

Across investigated temperatures ($T \in [-23, -16]$ °C), results from the NC State CS show that INAS is indistinguishable between FC, MCC, and NCC for all temperatures within experimental uncertainty, except for $T > -18$ °C where $n_{s,geo,NCC}$ is less than that of FC and MCC. Overall, the

NCC spectrum is narrower than the FC and MCC spectra, suggesting that the distribution of active sites for NCC is slightly more homogenous. The data connect with the $n_{s,geo}^{H15MCC,wet}$ parameterization at $T = -22$ °C, but falls below by ~ 1 order of magnitude at $T = -23$ °C. The data intersect with the $n_{s,geo}^{H15NX,wet}$ parameterization in the $-20 < T < -18$ °C range. However, the $n_{s,geo}^{H15NX,wet}$ has a steeper slope with temperature and thus overpredicts and underpredicts

$n_{s,geo,cellulose}$ at colder and warmer temperatures, respectively.

### 4.3.19. WISDOM

Over the investigated temperature range given in **Table 2**, WISDOM shows $n_{s,geo,MCC} > n_{s,geo,NCC}$ (**Figs. 6r and 7j**). The MCC result exhibits broader spectrum with $\Delta \log(n_{s,geo})/\Delta T$ of 0.26 than NCC ($\Delta \log(n_{s,geo})/\Delta T = 0.31$). The observed relation between widening of spectra and increased

$n_{s,geo}(T)$ suggests that giant particles have increased diversity in immersion freezing as compared to submicron particles. Looking at the overall NCC data (**Fig. 7.c.iii**), nearly all aqueous suspension techniques, independently of the drop volume, agree with the WISDOM data and all point towards the AIDA data. We remark that the WISDOM team followed the suggested sample handling details described in **Sect. 3.1**.



### 4.4. Surface Structure of Cellulose Samples

We will discuss possible explanations for the observed diversity of data from different techniques in detail below. A detailed discussion of the samples comparison (surface difference) is given in this sub-section. **Figure 9** shows a representative SEM image and a

processed image for MCC. As can be seen in **Fig. 9a**, our cellulose surface possesses substantial amount of line structures and defects that may provide thermodynamically preferential condition to suppress the energy barrier of crystallization and perhaps induce different interactions with water vapor and/or super-cooled water droplets (*Page and Sear*, 2006). Brighter regions of the line structures in **Fig. 9b** correspond to structural peaks whereas darker

parts represent troughs on the surface.

       **Figure 10** shows the surface density of these submicron structures on MCC as well as FC. Interestingly, the lengths of linear peaks are log-normally distributed on both MCC and FC particles with modes of ~0.6 and 0.7 µm, respectively. Moreover, the line structure length of FC particles is slightly larger but less abundant than those of MCC particles. At the mode size,

the structure density exceeds 0.4 µm$^{-2}$ (4 x 10$^{11}$ m$^{-2}$) for MCC and 0.3 µm$^{-2}$ (3 x 10$^{11}$ m$^{-2}$) for FC. Note that there is none for NCC. In addition, we also examined seven of >10 µm MCC particles and confirmed they had similar features as <10 µm particles (not shown).

       **Figure 11** shows TEM and SEM images of NCC particles at various magnifications. Unlike MCC and FC, there exist no notable surface defects on the NCC surface. As shown in

the TEM images, NCC seems to be composed of single fiber with 10s nm width and 500-800 nm length. At a given aqueous concentration (0.03 wt%), some NCC fibers aggregate each other, forming particulate aggregates of >1µm; however, there are less abundant agglomerations as compared to MCC and FC based on our SEM observations (**Fig. 11 e and f**).

       Together with our offline characterization of sample physico-chemical properties

(**Sects. 2.2**), we observed the presence of considerable amount of surface porosity and line structures on MCC and FC type particles. With a mode size of >0.6 µm, the surface density of these surface structures is estimated to be at least 3 x 10$^{11}$ m$^{-2}$. This density is almost equivalent to the observed maxima of $n_{s,geo,MCC}$ (**Table 8**), suggesting these structures may act as ice active sites and may be responsible for heterogeneous freezing, assuming the density of

these linear structures correlate with that of pores, acting as ice active sites. In contrast, there is no surface structure observed for submicron NCC as it mainly retains a single fibrous form. Most importantly, our observation suggests that submicron-sized pores that are uniquely abundant on MCC and FC may be, at least partially, responsible for the observed differences in ice nucleation efficiency amongst materials (i.e., $n_{s,\,MCC/FC} > n_{s,\,NCC}$) prescribed in **Sect. 4.2**. It

is, however, important to note that our method is limited to measure line structures of

approximately >0.25 μm. The structures of <0.25 μm are presumably considered as noise because of poor SEM resolution. Nonetheless, this limitation does not rule out the possibility of a capillary condensation effect (i.e., inverse Kelvin effect) of nano-sized pores on ice nucleation enhancement (*Marcolli*, 2014). Hence, further detailed investigation of the

influence of <0.25 μm ice nucleation active sites is necessary in the future.

### *4.5. Experimental Parameters*

This section addresses the relationship between experimental conditions/parameters and ice nucleation results to find a potential controlling factor of the observed measurement diversity in $T$ and $n_{s,geo}(T)$. Particularly, we discuss the influence of impurities within water towards

freezing (**Sect. 4.5.1**), quantifiable variables (**Sect. 4.5.2**) and nominal experimental parameters (**Sect. 4.5.3**) on our immersion freezing measurements.

### 4.5.1. Water Freezing Spectra

Heterogeneous nucleation experiments often suffer from unknown ice active contributors or foreign contaminants suspended in supercooled droplets, triggering non-homogeneous

freezing at supercooled temperatures ($T$ > -38 °C). Even with high purity water, it is difficult to eliminate the contribution of heterogeneous INPs in water, especially when using droplets on the microliter scale (*Whale et al.*, 2015 and references therein). To our knowledge, only a small number studies have reported their microliter water droplets to produce freezing spectra with negligible artifacts and reproduce freezing temperatures close to the homogeneous limit

predicted by CNT [*Tobo*, 2016; *Reicher et al.*, 2018; *Polen et al.*, 2018; *Peckhaus et al.*, 2016; *Fornea et al.*, 2009 – note the data is not shown in *Fornea et al.* (2019)]. To understand the contributions of the impurities within water towards freezing results, we further analyzed the immersion freezing results of various purity grade water used in aqueous suspension experiments.

25         **Figure 12** shows frozen fraction spectra of pure water with different grades and freezing temperatures of background INP per liter in the water. Various freezing temperatures seen in **Fig. 12a** suggest that freezing behavior of the water depends on the droplet size and several types of water purity grades. Clearly, the comparison of background freezing of different droplet volumes (1, 3 and 5 μL) evaluated by WT-CRAFT indicates that larger droplet

volume promotes early freezing at high temperatures. Thus, despite unknown source of such an early onset, the probability of undesired INP inclusion seems – as expected – to correlate with individual droplet size. As apparent in **Fig. 12b**, homogeneous nucleation can occur at higher temperatures than -38 °C (*Koop and Murray*, 2016). For instance, 10 μL droplets would



possess 50% activation at just below -33 °C with a cooling rate of 1 °C min$^{-1}$. The WISDOM measurements with 0.6 nL of DI water are consistent with homogeneous nucleation.

The observed heterogeneous freezing of the water may not solely reflect impurity in the water as it is inherently related to other system artifacts, such as variation in heat conduction and droplet $T$, contribution of a supporting substrate and dissolved foreign gases. It is also noteworthy that using autoclaved sterile water did not hinder the background droplet freezing on WT-CRAFT, implying negligible biological contribution to the observed water droplet freezing. In addition, it has been shown that the surface on which microliter droplets are supported also introduces background freezing sites, with ultra pure silicon or Teflon surfaces producing less background freezing than a hydrophobic glass surface (*Diehl et al*., 2001; *Price et al*., 2018). The characterization of water quality to identify what causes the observed dominant background freezing in deionized water is beyond the scope of our investigation. However, determining the best possible practice to make sure the freezing temperatures of pure water droplets <-30 °C or lower is important in aqueous suspension experiments (*Knopf et al*., 2018; *Price et al*., 2018; *Polen et al*., 2018). For example, using microfluidically generated sub-micro liter drops and proper substrate condition (e.g., where the droplets are completely surrounded by oil and not in contact with the substrate) may be the key (*Tarn et al*., 2018; *Polen et al*., 2018). Another key is to check the background freezing on a routine basis. Obtaining absolutely clean water is conceivably challenging. Perhaps, running a control experiment with commercially available HPLC water may provide complementary insight on the inter-system offset. *Polen et al*. (2018) recently evaluated a series of different substrates and water purification strategies to reduce background freezing interference in droplet freezing assays. They propose a series of recommendations regarding experimental methods and data analysis strategies to reduce and properly account for these background freezing interferences. Note that the shift in freezing temperatures in **Fig. 12c** may also in part derive from the deviation in INP detection methods or variation in heat conduction and droplet $T$. A systematic calibration of the temperature sensor (and associated freezing/melting point) would benefit increasing overall accuracy and precision of droplet assay techniques. It is also important to note that the apparent steep increase in INP concentrations for the WISDOM device at temperatures below about -34 °C (**Fig. 12c**) does not imply that the water droplets in these experiments contained numerous INPs. Instead, the observed sharp increase in freezing rates of these rather small (<100 µm) droplets, which might be particle-free, is most probably due to homogeneous ice nucleation. The observation agrees with previous studies of homogeneous ice nucleation in droplets of this size and published homogeneous ice nucleation rates (*Riechers et al*., 2013; *Ickes et al*., 2015).



### 4.5.2. Nominal Experimental Parameters

The discussion of the experimental parameters, which may be responsible for the observed diversity of ice nucleation data, is now provided. This section discusses two more issues which might contribute to the observed deviations. As seen in **Tables 5 and 6**, experimental procedures are diverse, potentially responsible for abovementioned deviations in quantifiable experimental parameters. For example, the ice detection methods deviate, highly depending on the size and number of supercooled droplets examined. Thus, the standardization of ice detection is important to minimize the measurement diversity. Correspondingly, the false/positive image analysis should be standardized not to miscount half frozen half unfrozen droplets (*Wright and Petters*, 2013). The 8bit mean gray value image analysis procedure introduced in *Budke and Koop* (2015) is ideal and recommended to the new cold stage users. Other emerging technologies (e.g., application of IR to detect the latent heat release and droplet freezing) may become available in the future (*Harrison et al*., 2018). On the other hand, *in situ* methods detecting droplets that were grown on single particles typically use OPCs for ice counting (except microscopy-combined individual freezing observation apparatus, such as EDB, FRIDGE and DFPC-ISAC). Detecting small ice crystals and separating them from droplets of the overlapping optical size range is a challenge *(Vochezer et al*., 2016). In LACIS, a change in depolarization is used to discriminate between frozen and liquid droplets (*Clauss et al*., 2013). A depolarization technique has been implemented in other ice nucleation methods (*Nicolet et al*., 2010; *Garimella et al*., 2016). A new technology of optical scattering methods (e.g., *Glen et al*., 2013; 2014) was recently introduced to improve the small ice detection capability.



### 5. Conclusion and Future Outlook

This paper presents the immersion freezing efficiencies of giant and submicron cellulose particles of three different types evaluated by a total of twenty IN instruments at supercooled temperature conditions. Three cellulose samples examined in this study showed a propensity

to nucleate ice, and their ice nucleation activity are comparable to another test system (i.e., illite NX) that we have previously evaluated. On average, giant cellulose samples are more ice active than the nano cellulose one at $T$ lower than -20 °C although the difference is not apparent for all temperatures when considering experimental uncertainty. Electron microscopy revealed that giant cellulose particles possess surface features such as fibrous

structures that may act as the ice nucleation active site and influence the immersion freezing efficiency. This surface feature was unique for MCC and FC samples, but was not observed for the cellulose samples (NCC).

      Our work also provides a comprehensive dataset of experimental variables in INP measurement techniques to complement our insufficient knowledge regarding inter-method

diversity that, when filled, will enhance the credibility of our experiments to evaluate INP abundance in the atmosphere. Strikingly, our results indicate that the overall diversity derived from comparing techniques is significant when compared to the individual uncertainties of each instrument.

      The observed diversity amongst measurement techniques for cellulose is larger than

that observed for a mineralogically heterogeneous illite NX sample described in our previous inter-comparison study (H15b). For illite NX, the deviations in $T$ are within 8 °C (H15b) while they span 10 °C for cellulose. For $n_{s,geo}(T)$, while the span in results covers a maximum of three orders of magnitude for illite NX, they span 4 orders of magnitude for cellulose. These diversities suggest the complex surface structure and compositional heterogeneity may play a

substantial role to explain the diversity. This also implies that the cellulose system might not be suitable as a calibrant.

      Observed deviations could arise from a number of sources. As verified in this manuscript, there are many experimental variables involved in currently available INP measurement techniques, and such a diverse variation seems to yield significant data diversity

and limit the instrument validation by distributing any reference bulk materials. To at least qualitatively examine what experimental parameters predominantly generate the $n_{s,geo}(T)$ diversity, the MCC results of a selected number of measurements derived under similar experimental condition were systematically compared. Our results show that two distinct modes of more and less active ice nucleation were found at higher temperatures for dry



dispersion and aqueous suspension results, respectively. To further validate the INP measurement instruments using reference INPs in the future, we suggest the following six points:

**1)** **Working with similarly produced samples:** As described in **Sect. 4.3.7**, our cellulose powders (especially MCC) promptly settle in water. Sampling a filter of size segregated cellulose generated by means of dry dispersion from a large volume chamber after letting giant MCC settle out and running it on a droplet freezing assay (e.g., **Sects. 4.3.2 and 4.3.3** DFPC-ISAC and FRIDGE) is important to assure working with the same

sample. Otherwise, aerosolising and then doing the ice nucleation experiment versus suspending particles in water might result in different particle populations. Knowing the sample volume of air, $V_s$, and liquid suspension volume, $V_w$, we can estimate immersion freezing efficiency of the sample particles in terms of INP concentration per volume of air [$n_{INP} = c_{INP}(T)\left(\frac{V_w}{V_s}\right)$]., which may be a better ice nucleation

parameter for the instrument comparison.

   **2)** **Sample stability analysis:** Chemical and structural changes during sample processing (e.g., *Lützenkirchen et al.*, 2014) should certainly be considered more carefully. Depending on the aerosolization method, the surface properties can be altered even for the same sample (see **Sect. 2.2**). For instance, the changes in particle size,

morphology and hygroscopicity can occur for atomized particles from a suspension of the powder in water, compared to the dry powder (*Koehler et al.*, 2009; *Sullivan et al.*, 2010). Understanding the effect of alteration in particulate properties on IN (e.g., *Polen et al.*, 2016) must be studied in the future.

   **3)** **Interfacial effect characterization:** Since the cellulose is a strong desiccant and

absorbs a lot of water from the droplet, pre-exposure to humidified condition may create partially immersed solid-liquid interfacial condition. An effect is viable. For instance, giant particles (MCC and FC) partially immersed but half exposed to air may create the interfacial condition preferable for ice formation. This quasi-contact (perhaps also condensation) freezing process may be analogous to the dry dispersion

techniques (with different induction time). The future study to visually inspect this mechanism by means of microscopy (*Kiselev et al.*, 2017) and verify it as an atmospherically representative process is an imperative task.

   **4)** **Method Standardization:** Standardization of our methods (e.g., ice detection and in particular INP sampling and treatment) may be one route to reduce the prevailing



measurement diversity. Evidently, we verified that the aqueous measurements with smaller droplets and less aerosol exerted high $n_{s,geo}(T)$ of cellulose samples (**Sect. 4.3.14**). A similar observation is addressed in *Beydoun et al*. (2016). As atmospheric cloud droplets range over sizes up to some tens of micrometres (*Miles et al*., 2000), using an atmospherically relevant range of water volume or at least tenth of micro-liter scale may be a key to improve our measurement comparability in the future. Such effort may reduce the diversity in experimental conditions and unify the experimental parameters (e.g., $\Delta\log(n_{s,geo})/\Delta T$). Currently, given parameters are treated as if free variables, certainly contributing to the data diversity. A community-wide effort to quantify nominal characteristics of each technique (e.g., background correction and sample pre-treatment) is another key to achieve more precise and accurate INP measurements (*Polen et al*., 2018). For future works, aqueous suspension measurements aligned with the protocol are desired. This might warrant the particle size distribution of the steady-state suspension, perhaps similar to what is examined in the cloud simulation chamber experiments. Alternative strategy is to rigorously examine the causes and clearly define the limitations of individual techniques. Nonetheless, we believe a current diversity in techniques is beneficial at least at this point, in particular because they allow different types of approaches for identifying new INPs.

5) **Active site validation:** One of the biggest uncertainties in the $n_{s,geo}(T)$ concept is the interpretation of particle surface area (H15b). More rigorous understanding of the true surface area of the system by parameterising *SSA* as a function of particle concentration in a drop is a crucial step to constrain the $n_{s,geo}(T)$ concept as this parameter obviously varies amongst experiments as presented in this work (**Sect. 2.1**). Given the size-dependence of $n_{s,geo}(T)$ for MCC discussed in **Sect. 4.3.4**, varying concentration to access a wider freezing temperature range and stitching the $n_{s,geo}(T)$ spectra obtained from different concentrations together may be problematic (*Beydoun et al*., 2016). This approach may create an issue especially towards high *T*, where highly concentrated suspension droplets are typically utilized to diagnose their freezing ability. High particle concentrations also promote particle aggregation and gravitational settling out of the droplet (*Beydoun et al*., 2016; *Emersic et al*., 2015).

In conclusion, we have shown that several types of cellulose have the capacity to nucleate ice as efficiently as some mineral dust samples. Given cellulose within plant residue is present in the atmosphere, it represents a poorly characterised non-proteinaceous INP type. While the



diverse instruments employed in this study agree in that cellulose has the capacity to nucleate ice, their quantitative agreement is poor. Unfortunately, it is not possible as yet to say what the cause of this disagreement is. We suggest a number of topics that future studies could address in order to better understand and resolve this discrepancy. Nevertheless, we show

5    that cellulose has the potential to be an important atmospheric ice nucleating particle and more work is warranted.





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

### Acknowledgements

The INUIT (Ice Nuclei research UnIT) is a German Research Foundation (DFG) funded multi-institutional project (FOR1525, www.ice-nuclei.de). The objective of INUIT is to comprehensively study the atmospheric heterogeneous ice formation with collaboration among various research institutions. The entire INUIT members and associated partners acknowledge Birte Hülsen und Susanne Staechelin for central coordination & administration

of the INUIT research unit. We thank  Dr. Shaul Lapidot, Dr. Clarite Azerraf and Melodea Ltd. for providing the NCC sample for this research.

         The AIDA team (OM and NH) and the INKA team (KH and TS) acknowledges the IMK-AAF engineering and infrastructure group (Georg Scheurig, Rainer Buschbacher, Tomasz Chudy, Olga Dombrowski, Jens Nadolny, Frank Schwarz and Steffen Vogt) for their continued

support throughout INUIT-1 and -2. The CSU-CFDC team (KJS, GPS and PJD) acknowledges support from U.S. National Science Foundation (NSF) grant AGS1358495. GPS additionally acknowledges support from NSF postdoctoral grant AGS1433517. The CNR-ISAC team acknowledges support funding for their research from the European Union's Seventh Framework Programme FP7-ENV-2013 project BACCHUS (grant no. 603445) and F. Corticelli

(Institute for Microelectronics and Microsystems IMM-CNR) for SEM observations. The FRIDGE group acknowledges funding by DFG in the Research Unit FOR 1525 (INUIT) under BI 462/3-2. For the LACIS work, HW was funded within the INUIT subproject WE 4722/1-2. The MRI cloud simulation chamber experiments were partly supported by JSPS KAKENHI Grant Numbers 23244095 and 17H00787. Support of the PNNL-CIC work for Gourihar Kulkarni was provided

by the US Department of Energy (DOE) Office of Biological and Environmental Research (OBER) Atmospheric Research Systems Program (ASR). The Pacific Northwest National Laboratory is operated for DOE by Battelle Memorial Institute under contract DE-AC05-76RLO 1830. In addition, OM, NH and GK acknowledge the EMSL general use grant (proposal ID 49077) for supporting the PNNL-CIC work. The BINARY group (CB, EJ and TK) acknowledges funding from

DFG under the INUIT subproject KO 2944/2-2, and also acknowledge S. Robrecht, who designed and performed experiments that contributed to the development of the recommended suspension preparation protocol. The CMU-CS team (MP, HB, and RCS) was supported by the National Science Foundation (grant number CHE-1554941, and a Graduate





Research Fellowship for MP). The NIPR-CRAFT experiments were partly supported by JSPS KAKENHI Grant Numbers 15K13570, 16H06020. KC and NH thank the support of NSF-EAPSI for the validation of NCC as well as microscopy characterization of our cellulose samples conducted in collaboration with YT and KA in Japan. The Leeds group thank the European

Research Council (ERC, 240449 ICE; 632272 IceControl; 648661; MarineIce; 713664 CryoProtect) and the Natural Environment Research Council (NERC, NE/I019057/1). The part of the research based on LINDA measurements was supported by the Swiss National Science Foundation (SNF) through grant nos. 200021_140228 and 200020_159194. For M-AL and M-WT, MS and AM acknowledge support from Deutsche Forschungsgemeinschaft (DFG) under

contract SZ260/4-2 within Research Unit INUIT (FOR 1525). The NC State group (HT and MDP) acknowledges funding from NSF-AGS 1450690. The NC State group thanks Danielle Dillane, Mary Hester, Chris Rohrbach, Margaret Scott, Hannah Tekleab, and Mark Wu for their help with collecting the raw data. The WISDOM team (YR and NR) gratefully acknowledge support from the Ice Nuclei Research Unit (INUIT) of the German DFG, The Helen Kimmel Center for

Planetary Sciences and the DeBotton center for Marine Sciences. NH and CW thank the HEAF and Killgore research grant for the development and implementation of WT-CRAFT. This material is based upon work supported by the U.S. Department of Energy, Office of Science, Office of Biological and Environmental Research (DE-SC0018979).

       For offline mass-spectrometry, JS and HC acknowledge funding from DFG under the

INUIT subproject SCHN 1138/2-2. Additional support (AZ, DMB, KS) was provided by the U.S. Department of Energy (DOE) Office of Biological and Environmental Research (OBER) Atmospheric Research Systems Program (ASR). The part of the research was performed using EMSL, a DOE Office of Science User Facility sponsored by the Office of Biological and Environmental Research and located at Pacific Northwest National Laboratory.

25       For offline microscopy, KA thanks the support of the Global Environment Research Fund of the Japanese Ministry of the Environment (2-1703) and JSPS KAKENHI (grant numbers JP16K16188, JP15H02811 and JP16H01772). NH thank Soeren Zorn and Konrad Kandler for a useful discussion regarding the potential artifact analyses at the initial stage.

       The article processing charges for this open-access publication have been covered by

a Research Centre of the Helmholtz Association.

**Author contributions**

J. Curtius and O. Möhler proposed the framework of this collaborative multi-institutional laboratory work. The overall manuscript, coordinated and led by N. Hiranuma, was a collaborative effort of the partners of the INUIT group and associated partners. N. Hiranuma



and O. Möhler conceived the AIDA experiments, analyzed and discussed the results and contributed to the AIDA text. K. Suski and G Schill performed CFDC experiments and K. Suski, G. Schill and P. DeMott analyzed CFDC experimental data. M. Piazza performed the DFPC experiments and analysis with the support of A. Nicosia. He also contributed to the data
elaboration. F. Belosi and G. Santachiara contributed to the design of the cellulose aerosol generation systems and data elaboration. F. Belosi oversaw the DFPC ice nucleation measurements. D. Weber performed the FRIDGE experiments, and contributed to the associated data analysis with a support of H. Bingemer. T. Schiebel and K. Höhler performed and analyzed INKA experiments. S. Grawe performed the LACIS measurements and data
evaluation which were coordinated and overlooked by H. Wex. K. Yamashita, T. Tajiri, A. Saito and M. Murakami performed MRI cloud simulation chamber experiments and contributed to the associated data analysis in collaboration with O. Möhler and N. Hiranuma. G. Kulkarni carried out the PNNL-CIC experiments and data analysis. E. Jantsch and T. Koop designed the BINARY experiments, which were performed by E. Jantsch. C. Budke, E. Jantsch and T. Koop
analyzed and discussed the data, and E. Jantsch and T. Koop contributed the associated text. M. Polen, H. Beydoun and R. Sullivan performed the CMU-CS experiments and their analysis. Y. Tobo and K. Cory performed the NIPR-CRAFT experiments. T. Whale performed µl-NIPI experiments. T. Whale and B. Murray analysed and discussed the data and contributed the associated text. E. Stopelli conduced the measurements with LINDA with a support of F. Conen.
M. Szakáll, O. Eppers and A. Mayer performed the M-AL and M-WT measurements and evaluated and discussed the experimental results, and contributed to the associated text. H. Taylor and M. Petters coordinated and carried out the NC State CS experiments. M. Taylor helped with initial data processing. M. Petters performed the final data processing and contributed the associated text. N. Reicher, Y. Rudich and B. Bhadori performed and analyzed
WISDOM experiments, L. Segev designed freezing detection program and helped with the data analysis. C. Whiteside and K. Cory designed performed the WT-CRAFT experiments, and contributed to the associated data analysis and text under the guidance of N. Hiranuma. J. Schneider and H.-C. Clemen conducted the ALABAMA laboratory and field measurements. A. Zelenyuk, K. Suski and D. Bell performed the miniSPLAT and multi-dimensional
characterization experiments, and accompanying data analysis. For offline microscopy, K. Adachi performed electron microscopy and image analyses. All authors discussed the results and contributed to the final version of the manuscript.

**Competing interests**

The authors declare no competing financial interests.



**Data Records**

Within the framework of INUIT, we established a new community database including the laboratory results on ice nucleation with access for registered users. The tabulated data are available in a publically accessible MySQL portal at http://imk-aaf-s1.imk-aaf.kit.edu/inuit/.

This database helps the users to evaluate and interpret the comprehensive laboratory ice nucleation results measured over the past years. It also provides a good basis for a database with a wider public access.

**Usage Notes**

All data associated with this study will be made available without any barriers to the user. Any

disputes about the use of other groups' data, particularly with respect to publications, will be resolved by the INUIT coordinators.

**Figures**

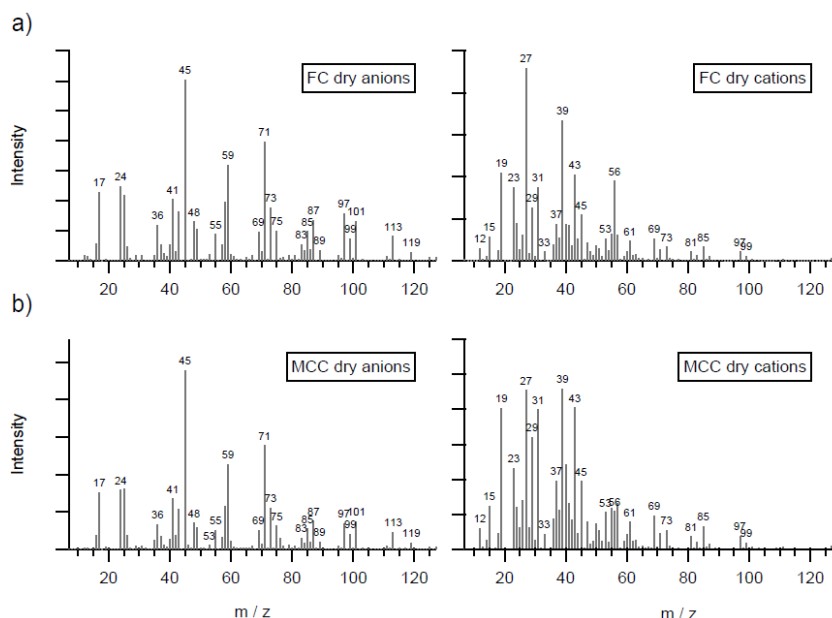

**Figure 1**. Laboratory reference mass spectra of dry dispersed cellulose particles with

15 ALABAMA. a) Fibrous cellulose (FC), b) Microcrystalline cellulose (MCC), left: anions, right: cations. These mass spectra represent between 60 and 75% of the particles (FC: 1585 out of 2071; MCC: 193 out of 329). The remaining particles show either higher molecular fragmentation and are therefore useful to identify molecular structures or show signs of contamination.



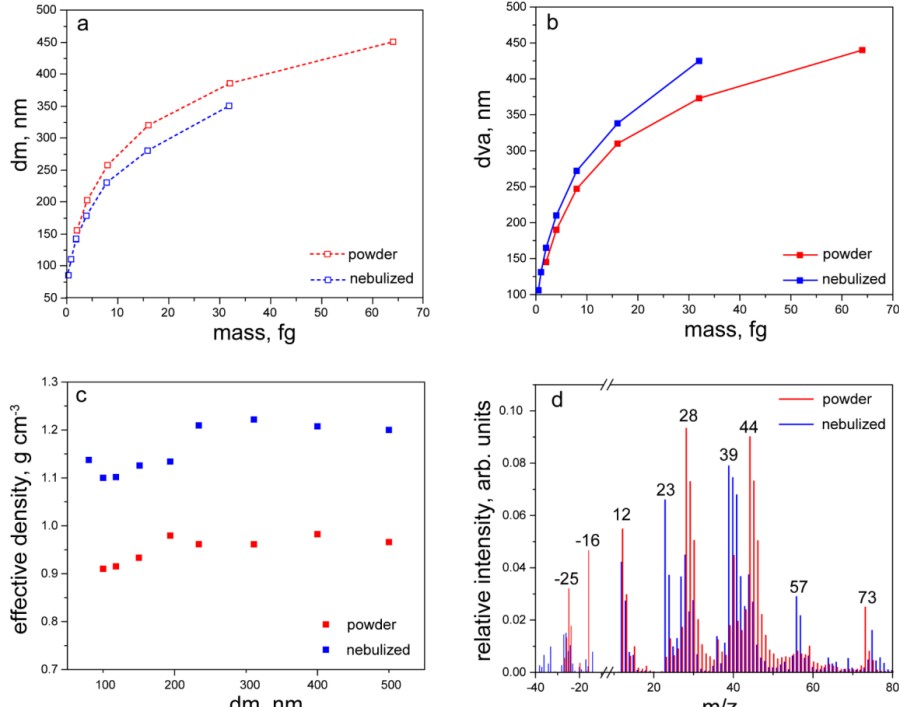

**Figure 2.** Aerosol particles mobility diameter ($d_m$) (a), vacuum aerodynamic diameter ($d_{va}$) (b), effective density (c) and mass spectra (d) of dry powder (red) and nebulized (blue) MCC particles.




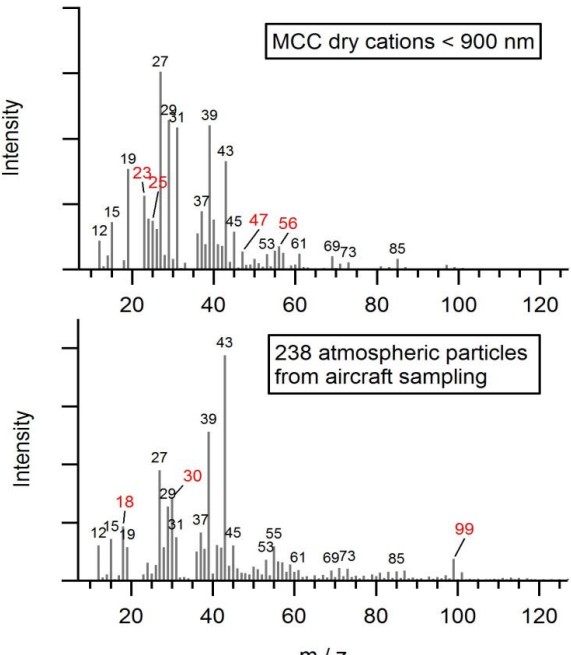

**Figure 3**. Comparison of atmospheric particles with laboratory cellulose measured by ALABAMA. Upper panel: Averaged mass spectrum of 22 MCC cation spectra of particles smaller than 900 nm ($d_{va}$). Lower panel: Averaged mass spectrum of 238 atmospheric cation mass spectra selected using the marker peaks. Between 0.5 and 1.0% of the atmospheric particle fulfilled the marker peak criteria. The overall correlation coefficient ($r^2$) of the two spectra shown here is 0.58. Ions that significantly different between the display mass spectra are labelled in red.





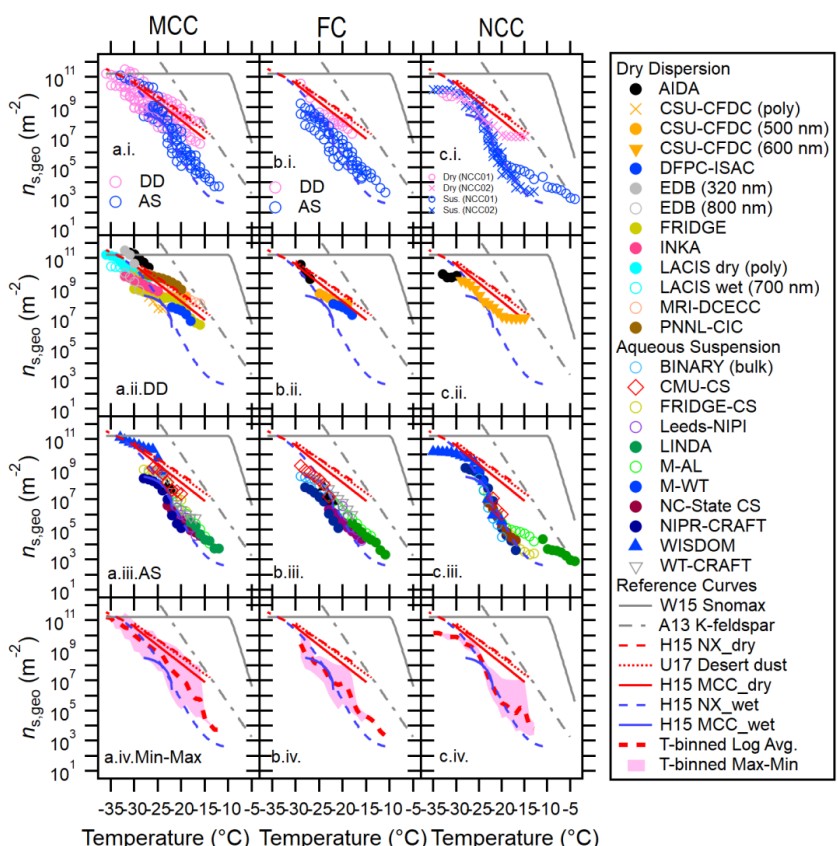

**Figure 4**. Immersion freezing $n_{s,geo}(T)$ spectra for MCC (a), FC (b) and NCC (c) from different techniques. Dry dispersion results (DD, pink markers) and aqueous suspension results (AS, blue markers) are shown in (i) to highlight the difference between these two subsets. Inter-comparisons of DD and AS for each cellulose sample type using $T$-binned $n_{s,geo}$ are presented in (ii) and (iii), respectively. The log average of all results as well as the deviation between maxima and minima of $n_{s,geo}(T)$ are shown in (iv). Reference immersion freezing $n_s(T)$ spectra for MCC (H15a) illite NX (H15b), Snomax (*Wex et al.*, 2015), desert dusts (U17; *Ullrich et al.*, 2017) and K-feldspar (A13; *Atkinson et al.*, 2013) are also shown (See **Sect. 4.1**). For NCC, the results from two different batches (NCC01 from Dec 2014 and NCC02 from May 2015) are shown.





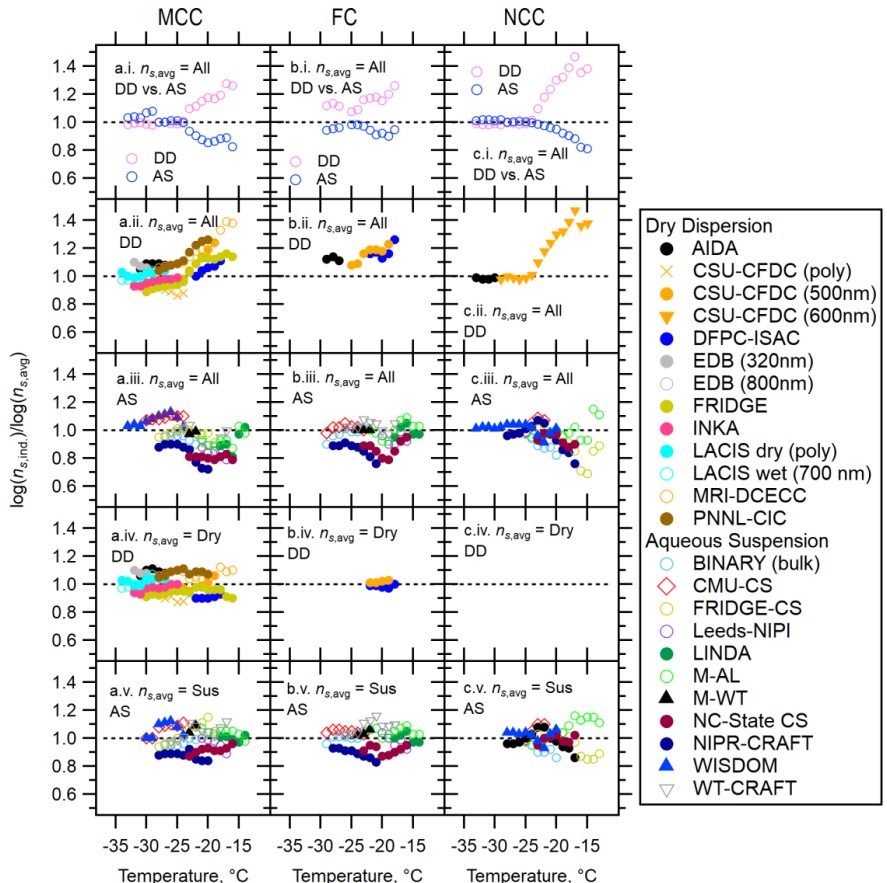

**Figure 5**. *T*-binned ratios of the interpolated individual measurements to the average of the data, $\log(n_{s,ind.})/\log(n_{s,avg})$, based on the geometric surface area ($n_{s,geo}$) for MCC (a), FC (b) and NCC (c). *T*-binned $\log(n_{s,ind.})/\log(n_{s,avg})$ are presented for (i) ratios of the log average to dry dispersion measurements (DD) or aqueous suspension measurements (AS) to the log average to all the data (All), (ii) ratios of the individual DD measurements to All, (iii) ratios of the individual AS measurements to All, (iv) ratios of the individual particle dispersion measurements to DD and (v) ratios of the individual aqueous suspension measurements to AS. The black dotted line represents $\log(n_{s,ind.})/\log(n_{s,avg}) = 1$. Panel c.iv is left blank since only one dataset is available at each temperature; thereby, no differences can arise.





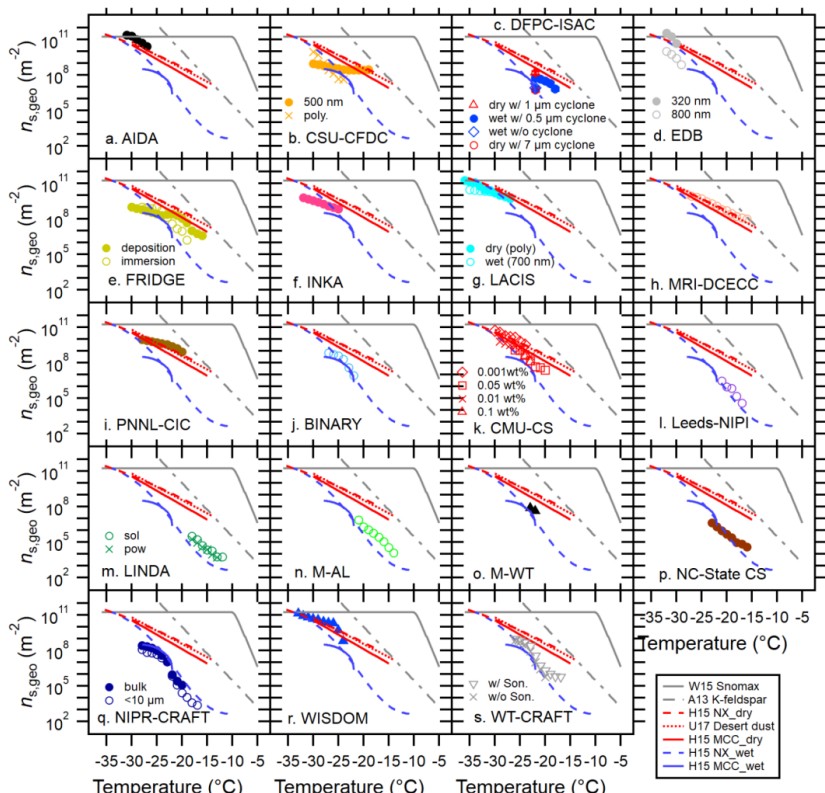

**Figure 6**. Inter-comparison of 20 INP measurement methods for MCC using $T$-binned $n_{s,geo}$. FRIDGE results of default (solid square) and imm.mode (open diamond) measurements are both presented in (e). Reference immersion freezing $n_s(T)$ spectra for MCC (H15a) illite NX (H15b), Snomax (*Wex et al.*, 2015), desert dusts (U17; *Ullrich et al.*, 2017) and K-feldspar (A13; *Atkinson et al.*, 2013), ATD and are also shown (See **Sect. 3.2**).





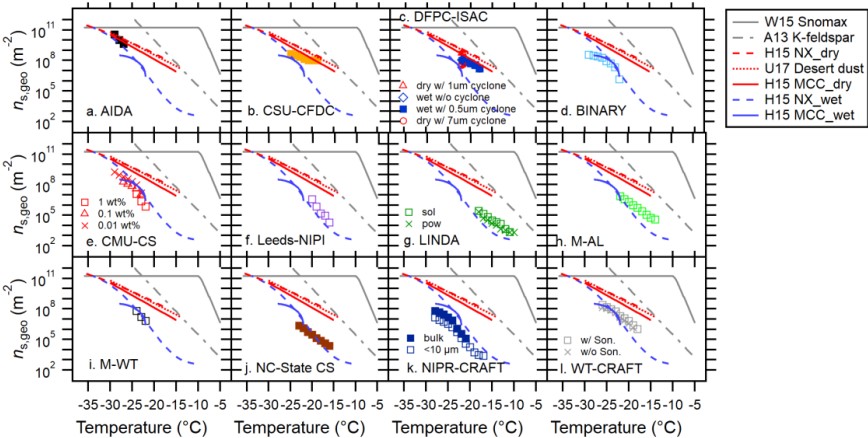

**Figure 7**. Inter-comparison of 12 INP measurement methods for FC using $T$-binned $n_{s,geo}$. Reference immersion freezing $n_s(T)$ spectra are provided as in **Fig. 6**.

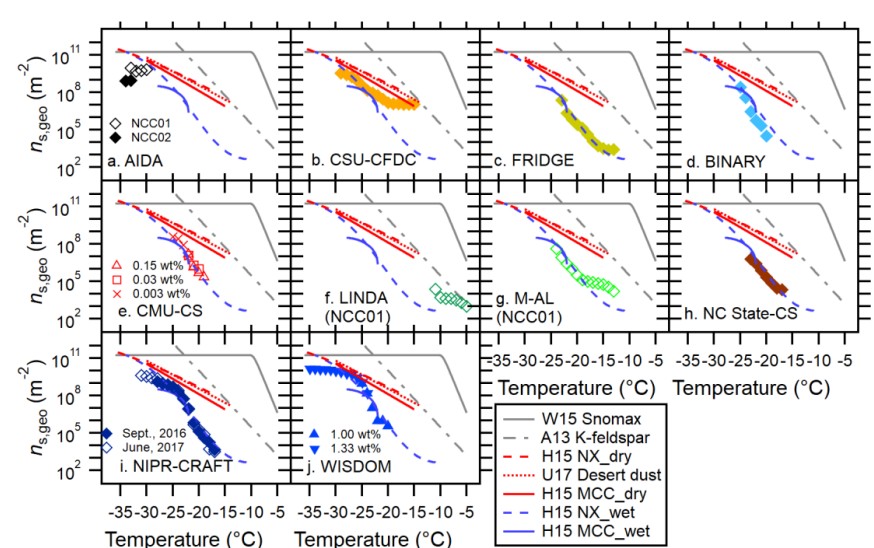

**Figure 8**. Inter-comparison of 11 INP measurement methods for NCC using $T$-binned $n_{s,geo}$. Reference immersion freezing $n_s(T)$ spectra are provided as in **Fig. 6**. Note: unless otherwise specified, the data are for NCC02.



a) SEM image                    b) Line image

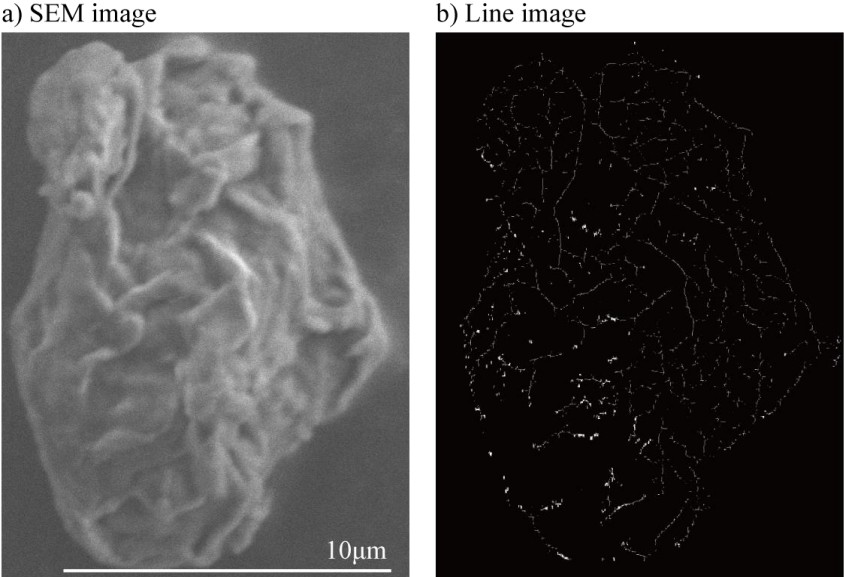

**Figure 9**. An example of surface image analysis. SEM image of a MCC particle (a) and its extracted surface line structure image analysed using an Interactive Data Language (IDL) program (b).

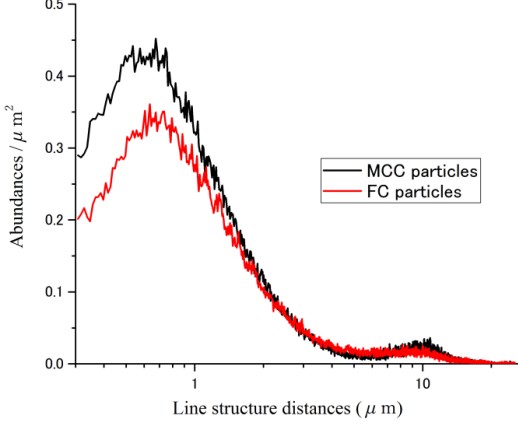

**Figure 10**. Surface abundance of line structures scaled to the particle surface area as a function of line structure length for MCC and FC particles. Peaks with smaller than 0.2 μm include noise and are excluded.



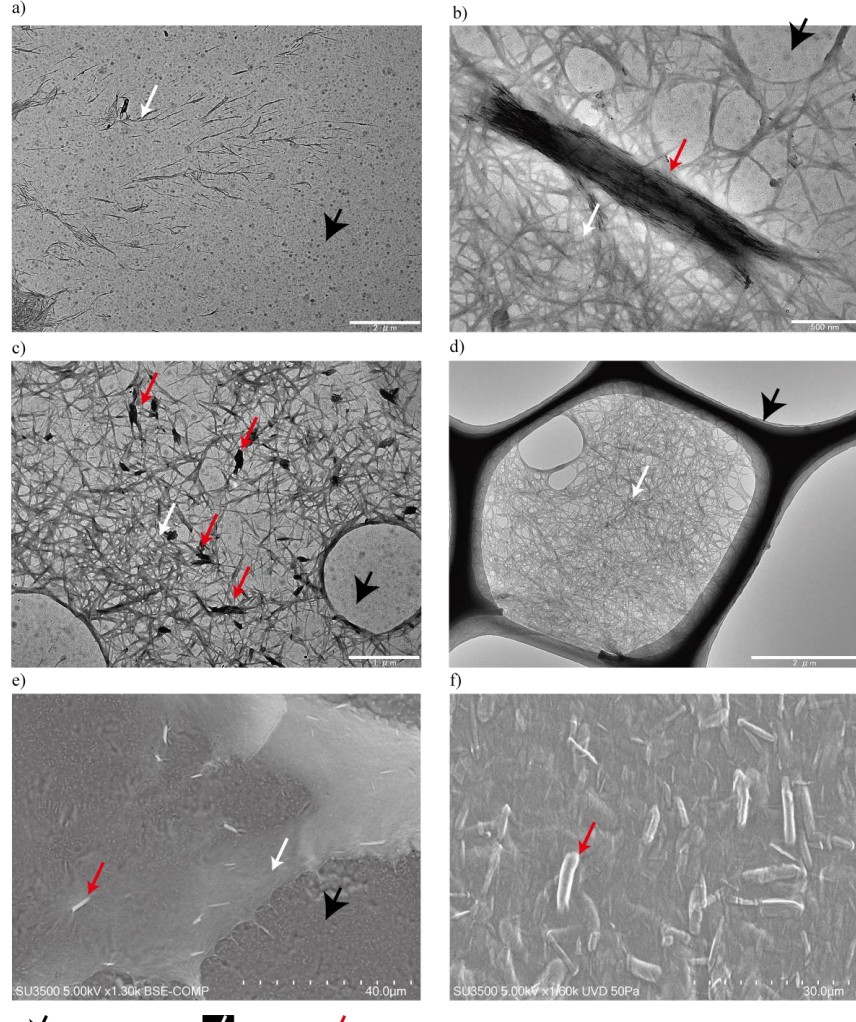

➤ Carbon substrate  ⬛ NCC fibers  ↗ Particulate aggregates of NCC fibers

**Figure 11**. TEM and SEM images of NCC samples. individual NCC fibers over a formvar carbon substrate (a). They form networks (white arrows) with some particulate aggregates (red arrows) (b and c). A stack of NCC fiber (white arrow) within a hole of lacey carbon substrate (black arrow) (d). SEM images of a layer with particulate NCC (red arrows) (e and f).



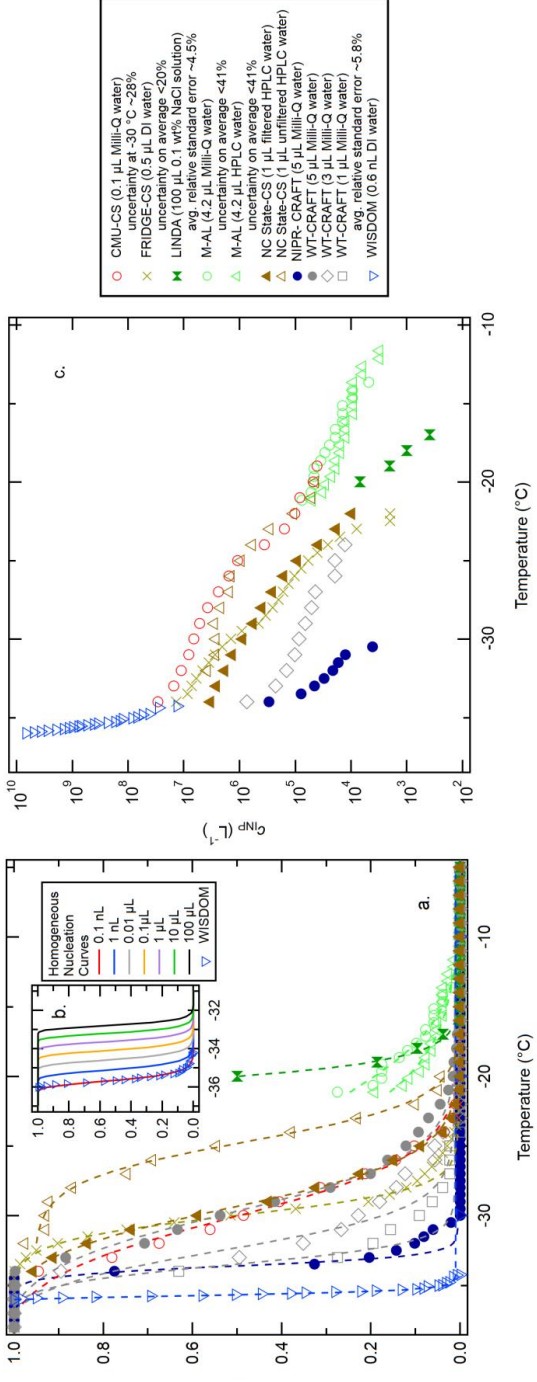

**Figure 12.** Freezing spectra of different water used for aqueous suspension methods. Three spectra with respect to measured frozen fraction, *FF* (a), homogeneous nucleation *FF* curves with a cooling rate of 1 °C min⁻¹ based on *Koop and Murray* (2016) (b), and INP per unit volume of water, $c_{INP}$ (c), are shown as a function of temperature. Sigmoidal fits are also shown in (a). Relevant experimental uncertainties are also included in the figure legends. Conversion from *FF* to $c_{INP}$ (vice versa) is prescribed in **Eqns. 3-4**. Note that the cumulative frozen fraction (b) is estimated using the nucleation rate derived from a polynomial fit to the CNT based parameterization in Fig. 4 of *Koop and Murray* (2016).



**Tables**

**Table 1.** Properties of micro-crystalline cellulose (MCC), fibrous cellulose (FC) and nano crystalline cellulose (NCC).

| System | MCC (Aldrich, 435236) | FC (Sigma, C6288) | NCC (Melodea, WS1)* |
|---|---|---|---|
| Chemical Formula | $(C_6H_{10}O_5)_n$ | $(C_6H_{10}O_5)_n$ | $(C_6H_9O_5)_n\ (SO_3Na)_x$ |
| Product Form | Powder | Powder | 3wt% thixotropic gel (viscosity ~4,665 ± 200 cP at 25 °C) in deionized water |
| [1]Density, g m⁻³ | ~1.5 | ~1.5 | ~1.0-1.1 |
| [2]Geometric Mode Diameter (± standard deviation) of dispersed particles, μm | 1.22 ± <0.1[3, 4] | 1.13 ± <0.1[5, 4] | 0.21 ± <0.1[6, 7] |
| SEM-based Mode Diameter of bulk materials (± standard deviation), μm | 54.24 ± 6.2 | >65 | 2.68 ± 0.3[8] |
| Manufacturer-reported Diameter | 51 μm | N/A | 5-20 nm width, 100-500 nm length |
| Aspect Ratio | 1.80-2.30 (4976/3)[9] | ~2.03 (371/1) | 2.30-2.93 (764/2) |
| [10]Geometric *SSA*, m² g⁻¹ | 3.35 ± 0.1 | 3.35 ± 0.5 | 18.59 ± 2.5 |
| [11]SEM-based *SSA* of residuals in 0.03wt% of 5 μL droplet, m² g⁻¹ | 0.068 | 0.087 | 1.24 |
| [12]BET-based *SSA*, m² g⁻¹ | 1.44 ± 0.10 | 1.31 ± 0.10 | 8.00 ± 1.00 |
| Crystallinity | ~80% (Cellulose Iβ crystallographic structure)[13] | N/A | 87% (Cellulose Iβ crystallographic structures)[14] |

*Two NCC samples from different batches, namely non-sterile NCC (NCC01) and freshly generated NCC (NCC02), were used for the IN characterization.
[1]Bulk density values according to manufacturers
[2]Based on $\Delta S/\Delta \log D_{ve}$ from ADIA measurements
[3]Measured by a combination of SMPS and APS at AIDA (INUIT06_1, 17, 31, 42, 43, 44, 45, 46, 54)
[4]Dry particles were dispersed into the AIDA chamber using a rotating brush generator (RBG1000, PALAS).
[5]Measured by a combination of SMPS and APS at AIDA (INUIT06_6, 14)
[6]Measured by a combination of SMPS and APS at AIDA (INUIT08_6, 7, 9, 10)
[7]Water-suspended NCC was aerosolized using the customized-atomizer (*Wex et al.*, 2015).
[8]The SEM-based mode diameter of atomized NCC is 0.28 ± <0.1 μm, which is similar to that of bulk NCC.
[9]Average aspect ratio per substrate: the numbers in bracket represent a total number of particles/substrate(s) analyzed under SEM for each subset.
[10]Geometric *SSA* is derived from ADIA measurements (i.e., fraction of total surface area concentration to total mass concentration estimated from a combination of SMPS and APS; See **Fig. S1**). The particles in AIDA were all <10 μm in diameter.
[11]Measured using droplet residuals derived from 5 μL of 0.03wt% suspension. Uncertainty is not given because all individual particle counts were compiled to calculate the *SSA* value of each sample.
[12]*Brunauer et al.*, 1938
[13]*Nishiyama et al.*, 2002
[14]*Aulin et al.*, 2009




**Table 2.** Summary of INUIT measurement techniques and instruments. Their acronyms are available in the Supplementary Information. ID 1-9 and ID 10-20 represent dry dispersion measurements and suspension techniques, respectively (Alphabetical order).

| ID | Instrument | Description | Mobile? | Reference | Cellulose type MCC | FC | NCC | Investigable T-range | Investigated T-range for this study | SSA (m² g⁻¹)** |
|---|---|---|---|---|---|---|---|---|---|---|
| 1 | *AIDA | CECC | No | Möhler et al., 2003; Niemand et al., 2012 | x | x | x | -100 °C < T < -5 °C | MCC: -31 °C < T < -27 °C / FC: -29 °C < T < -27 °C / NCC: -33 °C < T < -30 °C | MCC (poly): 3.35 / FC (poly): 3.35 / NCC (poly): 18.59 |
| 2 | CSU-CFDC | Cylindrical-walled CFDC | Yes | DeMott et al., 2015 | x | x | x | -34 < T < -9 °C | MCC: -30 °C < T < -23 °C / FC: -25 °C < T < -19 °C / NCC: -29 °C < T < -25 °C | MCC (poly): 2.09 / MCC/FC (500 nm): 8.00 / NCC (600 nm): 6.67 |
| 3 | DFPC-ISAC | Substrate-supported diffusion cell | No | Santachiara et al., 2010; Belosi et al., 2014 | x | x | | -22 °C < T < -10 °C | MCC: -22 °C < T < -18 °C / FC: -22 °C < T < -18 °C | MCC (poly): 0.71-4.59 / FC (poly): 0.81-4.95 |
| 4 | *EDB | Electrodynamic balance levitator | No | Hoffmann et al., 2013a; 2013b | x | | | -40 °C < T < -1 °C | MCC: -32 °C < T < -29 °C | MCC (320 nm): 7.4 / MCC (800 nm): 1.3 |
| 5 | *,¹FRIDGE_default | Substrate-supported diffusion cell | Yes | Schrod et al., 2016 | x | | | -30 °C < T < 0 °C | MCC: -30 °C < T < -16 °C | MCC (poly): 1.82 |
| 6 | *INKA | Cylindrical plates CFDC | No | Schiebel, 2017 | x | | | -60 °C < T < -10 °C | MCC: -32 °C < T < -25 °C | MCC (poly): 3.35 |
| 7a | *,²LACIS_dry | Laminar flow tube | No | Hartmann et al., 2011; Wex et al., 2014 | x | | | -40 °C < T < -5 °C | MCC: -36 °C < T < -27 °C | MCC (poly): 7.00 |
| 7b | *,³LACIS_wet | Laminar flow tube | No | Grawe et al., 2016 | x | | | -40 °C < T < -5 °C | MCC: -35 °C < T < -30 °C | MCC (700 nm): 5.70 |
| 8 | MRI-DCECC | Dynamic CECC | No | Tajiri et al., 2013; Hiranuma et al., 2015a | x | | | -100 °C < T < -0 °C | MCC: -28 °C < T < -16 °C | MCC (poly): 1.36 |
| 9 | PNNL-CIC | Parallel plates CFDC | Yes | Friedman et al., 2011 | X | | | -55 °C < T < -15 °C | MCC: -28 °C < T < -20 °C | MCC (600 nm): 6.67 |
| 10 | *BINARY | Cold stage-supported droplet assay | No | Budke and Koop., 2015 | X | x | x | -30 °C < T < -0 °C | MCC: -27 °C < T < -22 °C / FC: -29 °C < T < -22 °C / NCC: -25 °C < T < -20 °C | MCC (bulk): 0.068 / FC (bulk): 0.087 / NCC (bulk): 1.24 |
| 11 | ⁵CMU-CS | Cold stage-supported droplet assay | No | Polen et al., 2016; Beydoun et al., 2017 | X | x | x | -30 °C < T < -0 °C | MCC: -30 °C < T < -20 °C / FC: -29 °C < T < -22 °C / NCC: -25 °C < T < -19 °C | MCC (bulk): 0.068 / FC (bulk): 0.087 / NCC (bulk): 1.24 |





| ID | Instrument | Description | Mobile? | Reference | Cellulose type | | | Investigable T-range | Investigated T-range for this study | SSA (m² g⁻¹)** |
| --- | --- | --- | --- | --- | --- | --- | --- | --- | --- | --- |
| | | | | | MCC | FC | NCC | | | |
| 12 | *FRIDGE-CS | Cold stage-supported droplet assay | Yes | Hiranuma et al., 2015b | x | | x | $-29\,°C < T < 0\,°C$ | MCC: $-28\,°C < T < -19\,°C$ / NCC: $-23\,°C < T < -13\,°C$[6] | MCC (poly): 1.71[7] / NCC (bulk): 1.24 |
| 13 | Leeds-μL-NIPI | Nucleation by immersed particles instrument | Yes | Whale et al., 2015 | x | x | | $-36\,°C < T < \sim 0\,°C$ | MCC: $-21\,°C < T < -17\,°C$ / FC: $-20\,°C < T < -16\,°C$ | MCC (bulk): 0.068 / FC (bulk): 0.087 |
| 14 | [8]LINDA | Immersion mode ice spectroemeter | Yes | Stopelli et al., 2014 | x | x | x | $-18\,°C < T < \sim 0\,°C$ | MCC: $-18\,°C < T < -12\,°C$ / FC: $-18\,°C < T < -11\,°C$ / NCC: $-11\,°C < T < -4\,°C$ | MCC (bulk): 0.068 / FC (bulk): 0.087 / NCC (bulk): 1.24 |
| 15 | *M-AL | Acoustic droplet levitator | No | Diehl et al., 2014 | x | x | x | $-30\,°C < T < \sim 0\,°C$ | MCC: $-21\,°C < T < -14\,°C$ / FC: $-22\,°C < T < -14\,°C$ / NCC: $-24\,°C < T < -13\,°C$ | MCC (bulk): 0.068 / FC (bulk): 0.087 / NCC (bulk): 1.24 |
| 16 | *M-WT | Vertical wind tunnel | No | Szakáll et al., 2010; Diehl et al., 2011 | x | x | | $-30\,°C < T < \sim 0\,°C$ | MCC: $-23\,°C < T < -12\,°C$ / FC: $-24\,°C < T < -22\,°C$ | MCC (bulk): 0.068 / FC (bulk): 0.087 |
| 17 | NC State-CS | Cold stage-supported droplet assay | No | Wright and Petters, 2013 | x | x | x | $-40\,°C < T < \sim 0\,°C$ | MCC: $-24\,°C < T < -16\,°C$ / FC: $-24\,°C < T < -16\,°C$ / NCC: $-24\,°C < T < -17\,°C$ | MCC (bulk): 0.068 / FC (bulk): 0.087 / NCC (bulk): 1.24 |
| 18a | NIPR-CRAFT | Cold stage-supported droplet assay | No | Tobo, 2016 | x | x | x | $-34\,°C < T < \sim 0\,°C$ | MCC: $-28\,°C < T < -20\,°C$ / FC: $-28\,°C < T < -21\,°C$ / NCC: $-31\,°C < T < -17\,°C$ | MCC (bulk): 0.068 / FC (bulk): 0.087 / NCC (bulk): 1.24 |
| 18b | [9]NIPR-CRAFT_<10 μm | Cold stage-supported droplet assay | No | Tobo, 2016 | x | x | | $-34\,°C < T < \sim 0\,°C$ | MCC: $-28\,°C < T < -17\,°C$ / FC: $-28\,°C < T < -17\,°C$ | MCC (<10 μm): 3.35[10] / FC(<10 μm): 3.35[10] |
| 19 | WISDOM | Microfluidic device-supported droplet assay | No | Reicher et al., 2018 | x | | x | $-40\,°C < T < \sim 0\,°C$ | MCC: $-33\,°C < T < -24\,°C$ / NCC: $-35\,°C < T < -20\,°C$ | MCC (bulk): 0.068 / NCC (bulk): 1.24 |
| 20 | WT-CRAFT | Cold stage-supported droplet assay | No | Tobo, 2016 | x | x | | $-34\,°C < T < \sim 0\,°C$ | MCC: $-26\,°C < T < -17\,°C$ / FC: $-26\,°C < T < -18\,°C$ | MCC (bulk): 0.068 / FC (bulk): 0.087 |

*Instruments of INUIT project partners, **Specific surface area; poly = polydispersed particles, homo = homogenised particles, bulk = bulk material, 1. Default deposition nucleation mode operation, 2. Experiments with dry dispersed-aerosol injection, 3. Experiments with atomized-aerosol injection, 4. Homogenized-sample data, 5. 0.001-1 wt% aqueous suspensions employed, 6. Experiments with 1.2 wt% non-diluted suspension, 7. TSI-OPS basis, 8. Experiments with both dissolved mass in solution and dry powder mass, 9. Experiments with size-selected (<10 μm) particles, 10. The AIDA-derived geometric SSA value (3.35 m² g⁻¹) was used since it accounts for only <10 μm particles.





**Table 3.** Quantitative method descriptions of dry dispersion techniques

| ID | Instrument | Aerosol size | SSA (m² g⁻¹)[1] | Droplet size (volume) | Droplet number examined per experiment | Typical ratio of the MCC size to the droplet size | Cooling rate or ice nucleation time | Ice nucleation parametrization[2] | Uncertainties | $\Delta\log(n_{s,geo})/\Delta T$ for MCC)* | Solution wt% (if used) |
|---|---|---|---|---|---|---|---|---|---|---|---|
| 1 | AIDA | MCC and FC: polydisperse (mode ~1.2 µm) NCC: Polydisperse (mode ~200 nm) | MCC: 3.35, FC: 3.35, NCC: 18.59 | 9.38 µm on average ($4.32 \times 10^{-7}$ µL) | $2.73 \times 10^9$ to $7.19 \times 10^{10}$ assuming full droplet activation in 84 m³ vessel | 0.13 | $0.90 \pm 0.2$ °C min[1,3] | Eqn. (1); AF using a combination of CPC, SMPS and APS for aerosol count | Temperature ± 0.3 °C (Möhler et al., 2003), $RH_w$ ± 5%, respectively (Fahey et al., 2014), $n_{s,geo}(T)$ for immersion freezing of ± 35% (Steinke et al., 2011) | 0.24 | NCC: 0.14[4] |
| 2 | CSU-CFDC | MCC: both polydisperse (mode at ~1.3 µm) and 500 nm (DMA 3081, TSI), FC: 500 nm (DMA 3081, TSI), NCC: 600 nm (DMA 3081, TSI) | MCC (poly): 2.09, MCC (500 nm): 8.00, FC (500 nm): 8.00, NCC (600 nm): 6.67 | ~2.6 µm ($9.20 \times 10^{-9}$ µL) for 0.5 µm dry particles at 5% $SS_w$ and a CFDC temperature of -30 °C according to the model result; For a 1.5 micron dry particle, the droplet size for 105% RH is 3.0 microns ($1.41 \times 10^8$ µL) | MCC and FC: 150,000; NCC: 1,500,000 | 0.19 (500 nm) - 0.40-0.50 (poly) | N/D (No Data) | Eqn. (1); AF using a combination of CPC, SMPS and APS for aerosol count | Temperature ± 0.5 °C, $n_{s,geo}(T)$ for immersion freezing of ± 60%, $RH_w$ ± 1.6, 2 and 2.4% at -20, -25, and -30 °C, respectively | 0.05 (500 nm) - 0.39 (poly) | NCC: 0.03 |
| 3 | DFPC-ISAC | MCC and FC: polydisperse (mode ~300 nm) | MCC: 0.71-4.59 FC: 0.81-4.95 Values varied depending on the cyclone impactor cut-size[5] | N/D | ~300-400 (examined crystals) | N/A (Deposition) | 15 min | Eqn. (1); AF using a total aerosol count of OPC (> 0.3 µm diameter) | Temperature ± 0.1 °C, Saturation ratio, $S_w$ at -22 °C of 1.02 ± 0.01, OPC error of ± 33%, The overall $n_{s,geo}(T)$ uncertainties of ~35% | 0.24 | MCC and FC: 0.1 |
| 4 | EDB | MCC: 320 and 800 nm (DMA 3081, TSI)[6] | MCC (320 nm): 7.4, MCC (800 nm): 1.3 | 90 ± 5 µm ($3.82 \times 10^{-4}$ ± $6.54 \times 10^{-8}$ µL) | 100-200 (Hoffmann et al., 2013a; 2013b) | 0.0036-0.0089 (Contact) | <30 s | FF derived from the ratio of ice crystals to the total number of droplets[7] | Temperature ± 0.2 °C, $n_{s,geo}(T)$ for immersion freezing of ~two orders of magnitude (in part because of the aspherical shape of the particles) | 0.38-0.44 | N/A |



| ID | Instrument | Aerosol size | SSA (m² g⁻¹)[1] | Droplet size (volume) | Droplet number examined per experiment | Typical ratio of the MCC size to the droplet size | Cooling rate or ice nucleation time | Ice nucleation parameterization[2] | Uncertainties | $\Delta\log(n_{s,geo})/\Delta T$ for MCC* | Solution wt% (if used) |
|---|---|---|---|---|---|---|---|---|---|---|---|
| 5 | FRIDGE-default | MCC: polydisperse (rather equally distributed from 300nm-5μm, no mode derivable) | MCC (dep.): $1.82^8$; NCC: N/D (presumed to be same as AIDA) | No supercooled droplets are formed when FRIDGE works in a default mode. | No droplets (default mode), activated INPs: 100-1000[9] | N/A (Deposition) | 100 s | Eqn. (1); AF is derived from the ratio of ice crystals on a wafer and total number of aerosols is estimated by an TSI OPS (0.3-10 μm diameter). | Temperature ± 0.2 °C, $n_{s,geo}(T)$ ±40% at -20 °C, The $n_{s,geo}(T)$ error may become lower with decreasing temperature. | 0.17 | N/A |
| 6 | INKA | MCC: polydisperse (same as AIDA) | N/D (presumed to be same as AIDA) | N/D | Not Provided | N/D | ~10 s | Eqn. (1); AF using a combination of CPC, SMPS and APS for aerosol count | Temperature ± 1.0 °C, $S_w$± 5% | 0.14 | N/A |
| 7 a | LACIS_dry | MCC: polydisperse (mode size 0.6 μm) | MCC (poly): 7.00 | ~5 μm ($6.54 \times 10^{-8}$ μL) | >2000 | 0.12 | 1.6 s (Wex et al., 2014; Hartmann et al., 2011) | Eqn. (1); FF (full expression, not approximated) | Temperature ± 0.3 °C, The error in $n_{s,geo}(T)$ at -31 °C is ~25% | 0.17 | N/A |
| 7 b | LACIS_wet | MCC: 700 nm (DMA type Vienna Hauke medium) | MCC (700 nm): 5.70 | ~5 μm ($6.54 \times 10^{-8}$ μL) | >2000 | 0.14 | | | | 0.05 | MCC: 1.0 |
| 8 | MRI-DCECC | MCC: polydisperse (mode diameter of ~2.2 μm.) | MCC (poly): 1.36 | <30 μm (<$1.41 \times 10^{-5}$ μL) | $4.66 \times 10^8$ to $1.92 \times 10^9$ (H15a) assuming full droplet activation in 1.4 m³ vessel | 0.35 | 2.4-2.8 °C min⁻¹ | Eqn. (1); AF using a combination of CPC, SMPS and APS | Temperature ±1.0 °C, 61% percent relative uncertainty in $n_{s,geo}(T)$ (Hiranuma et al., 2015a) | 0.17 | N/A |
| 9 | PNNL-CIC | MCC: 600 nm (DMA 3081, TSI) | MCC (600 nm): 6.67 | ~5 μm ($6.54 \times 10^{-8}$ μL) | Not Provided | 0.12 | ~12 s | Eqn. (1); AF based on the CPC aerosol count | Temperature ± 1.0 °C, $RH_w$± 3%. The $n_{s,geo}(T)$ error is ~± one order of magnitude at any $n_{s,geo}(T)$ space.[10] | 0.13 | N/A |

*The slope parameters of the other sample types for each technique are discussed in **Sect. 4.3.** , 1. Specific surface area, 2. Activated Fraction (AF) or Frozen Fraction (FF) - AF is calculated as the ratio of detected ice crystals to the number of total aerosol particles measured, whereas FF is derived from the ratio of ice crystals to the total particles detected in the subset of the sample (e.g., # of droplets) (Burkert-Kohn et al.,





2017). Our observation suggests that *AF*-based techniques appear to show higher $n_{s,geo}(T)$ than *FF*-based ones at $T > -16\,°C$. This is opposite to the observation addressed in *Burkert-Kohn et al.* (2017), where two in-situ *FF* techniques (including LACIS) showed *FF* that were roughly a factor of 3 above the *AF* values determined from two CFDCs., A similar observation is addressed in *Burkert-Kohn et al.* (2017)., 3. Average ± standard error calculated using the data recorded every five seconds for 90-400 sec (0.65-1.11 °C min⁻¹), 4. ~3 mL of 3wt% NCC in 100 mL of Milli-Q H₂O, 5. Summarized in **Table 9** - relevant discussions are give in **Sect. 4.3.2**., 6. Surface area has been calculated from SEM images of MCC particles collected on Nuclepore membrane filters., 7. *FF* was then converted into probability of freezing on a single collision ($e_c$) taking into account the rate of collision., 8. Measured with an OPS and corrected for a factor of 0.45, 9. The optimum number of INPs is 100-1000. The average number of cellulose particles per wafer was ~2x10⁵., 10. Complete activation of water droplets was not observed; therefore, there may have been the chance of underestimating the INP concentration.





**Table 4.** Quantitative method descriptions of aqueous suspension techniques

| ID | Instrument | Aerosol size | SSA (m² g⁻¹)[1] | Equivalent droplet size (volume) | Droplet or vial number examined per experiment | Typical ratio of the MCC size to the droplet size[2] | Cooling rate (°C min⁻¹) | IN parameterization[3] | Uncertainties | Δlog($n_{s,geo}$)/ΔT for MCC* | Solution wt% |
|---|---|---|---|---|---|---|---|---|---|---|---|
| 10 | BINARY | Bulk (Table 1) | MCC (bulk): 0.068 FC (bulk): 0.087 NCC (bulk): 1.24 | 1,046 μm (0.6 μL) | 36 or 64 | 0.019 (0.001 wt%) | 1.0 | Eqns. (3)-(5); *FF* | Temperature ±0.3 °C[4], $n_m(T)$ ±20% based on Gaussian error calculation and 35% for the maximal error | 0.38 | All: 0.001 to 0.1 |
| 11 | CMU-CS | Bulk (Table 1) | MCC (bulk): 0.068 FC (bulk): 0.087 NCC (bulk): 1.24 | 576 μm (0.1 μL) | 30-40 | 0.009 (0.0001 wt%) | 1.0 | Eqns. (3)-(5); *FF* | Temperature ±0.5 °C, *FF* uncertainties are on average 46, 57 and 75% for NCC, FC and MCC based on 95% confidence levels. | 0.20 | MCC: 0.0001, 0.01, 0.05 and 0.1; FC: 0.01, 0.1 and 1; NCC: 0.003, 0.03 and 0.1 |
| 12 | FRIDGE-CS[5] | Bulk (Table 1) and polydisperse (no mode derivable) | MCC (poly): 1.71 NCC (bulk): 1.24 | 985 μm (0.5 μL) | ~100[6] | 0.0087 (0.0001 wt%) | 1.0 | Eqns. (3)-(5); *FF* | Temperature ±0.2 °C, $n_{s,geo}(T)$ >20%[7] | 0.31 | MCC: 0.00010, 0.00020, 0.00043, FC: 0.00201, 0.00269, 0.02368, NCC: 0.049, 0.0049, 0.00049, 0.000049, 0.0000049 |
| 13 | Leeds-μl-NIPI | Bulk (Table 1) | MCC (bulk): 0.068 FC (bulk): 0.087 | 1,241 μm (1 μL) | ~40 | 0.0874 (0.1 wt%) | 1.0 | Eqns. (3)-(5); *FF* | Temperature ±0.4 °C, Our $n_{s,geo}(T)$ error bars are calculated by propagating the uncertainties from droplet volume and weighing of the cellulose and water (*Whale et al.*, 2015). | 0.47 | MCC and FC: 0.1 |
| 14 | LINDA | Bulk (Table 1) | MCC (bulk): 0.068 FC (bulk): 0.087 NCC (bulk): 1.24 | Bulk solution (100 μL) | 52 | 0.0874 (0.1 wt%) | 0.4 | Eqns. (3)-(5); *FF* | Temperature ±0.2 °C, cumulated uncertainties (counts and temperature) of $n_{s,geo}(T)$ -48% to +64% for counts of 1 INA/mL, uncertainties of -36% to +59% for counts of 10 INA/mL | 0.29 | All: 0.1[8] |
| 15 | M-AL | Bulk (Table 1) | MCC (bulk): 0.068 FC (bulk): 0.087 NCC (bulk): 1.24 | 1,900-2,100 μm (3.59-4.85 μL) | 100 | 0.0874 (0.1 wt%) | N/A | Eqns. (3)-(5); *FF* | Temperature ±0.7 °C, Our $n_{s,geo}(T)$ uncertainties for MCC, FC and NCC are on average 33%, 17% and 23%, respectively.[9] | 0.40 | MCC and FC: 0.1 and 1, NCC: 0.001, 0.01 and 0.1 |


| ID | Instrument | Aerosol size | SSA (m² g⁻¹)[1] | Equivalent droplet size (volume) | Droplet or vial number examined per experiment | Typical ratio of the MCC size to the droplet size[2] | Cooling rate (°C min⁻¹) | IN parameterization[3] | Uncertainties | $\Delta\log(n_{s,geo})/\Delta T$ for MCC* | Solution wt% |
|---|---|---|---|---|---|---|---|---|---|---|---|
| 16 | M-WT | Bulk (Table 1) | MCC (bulk): 0.068 FC (bulk): 0.087 | 700 µm (0.18 µL) | 50 | 0.0874 (0.1 wt%) | Isothermal | Eqns. (3)-(5); FF | Temperature ± 0.5 °C, The $n_{s,geo}(T)$ errors for MCC and FC are 26-48% and 32-53%, respectively,[10] | 0.26 | MCC and FC: 0.1 |
| 17 | NC-State CS | Bulk (Table 1) | MCC (bulk): 0.068 FC (bulk): 0.087 NCC (bulk): 1.24 | 1,241 µm (1 µL) | 64 (MCC and NCC) 200 (NCC) | 0.874 (1 wt%) | 2.0 | Eqns. (3)-(5); FF | Temperature ± 1 °C for MCC and FC, and ± 0.2 °C NCC, based on manufacturer specified thermistor accuracy. Uncertainties in INP concentration per unit liquid are derived based on one standard deviation of INP concentrations derived at each whole Kelvin across each experiment on the sample.[11] | 0.29 | MCC and FC: 1.0, NCC: 0.05 |
| 18 | NIPR-CRAFT | Bulk (Table 1) and <10 µm[12] | MCC (bulk): 0.068 MCC (<10 µm): 3.35[13] FC (bulk): 0.087 FC (<10 µm): 3.35[13] NCC (bulk): 1.24 | 2,122 µm (5 µL) | 49 | 0.0041-0.0188 (0.00001-0.001 wt%) | 1.0 | Eqns. (3)-(5); FF | Temperature ± 0.2 °C | 0.41 | All: 0.00001, 0.001 and 0.1 |
| 19 | WISDOM | Bulk (Table 1) | MCC (bulk): 0.068 NCC (bulk): 1.24 | 34-96 µm (0.02-0.46 nL) | 120-550 | 0.0693 (0.05 wt%) | 1.0 | Eqns. (3)-(5); FF | Temperature ± 0.3 °C, The error in $n_{s,geo}(T)$ of 16% is based on 95% confidence interval. Further uncertainty may arise from the BET surface area uncertainty (12%) and droplet volume identification (7%). | 0.26 | MCC: 0.05, NCC, 1.00-1.33 |
| 20 | WT-CRAFT | Bulk (Table 1) | MCC (bulk): 0.068 FC (bulk): 0.087 | 1,789 µm (3 µL) | 49 | 0.0322 (0.005 wt%) | 1.0 | Eqns. (3)-(5); FF | Temperature ± 0.5. The $C_{NP}$ and $n_m$ uncertainties are ±23.5% based on the relative standard error of three measurements of 0.05 wt% FC (sonicated samples). | 0.36 | MCC and FC: 0.05 and 0.005 |

*The slope parameters of the other sample types for each technique are discussed in **Sect. 4.3.**, 1. Specific surface area, 2. The aerosol size is based on the mass equivalent aerosol diameter for the given weight percent, at which ice nucleation ability of MCC was evaluated for <-20 °C. This temperature range is directly comparable to the dry dispersion measurements., 3. Activated Fraction (AF) or Frozen Fraction (FF) - AF is calculated as the ratio of detected ice crystals to the number of total aerosol particles measured, whereas FF is derived from the ratio of ice crystals to the total particles detected in the subset of the sample (e.g., # of droplets) (Burkert-Kohn et al., 2017). Our observation suggests that AF-based techniques appear to show higher $n_{s,geo}(T)$ than FF-based ones at $T$ >-16 °C. This is opposite to the observation addressed in Burkert-Kohn et al. (2017), where two in-situ FF techniques (including LACIS) showed FF that were roughly a factor of 3 above the AF values determined from two CFDCs, 4. See Budke and Koop



(2015) for more details., 5. The dew point maintained to avoid evaporation/condensation while measuring. Note that we utilized aerosolized particles collected on filters and scrubbed with deionized water. The measured geometric SSA of dispersed MCC was 1.71 m² g⁻¹, 6. We typically carry out a set of >four runs with >~100 droplets per run., 7. Higher $n_{s,geo}(T)$ uncertainties may coincide with the high temperature quartile because the span of the confidence interval is relatively wider when there exists only few frozen droplets., 8. Suspension was prepared in two different ways for MC and FC. 1) solution of 0.1 wt% sonicated and vortexed, 2) powder in the vials and addition of NaCl 0.1 wt% solution to the desired final weight percent cellulose of 0.1 wt%. NCC prepared as 1). Cellulose fibers tend to sediment and form clumps in solution., 9. The $c_{INPS}(T)$ and $n_{s,geo}(T)$ uncertainties were calculated taking the errors of the frozen fractions of drops, the specific particle surface area, the particle masses per drop, and the drop sizes into account., 10. The $c_{INPS}$ and $n_s$ uncertainties include errors of the frozen fractions of drops, the specific particle surface area, the particle masses per drop and the drop sizes., 11. For each sample multiple experiments were performed. An experiment consists of working with the same stock sample, and placing n droplets in droplets on the cold stage, cooling the stage. For the next experiment a new set of slides and droplets are prepared (MCC – 3 experiments ~64 drops/experiment; FC – 4 experiments ~64 drops/experiment; NCC – 3 experiments ~64 drops/experiment; Filtered Water – 3 experiments ~200 drops/experiment; Unfiltered Water – 7 experiments ~64 drops/experiment). Individual INP spectra are binned to produce INAS concentrations in 1 K intervals. Reported INP spectrum's concentrations were produced by averaging the INAS concentration across each individual spectra. Note that droplets were placed on a hydrophobic glass slide and in contact with N₂. Oil immersion was not used., 12. Experiments with size-selected (<10 μm) particles, 13. The AIDA-derived geometric SSA value (3.35 m² g⁻¹) is used since it accounts for only <10 μm particles.



**Table 5.** Nominal method descriptions of dry dispersion techniques

| ID | Instrument | Dispersion method | Impactor type | Background correction method (if any) | Ice detection method | Valid data range | Sample pre-treatment | Solvent type (if used) |
|---|---|---|---|---|---|---|---|---|
| 1 | AIDA | MCC and FC: Rotating Brush (RBG1000, PALAS), NCC: modified atomizer[1] | Cyclone ($D_{50}$ of 5 µm) combined with Rotating Brush | Background was neglected and no corrections was applied.[2] | Ice number counting per unit volume of air with optical particle counters (WELAS 2300 and 2500, PALAS, *Benz et al.*, 2005) | For MCC and FC, to exclude any possible artifacts from the chamber operation (e.g., sparse ice peak detection during abrupt cooling at the beginning), we examined data for 90-400 sec after the initial cooling and 1 min averaged AF >0.5% (INUIT06_07 for MCC, INUIT06_14 for FC). For NCC, we examined data for 90-400 sec after the initial cooling and welas count ~>0.1 p cm⁻³ (CIRRUS01_58). | Grinding MCC/FC with a mortar and pestle, Sonicating NCC for 30 min prior to the injection | Milli-Q water for NCC |
| 2 | CSU-CFDC | MCC and FC: Flask in a sonic bath and blowing dry N₂ over the sample, NCC: Medical nebulizer | Inertial impactor (cut-size of 2.4 µm) | Background INP concentrations calculated by taking measurements through a filter for 2-3 minutes before and after the sample period were accounted.[3] | Ice number counting per unit volume of air with an optical particle counter (OPC; CLiMET, model CI-3100) | This CFDC provided data for condensation/immersion freezing at -21.2, -25.1 and -29.7 °C (a total of eight data points with two, two and four points at around each temperature, respectively), which extended to a warmer region than the AIDA measurements. As demonstrated in *DeMott et al.* (2015), higher RHw values (105%) are required for full expression of immersion freezing in CSU-CFDC. | N/A | DI water for NCC |
| 3 | DFPC-ISAC | MCC(dry): Custom-built flask dust generator[4], MCC(wet): Nubulizer (AGK 2000, PALAS) | Cyclone ($D_{50}$ of 7, 1 and 0.5 µm at 2, 12 and 3.5 lpm, respectively) | Background INP concentrations obtained by using blank filters (filters taken from the batch and processed into the DFPC chamber) were accounted.[5] | Visual inspection of individual freezing events based on an USB optical microscope (eScope) imagery and later inspected with ImageJ[6] | N/A | The suspensions were hand shaken before nebulization. A magnetic stirrer was used to keep the cellulose particles suspended. | MilliQ water for MCC and FC |
| 4 | EDB | Turbulent flow disperser[7] | Cyclone ($D_{50}$ of 1 µm) | N/A | Visual inspection of individual freezing events according to the enhancement of scattered light on the linear CCD array upon freezing (*Hoffmann et al.*, 2013a) | N/A | N/A | Milli-Q water for MCC |


| ID | Instrument | Dispersion method | Impactor type | Background correction method (if any) | Ice detection method | Valid data range | Sample pre-treatment | Solvent type (if used) |
|---|---|---|---|---|---|---|---|---|
| 5 | FRIDGE-default | Mixing powder samples with a magnetic stirrer[8] | 47 mm hydrophobic Fluoropore PTFE membrane with a 0.45 μm pore size bonded to a high-density polyethylene support produced by Merckmillipore® | The absolute number of ice crystals of a blank wafer was subtracted from the absolute number of ice crystals on a loaded wafer.[9] | Visual inspection of individual freezing events based on the CCD camera imagery of growing ice crystals | N/A | N/A | N/A |
| 6 | INKA | Rotating Brush (RBG1000, PALAS) | Cyclone ($D_{50}$ of 5 μm) combined with Rotating Brush | An experiment started with a 2 minutes background measurement while sampling through a particle filter.[10] | Ice number counting per unit volume of air with an optical particle counter (OPC; CLiMET, model CI-3100) | This CFDC provided data for condensation and/or immersion freezing at around -25, -27.5, -30 and -30.5 °C (a total of eight data points with two, two, three and one point at around each temperature, respectively). Since INKA is of the same operational design as the CSU-CFDC, here also higher $RH_w$ values (107%) were required for full expression of immersion freezing (*DeMott et al.,* 2015). | N/A | N/A |
| 7a | LACIS_dry | Flask with an electric motor and blowing particle-free pressurized air input over the sample | Cyclone ($D_{50}$ of 625 nm at 3 lpm) | N/A[11] | Ice number counting per unit volume of air according to the custom-built optical particle spectrometer, called TOPS-Ice (Thermo-stabilized Optical Particle Spectrometer for the detection of Ice; *Clauss et al.,* 2013) | N/A | N/A | N/A |
| 7b | LACIS_wet | Modified atomizer[1] | N/A | | | N/A | We sonicated the sample for 10 minutes. The cumulative time required to obtain a sufficiently high number concentration at 700 nm was a week.[12, 13] | MilliQ water for MCC |





| ID | Instrument | Dispersion method | Impactor type | Background correction method (if any) | Ice detection method | Valid data range | Sample pre-treatment | Solvent type (if used) |
|---|---|---|---|---|---|---|---|---|
| 8 | MRI-DCECC | Rotating Brush (RBG1000, PALAS) | Cyclone ($D_{50}$ of 2.5 µm and 1.0 µm) | No corrections were applied. Prior to experiments, a blank expansion was carried out to confirm the background non-IN active particle concentration of <0.1 cm⁻³. | Ice number counting per unit volume of air with optical particle counters (WELAS Promo2000H, PALAS, Benz et al., 2005) | N/A | N/A | N/A |
| 9 | PNNL-CIC | SSPD (Model 343, TSI) | N/A | Background INP concentrations calculated by taking measurements through a filter for 5 minutes before and after the sample period were accounted.[14] | Ice number counting per unit volume of air with an optical particle counter (OPC; CLiMET, model CI-3100). | 0.01 < AF < 0.95 - Below 0.01 fraction, sensitivity of the instrument became an issue and was dependent upon particle concentration. Upper limit was governed by the particle losses in the system. | N/A | N/A |

1. Similar to the commercially available atomizer (TSI 3076) drilled through an opposite orifice (Wex et al., 2015), 2. A blank reference expansion (Hiranuma et al., 2014) was carried out prior to a series of experiments to achieve the background non-IN active particle concentration in the chamber of <0.3 cm⁻³., 3. A weighted average of the background INP concentration is calculated from the two filter periods and is subtracted from the average INP concentration of the sample period (Schill et al., 2016)., 4. Flow rate of ~12 lpm was employed. Cyclones (SCC, BGI, Inc.) were deployed downstream of the flask to exclude particles larger than certain aerodynamic diameter with varied cut-sizes (Table 9)., 5. In order to measure water background, we nebulized pure Milli-Q grade water onto Millipore filters and examined residuals to make sure no presence of water impurity. The filters were then processed with our DFPC chamber at -22 °C. The averaged crystal number on filter of seven was subtracted from the crystal number measured using cellulose samples (typically the order of two hundreds)., 6. N$_{ice}$ is estimated by ImageJ software, followed by the Poisson statistic., 7. A flask containing cellulose and bronze beads is mixed with a magnetic stirrer and a synthetic air flow of 1 lpm., 8. Dry dispersion of cellulose into purified compressed air produced an aerosol concentration of approx. 10 cm⁻³ (MCC) and 40 cm⁻³ (FC)., 9. Background and particle losses (i.e., sampling efficiency, 90% of the surface of the wafer are analyzed) were accounted in our background corrections. Sampling volume was adjusted to avoid overloading of the wafers, water vapor depletion and merging of ice crystals before they were counted. So, the volume effect was neglected., 10. This procedure allowed to determine the background INPs caused by the chamber itself, which was then considered in the data analysis. In addition, particle losses in the sampling line were found to be negligible., 11. We did not observe any contribution from impurities in the water. For the detection of the homogeneous freezing limit, we used ammonium sulfate (dissolved in MilliQ water and sprayed with an atomizer) as seed particles for the droplets. We detected the first freezing of those highly diluted droplets at -38 °C. Hence, there was no need to correct the cellulose suspension data concerning the water background. We note that the experiment was stopped as soon as background originating from the ice covered walls was detected., 12. Swelling might have been an issue in the case of the suspension particles, because the sample needed to be prepared one week in advance. A 700 nm suspension particle was not necessarily comparable (in terms of chemical composition, morphology) to a 700 nm dry dispersed particle, but we did not investigate this further., 13. We found that the maximum of the size distribution depends on the suspension time of the cellulose particles. We measured size distributions directly after preparing the suspension, after one week and after two weeks, and observed size distribution broadening as well as a shift in mean diameter towards larger end., 14. A weighted average of the background INP concentration was calculated from the two filter periods and was subtracted from the average INP concentration of the sample period.


**Table 6.** Nominal method descriptions of aqueous suspension techniques

| ID | Instrument | Solvent type | Sample pre-treatment | Suspension Status[1] | Background correction method | Ice detection method | Valid data range |
|---|---|---|---|---|---|---|---|
| 10 | BINARY | Bidistilled water | MCC and FC: described in *Hiranuma et al.* (2015a), NCC: one min ultrasonic bath and at least 10 min stirring with a vortex shaker after dilution of a weighed sample until pipetting; storage at +3 °C | Continuous stirring | No additional correction applied. | CCD camera: the digital images obtained by a CCD camera (QImaging MicroPublisher 5.0 RTV) were analyzed at a frequency that depends upon the experimental cooling rate (*Budke and Koop*, 2015).[2] | FF 0.05-0.95[3] |
| 11 | CMU-CS | Milli-Q water | MCC and FC: left unrefrigerated; suspended and stirred with no further processing, NCC: left refrigerated until using the sample; followed the protocol given by INUIT | All suspensions were continually stirred while pipetting. Constant stirring was done with a teflon stirbar while droplets were pipetted. | Cutoff $T$ for background freezing was below -26 °C for these samples. All samples provided were given with the assumption that less than 10% of the FF would be attributable to water contamination. | Digital camera: The droplets were illuminated using a light-emitting diode light ring above the acrylic window, and the droplets were imaged using a stereomicroscope and digital camera (Amscope, *Polen et al.*, 2016).[4] | MCC and FC: *FF* 0.05-0.95[3], NCC: *FF* >0.05[5] |
| 12 | FRIDGE-CS (immersion) | DI water | No pre-treatment applied | The suspension tube was shaken every ~20 sec to achieve a homogeneous distribution of cellulose particles in all droplets.[6] | The frozen fraction of DI water was subtracted from that of the suspension samples.[7] | CCD camera: a CCD camera (2/3" CCD > 5 megapixels, 1 pixel ~ 400 μm²) was used to monitor and record the sample substrates.[8] | All range after the background correction |
| 13 | Leeds-μl-NIPI | Milli-Q water | Suspensions were stirred using a magnetic stirrer bar for approximately 30 min prior to pipetting out droplets. We did not sonicate suspensions. | Suspensions were continuously stirred during droplet preparation. | We used the freezing background and subtraction method described in *O'Sullivan et al.* (2015; i.e., Eqn. 1 and 2).[9] | Digital camera: The freezing of the droplets was monitored using a digital camera at a rate of one frame per sec. The first change in droplet structure (i.e., Fig. 2 of *Whale et al.*, 2015) leading to droplet freezing was taken to be the nucleation event, and this information was used to establish the fraction of droplets frozen as a function of $T$. | All range after the background correction |
| 14 | LINDA | 0.1 wt% NaCl solution | MCC and FC (Sus): 5 min sonication of suspension; manual shaking while pouring aliquots into vials, MCC and FC (Pow): 5 min sonication of grid with vials prior to analysis, NCC01: additional preliminary 15 min sonication of 3 wt% stock solution | Idle | No solvent vials froze until -18 °C. Therefore, no correction was applied. | CMOS camera: Images taken by a USB CMOS Monochrome Camera (DMK 72BUC02, The Imaging Source Europe GmbH, Bremen, Germany) were recorded every six sec (*Stopelli et al.*, 2014).[10] | All range after the background correction |





| ID | Instrument | Solvent type | Sample pre-treatment | Suspension Status[1] | Background correction method | Ice detection method | Valid data range |
|---|---|---|---|---|---|---|---|
| 15 | M-AL | CHROMASOLV water for HPLC (Sigma-Aldrich) | No pre-treatment applied | Idle[11] | The frozen fraction of HPLC water was subtracted from that of the suspension samples. | Digital camera and infrared thermometer: the drops were imaged by a digital video camera and the surface temperature of the drops were measured directly by an infrared thermometer with a temporal resolution of 0.5 sec (Diehl et al., 2014).[12] | FF 0.05-0.98[3] |
| 16 | M-WT | CHROMASOLV water for HPLC (Sigma-Aldrich) | No pre-treatment applied | Continuously stirring the suspension at a very low rate[13] | Since no freezing of HPLC water droplets was observed within the investigated temperature range, no background correction was applied.[14] | Visual observation during levitation[15] | FF 0.05-0.95[3,16] |
| 17 | NC-State CS | HPLC Grade water (Aldrich) | All solutions were sonicated for 10 min prior to experimenting on the cold stage. | Idle | Background subtraction or correction was not applied in this study because median freezing temperatures for cellulose occurred several °C warmer than that of reference HPLC water. | Microscope camera: The droplets were imaged with a regular camera lens that was outfitted with a 2592 x 1944 pixel resolution camera (Infinity 1-5C; Lumenera, Wright and Petters, 2013).[17] | Temperature bins with ≥ 2 freeze events across all repeats (n = 3-7) |
| 18 | NIPR-CRAFT | Milli-Q water | No pre-treatment applied | Occasionally shaking a suspension tube while pipetting/preparing droplets | No ice nucleation of water was observed until ~30°C. Therefore, no correction was applied. | Webcamera: individual droplet freezing events were monitored and recorded by a commercially available WEB camera (Tobo, 2016).[18] | MCC and FC: $FF$ > 0.04[5], NCC: $FF$ > 0.02-0.96[3] |
| 19 | WISDOM | Deionized water, biological grade | MCC: after sonication was applied, 30 min idle before droplets generation following the INUIT protocol, NCC: three cycles of sonication (by Hielscher vial-tweeter), 30 sec each, with 10 sec idle between | Idle (the time required to generate droplets was 30 sec) | Since all suspension droplets froze prior to the solvent's freezing, no correction was made. | Microscope camera: freezing experiments were observed under a light microscope (Olympus BX-51, 10X magnification, transmission mode) and a video file was recorded during the measurement with a temporal resolution of 1 sec (or temperature resolution of 0.017C for 1CPM cooling rate, Reichar et al., 2018).[19] | All range after the background correction |
| 20 | WT-CRAFT | Milli-Q water | MCC and FC: sonication of 50 mL suspension in a falcon tube for 15 min | Idle | No correction was made.[20] | Webcamera (same as NIPR-CRAFT): manual counting of cumulative number of frozen droplets based on the color contrast shift in the off-the-shelf Webcamera (all videos recorded)[21] | $FF$ > 0.05[5]; $T$ > -26 °C (<3% pure water activation) |

1. Status of the suspension solution while generating droplets/vial, 2. Three successive images were analyzed per 0.1K temperature interval, i.e., one image every 0.03K. Ice nucleation was determined optically based on the change in droplet brightness when the initially transparent liquid droplets became opaque upon freezing. This change in brightness was maximized by illuminating the droplets by LEDs at a low sideway angle from the top and also by the reflective top surface of the Peltier stage., 3. The FF range was restricted thereby limiting the valid data range, as a non-homogeneous particle distributions in bulk solution was presumed and, therefore, individual droplets leading to sparse nucleation at both low and high temperature boundaries are excluded. In order to exclude the effects of "pure" water freezing data beyond FF 0.95 and higher was eliminated (this is an alternative to a water background subtraction). The impact of this correction was small as the resulting $n_{s,geo}(T)$ difference was within a factor of two., 4. Images were taken at a resolution of 1600×1200 with magnification of 7.5X at 0.17 °C intervals. Arrays containing between 30 and 40 droplets could be visualized. An image was recorded every 10s. Images



were analyzed manually to determine the temperature at which a liquid droplet (appearing gray) had frozen (appearing black)., 5. to exclude early freezers often represent the contaminant interference, 6. Aerosol was generated by dry dispersion of MCC particles. The particle number size distribution of this aerosol in the 0.3-10 μm diameter range was measured by an optical particle counter (3330, TSI). MCC particles were collected by filtration of the aerosol using cellulose nitrate membrane filters (Millipore, HABP04700). After sampling, the filters were placed in vials with 10 mL of deionized water. Particles were scrubbed from the filters by agitating for 10 min in an ultrasonic bath., 7. The background freezing contributed to <3% at -25 °C, <10% at -27.5°C and <20% at -29 °C. No evaporation/condensation was assumed., 8. LabView software was used to download images and detect changes in brightness of droplets (by comparing real time images with a reference image taken prior to the ice nucleation)., 9. To correct for the impact of background freezing on our data, we subtracted the K(T) values for a best fit to the background freezing curve from the K(T) values for the ice nucleation data. Where the data overlaps with the 68% confidence interval for the background freezing points were considered indistinguishable from the background and are not included. The cellulose data did not significantly overlap our background freezing., 10. LED array illuminated polycarbonate plate holding 52 sample tubes from the bottom. Light intensity in the area of each tube lid was extracted from each image and recorded into a text file together with the temperature at the time the image was taken., 11. Before refilling the medical syringe used for injecting droplets into M-AL, the suspension was stirred for approx. 20 sec. Before injecting, the syringe was shaken in order to homogenize the cellulose distribution in droplets., 12. The video camera allowed for the visual observation of the freezing process. The infrared thermometer was used to measure the surface temperature of the freezing drops with an accuracy of 0.7 K, while a Pt100 sensor was located in the vicinity of the drop to measure the ambient temperature. The freezing was detected as a sudden increase of the surface temperature to 0 °C., 13. Before the droplet injection, the syringe was shaken in order to homogenize the cellulose distribution in droplets., 14. Before each experiment, we carried out background test measurements, i.e. measurements with pure water droplets. The pure water drops were levitated in the tunnel for <35 s to minimize the effect of evaporation., 15. The experimenter observed the behavior of the levitating droplet; when the droplet freezes, it becomes opaque and its floating behavior changes abruptly., 16. Every single droplet was kept floating in the vertical air stream of the M-WT until it froze (within <35 sec). Freezing event within the first five seconds after injecting were presumably emanated from freezing triggered by contaminants and abandoned from our analysis. Conceptually, ~five sec is needed for a droplet to adapt its surface temperature to the ambient temperature., 17. The observation area was enclosed in a clear acrylic box and flushed with dry nitrogen to prevent frosting. Images were recorded in ~0.17 °C intervals and stored for post-processing. When a water drop froze, the drop darkened from a nearly transparent, white circle to a fully black circle. An in-house-developed algorithm processed the images to automatically detect potential freeze events. Suspected freeze events were inspected manually and determined to be either a true freeze event, a false positive, or a freeze event induced by drops coming in contact with each other., 18. Based on the video image analysis, the number fractions of droplets frozen and unfrozen relative to the total number of droplets were counted every 0.5 °C., 19. Individual freezing events of the droplets were detected automatically by image processing using homemade LabVIEW program. In the first stage, the program detected the droplets and their diameters by a shape criterion using VISION software. In the second stage, every droplet is surrounded by a square to create array of pixels. The gray level values of the array are analyzed in each frame of the movie and compared to the liquid droplet values. When the droplets froze, the small crystals were scattering more light and the droplet darkens. Hence, the average brightness in the square array decreased and the automatic program recorded this brightness negative peak as a freezing point., 20. We ran 3x7 of pure water (solvent) in the side by side position of solution droplets during the experiment to make sure no pure water droplets started freezing prior to the completion of solution droplets freezing. This simultaneous measurement ensured no freezing emanated from water itself. We discarded the experiment if we observed the freezing event of pure water prior to that of solution droplet., 21. Ice nucleation was determined optically based on the change in droplet brightness when the initially transparent liquid droplets became opaque upon freezing. If the freezing temperature was not obvious for any droplets, the 8-bit grayscale images were assessed on the ImageJ software to determine the temperature of phase shift for suspicious droplets by varying the minimum threshold gray value of 155-175 at the fixed maximum threshold value of 255.



**Table 7**. List of the Gumbel cumulative distribution fit parameters to the $n_{s,geo}(T)$ for $T$-binned ensemble datasets of MCC, FC and NCC (All). The datasets are fitted in the log space. Besides All, fit parameters for ensemble maximum values (All$_{max}$), ensemble minimum values (All$_{min}$), suspension subset (AS), and dry dispersed particle subset (DD) are also included in this table. The correlation coefficient, $r$, for each fit is also shown. All $n_{s,geo}(T)$ values are in m$^{-2}$. $T$ is in °C.

| Fitted dataset | Fitted $T$ range | Fit Parameters $[n_{s,geo}(T) = \exp(a \cdot \exp(-\exp(b \cdot (T+c)))+d)]$ | | | | | |
|---|---|---|---|---|---|---|---|
| | | $a$ | $b$ (°C$^{-1}$) | $c$ (°C) | $d$ | $r$ | $\Delta\log(n_{s,geo})/\Delta T$ |
| All (MCC) | -36 °C < $T$ < -12 °C | 24.47 | 0.12 | 15.99 | 3.24 | 0.96 | 0.32 |
| All$_{max}$ (MCC) | -36 °C < $T$ < -12 °C | 23.19 | 0.19 | 14.36 | 3.28 | 0.83 | 0.33 |
| All$_{min}$ (MCC) | -36 °C < $T$ < -12 °C | 27.95 | 0.08 | 18.67 | 3.03 | 0.95 | 0.30 |
| DD (MCC) | -36 °C < $T$ < -16 °C | 24.12 | 0.08 | 12.56 | 4.69 | 0.91 | 0.20 |
| AS (MCC) | -33 °C < $T$ < -12 °C | 28.03 | 0.10 | 18.22 | 3.48 | 0.97 | 0.37 |
| | | | | | | | |
| All (FC) | -29 °C < $T$ < -11 °C | 22.25 | 0.11 | 15.95 | 3.62 | 0.88 | 0.33 |
| All$_{max}$ (FC) | -29 °C < $T$ < -11 °C | 23.78 | 0.13 | 16.85 | 4.79 | 0.94 | 0.40 |
| All$_{min}$ (FC) | -29 °C < $T$ < -11 °C | 21.88 | 0.08 | 16.85 | 3.15 | 0.58 | 0.26 |
| DD (FC) | -29 °C < $T$ < -18 °C | 26.97 | 0.07 | 18.12 | 6.85 | 0.89 | 0.28 |
| AS (FC) | -29 °C < $T$ < -11 °C | 22.57 | 0.09 | 16.05 | 3.46 | 0.92 | 0.29 |
| | | | | | | | |
| All (NCC) | -35 °C < $T$ < -13 °C | 19.30 | 0.14 | 19.48 | 6.59 | 0.90 | 0.31 |
| All$_{max}$ (NCC) | -35 °C < $T$ < -13 °C | 17.22 | 0.18 | 17.36 | 7.30 | 0.93 | 0.29 |
| All$_{min}$ (NCC) | -35 °C < $T$ < -13 °C | 17.39 | 0.21 | 19.88 | 6.30 | 0.89 | 0.32 |
| DD (NCC) | -33 °C < $T$ < -15 °C | 16.40 | 0.18 | 17.33 | 7.45 | 0.97 | 0.29 |
| AS (NCC) | -35 °C < $T$ < -13 °C | 15.35 | 0.28 | 20.83 | 8.53 | 0.98 | 0.30 |



**Table 8.** $T$-binned $n_{s,geo}$ values (in m$^{-2}$) of three different cellulose samples based on the log average of all available results at $T$ [i.e., **Fig. 4 (iv)**]. The first MCC column represents reference immersion freezing $n_{s,geo}(T)$ values for MCC from H15a. The numbers in parentheses are maxima and minima of $n_{s,geo}$ at $T$. The numbers in brackets represent the number of instruments used to calculate the log average $n_{s,geo}$ at $T$. The $n_{s,geo}(T)$ values derived from a single instrument are not included in this table.

| $T$ (°C) | MCC (H15a) | MCC (Max-Min) $n_{s,geo}(T)$ (m$^{-2}$) | | | FC (Max-Min) | | | NCC (Max-Min) | | |
|---|---|---|---|---|---|---|---|---|---|---|
| -34 | | 5.94E+10 | (1.26E+11 - 2.80E+10) | [2] | | | | | | |
| -33 | | 5.85E+10 | (1.35E+11 - 2.62E+10) | [3] | | | | 1.09E+10 | (1.37E+10 - 8.75E+09) | [2] |
| -32 | | 3.33E+10 | (3.41E+11 - 6.42E+09) | [6] | | | | 8.28E+09 | (1.29E+10 - 5.30E+09) | [2] |
| -31 | | 2.88E+10 | (1.74E+11 - 4.77E+09) | [7] | | | | 8.10E+09 | (1.20E+10 - 5.46E+09) | [2] |
| -30 | | 1.07E+10 | (8.20E+10 - 9.31E+08) | [11] | | | | 8.19E+09 | (1.01E+10 - 6.64E+09) | [2] |
| -29 | | 5.89E+09 | (4.20E+10 - 7.61E+08) | [10] | | | | 5.36E+09 | (8.02E+09 - 3.58E+09) | [2] |
| -28 | 1.18E+10 | 3.25E+09 | (2.74E+10 - 2.51E+08) | [12] | 2.82E+09 | (3.61E+10 - 3.55E+08) | [3] | 2.79E+09 | (6.40E+09 - 1.29E+09) | [3] |
| -27 | 8.55E+09 | 2.06E+09 | (2.19E+10 - 2.21E+08) | [13] | 6.55E+08 | (1.05E+10 - 6.47E+07) | [4] | 2.08E+09 | (5.13E+09 - 1.02E+09) | [3] |
| -26 | 6.81E+09 | 1.23E+09 | (2.03E+10 - 1.35E+08) | [12] | 4.22E+08 | (4.08E+09 - 4.89E+07) | [4] | 1.27E+09 | (2.90E+09 - 7.73E+08) | [3] |
| -25 | 5.03E+09 | 7.97E+08 | (6.06E+09 - 4.55E+07) | [12] | 1.49E+08 | (3.65E+08 - 2.84E+07) | [4] | 4.52E+08 | (1.05E+09 - 2.62E+08) | [5] |
| -24 | 2.97E+09 | 4.46E+08 | (4.34E+09 - 3.93E+07) | [11] | 1.13E+08 | (4.68E+08 - 1.43E+07) | [5] | 1.12E+08 | (2.67E+08 - 3.87E+07) | [6] |
| -23 | 1.87E+09 | 1.32E+08 | (3.39E+09 - 4.15E+06) | [10] | 6.50E+07 | (3.30E+08 - 7.12E+06) | [6] | 1.89E+07 | (9.37E+07 - 3.01E+06) | [8] |
| -22 | 1.23E+09 | 5.33E+07 | (2.48E+09 - 8.72E+05) | [11] | 1.51E+07 | (2.19E+08 - 1.19E+06) | [7] | 3.66E+06 | (5.52E+07 - 5.46E+05) | [8] |
| -21 | 6.30E+08 | 2.32E+07 | (1.61E+09 - 2.54E+05) | [11] | 7.43E+06 | (1.42E+08 - 3.79E+05) | [8] | 1.07E+06 | (2.83E+07 - 2.09E+05) | [7] |
| -20 | 5.14E+08 | 1.21E+07 | (7.99E+08 - 1.23E+05) | [11] | 5.45E+06 | (1.05E+08 - 1.29E+05) | [6] | 3.21E+05 | (1.41E+07 - 3.21E+04) | [7] |
| -19 | 2.94E+08 | 6.63E+06 | (2.95E+08 - 2.42E+05) | [9] | 6.58E+06 | (1.03E+08 - 2.94E+05) | [6] | 2.31E+05 | (1.20E+07 - 4.27E+04) | [5] |
| -18 | 1.60E+08 | 1.42E+06 | (1.60E+08 - 9.33E+04) | [8] | 3.11E+06 | (1.03E+08 - 1.34E+05) | [6] | 1.21E+05 | (1.16E+07 - 1.77E+04) | [5] |
| -17 | 1.15E+08 | 6.31E+05 | (1.15E+08 - 3.83E+04) | [7] | 5.46E+05 | (1.70E+07 - 7.43E+04) | [6] | 6.42E+04 | (1.13E+07 - 4.44E+03) | [5] |
| -16 | 9.69E+07 | 5.89E+05 | (9.69E+07 - 3.76E+04) | [5] | 1.08E+05 | (2.48E+05 - 4.27E+04) | [4] | 1.59E+05 | (1.12E+07 - 5.05E+03) | [3] |
| -15 | | 2.79E+04 | (3.64E+04 - 2.14E+04) | [2] | 4.36E+04 | (1.12E+05 - 1.86E+04) | [4] | 1.25E+05 | (1.11E+07 - 3.30E+03) | [3] |
| -14 | | 1.48E+04 | (1.82E+04 - 1.21E+04) | [2] | 4.40E+04 | (5.85E+04 - 3.31E+04) | [2] | 9.01E+03 | (3.57E+04 - 2.27E+03) | [2] |
| -13 | | | | | 2.88E+04 | (3.79E+04 - 2.19E+04) | [2] | 6.67E+03 | (1.68E+04 - 2.65E+03) | [2] |





**Table 9.** Summary of the geometric *SSA* of MCC and FC particles assessed by DFPC-ISAC. In general, high *SSA* values indicate the presence of small grains because the relative dominance of the mass to the surface becomes small.

| Exp_ID | Avg. *SSA* ($m^2\ g^{-1}$) | Stdev. *SSA* ($m^2\ g^{-1}$) |
|---|---|---|
| MCC_Dry_7um_cut-size | 0.8 | 0.09 |
| MCC_Wet_no_cyclone | 3.12 | 0.1 |
| MCC_Wet_0.5um_ cut-size | 3.48 | 0.13 |
| MCC_Dry_1um_ cut-size | 4.37 | 0.24 |
| FC_Dry_7um_ cut-size | 0.9 | 0.1 |
| FC_Wet_no_cyclone | 3.11 | 0.11 |
| FC_Wet_0.5um_ cut-size | 3.57 | N/A |
| FC_Dry_1um_ cut-size | 4.91 | 0.35 |