# Peer review of "A comprehensive characterization of ice nucleation by three different types of cellulose particles immersed in water"

_Atmospheric Chemistry and Physics, 2018_

## Referee Comment (RC1) · Anonymous Referee #1 · 20 Nov 2018

**Review of "A comprehensive characterization of ice nucleation by three different types of cellulose particles immersed in water: lessons learned and future research directions," by N. Hiranuma et al. 2018**

It is an intimidating task to review a manuscript with such an extensive and esteemed authorship and I know most if not all of these scientists to be highly accomplished and dedicated to their scientific endeavors. However, I find the submitted manuscript in its current form to be lacking in multiple aspects. It does not have clear thread or cohesion of a story, lacks motivation from different angles, and suffers from being difficult to read in many places. There are many examples of both run-on sentences which lack clarity and short sentence/thought fragments. I would encourage the authorship to take a heavy editorial hand during any revisions. This said, the summarized work is a very ambitious set of experiments etc. and perhaps the manuscript is penalized by simply trying to collect and summarize such a wide ranging project in one report. Below I try to summarize my areas of major concern and append a section of "specific comments" where, when I have had the time I note specific editorial areas for attention.

From the beginning I am left a bit confused about what is the primary purpose of the manuscript. Immediately in §*1.3 Goals* the authors say, "The main objective of this study is to examine how different ice nucleation instrument techniques compare when using chemically homogeneous biological material rather than multi mineral systems,..." This type of explicit statement and the general way the manuscript seems to be framed is as a summary of how well ice nucleation tests of a single substance in laboratories around the world can be compared. In this context would AMT not have been a better choice for manuscript submission? For ACP, I would expect the motivations for the research to more clearly focus on the importance of cellulose to the physics and chemistry of the atmosphere. In its current state the single paragraph on page 3 motivating cellulose as important in an atmospheric context is not particularly convincing.

Within §*2. Sample Preparation and Characterization* the authors spend considerable space discussing the use of Laser Ablation mass spectrometry to characterize samples and also discuss ambient ALABAMA measurements. To me the motivation for this type of characterization is not clear and I am left to surmise that ultimately a demonstration of the laboratory relevance to ambient measurements is the goal. It is unclear what such a hard ionization treatment of cellulose, which in the case presented results in mass spectra of fragmented materials, can really illuminate. Cellulose has a high molecular weight $m/z \geq 160$, which is the weight of its basic building block levoglucosan. Thus it is unsurprising with laser ablation one generates mass spectra with many fragments, why would one expect anything different? To me the only thing that clearly emerges from the mass spectrometry is that the examined substances included have high molecular weight, and thus fragment to yield peaks at many lower molecular weights – therefore I am left wondering how Figures 1, 2 (d) and 3 further the discussion. Do the authors intend to assert that there are clear cellulose fingerprints and that ultimately these are present in both Figure 3 panels? If this is the case then why is the choice made to use "average mass spectra"? Why not present some precise exemplar mass spectra, or perhaps use another technique to highlight peaks (lack of peaks) or peak combinations of interest (e.g., PMF)? Without considering the issue deeply, I think averaging these types of spectra will result in a loss of information. Furthermore, given the strong fragmentation I would think any ambient sample that included biomass materials (e.g., biomass burning) would to the eye look similar.

Overall my impression is that the mass spectrometry approach to particle composition taken within this manuscript contains either too little or too much information. Perhaps it would be better left to an entirely different report to present and discuss links between mass spectra of ambient samples and mass spectra of known cellulose samples? If the authors presume to have a strong case linking their characterizations to ambient measurements and therefore might make some statement about the atmospheric budget of cellulose, it is perhaps an important story beyond the scope of this manuscript. As it stands the link of the mass spectra to ice nucleation is never really revisited in the discussion and conclusions, making its presentation seem to add material without a clear purpose.

This section (*2. Sample Preparation and Characterization*) is also somehow representative of the lack of manuscript cohesion. It was discomfiting to begin reading about the sample specifics in *2. Sample Specifications*, including introducing some SEM information, only to 12 pages later come

across a section **3.4 Surface Structure Analyses**, were the writing very much gave the impression it should have led section 2. Likewise this is revisited again a further 20 pages later in **4.4 Surface Structure of Cellulose Samples** where Figures 9 – 11 are introduced. These or one of these could potentially have led the entire manuscript as an introduction to the material. Generally, I am uncertain that any part of the material characterization needs to be in the results section. First, it does not match with the stated objectives of the paper that an important result is physical/chemical characterization of cellulose samples. Second, breaking up the discussion of the material of study is one example of how the current manuscript lacks cohesion. That said the analysis in 3.4 is outright confusing, for me primarily due to the introduction of "line structures". Is the statement, "Followed by the background correction, line structures on the particle surfaces were clipped." supposed to mean something? Is this type of image analysis something that is well-known in the SEM lexicon? I am unable to follow the "line structure" analysis, or discern from the cited material whether or not it is simply my ignorance of some standard analysis. Naively, from Figures 9 and 10 I would think that "line structure" simply says something about surface roughness at a length scale concomitant with the measurement wavelength. However, Figures 9 and 10 which seem to relate directly to this section 3.4 are not even introduced until section 4.4.

**Itemized Scientific and Editorial Comments:**

In my estimation the tilde symbol (˜) is repeatedly misused. A low tilde ($\sim$) typically denotes 'similar to' in mathematical terms (or approximately in an informal sense, within an order of magnitude), whereas $\approx$ should be used for 'approximately equal to'. The high tilde symbol ˜ is mathematically meaningless but herein it appears the authors have used it to denote both $\sim$ and $\approx$.

*Specific Suggestions by Page and Line Number (page, line):*

- (3,27) remove "indeed"

- (3,32) to study heterogeneous ice... (strike "the")

- (4,1) "various yet meticulous" seems like a misuse of *yet*

- (4,10) "remarked the importance" – fragment

- (4,28) What to the authors mean by "concurrent study"?

- (5,1) Should simply be 'in 2015', strike "year"

- (5,2)  *the* sensitivity, also suggest ending becomes, '... ice nucleation instruments with respect to immersion...'

- (5,8-9) strike "alphabetical order according to the abbreviations" Order is not relevant.

- (5, 15) many institutions should be preceded by a 'the' e.g., the Pacific Northwest National Laboratory...double check for readability

- (5, 15-20)  This is a log run-on sentence. Break apart and/or change.

- (5,21) awkward use of "towards"

- (5, 28) "using" should be *used*

- (7, 10)  The "electron micrograph-assessed size of...." What does this mean?

- (8,2)  insert *and* before "droplet residuals"

- (8,22)  extra )

- (8,34) Use of ˜ . Here is should likely be $\approx$; see above comment.

- (10,13) "or/and" is typically 'and/or'

- (10,27)  perhaps use 'in more detail than what is reported by' in place of "in addition to what"

- (11,10) *the* U.S.

- (11,29) suggest: With this methodology, a total of 5637 particles () were analyzed and impurity inclusions of less than 0.25% were identified.

- (11,25) are known *to have* negligible...

- (11,26) strike "for" and "as"

- (11,33) sodium, which possibly ...

- (12,1) strike "up and"

- (12,5) should this be $\leq 3\%$

- (12,6) "wall" should be *walls*, strike "our"

- (12,7) AIDA expansion experiments...

- (12,8) change to: impurities negligibly impact the ice nucleation activity of cellulose at heterogeneous...

- (12,12) Should the $>$ be a $<$? If it is correct then it seems an upper bound should be provided.

- (14,16) differential mobility analyzer should likely be plural, as I presume each partner was using their own unit.

- (14,20) see tilde comment

- (14,24) Units are missing for droplet size in parenthesis.

- (14,33) The discussion of what is activated to droplets versus "activated fraction" is poorly structured. Are the authors using AF to mean droplet activation or freezing? Furthermore, even in systems (e.g., CFDCs) when it is assumed all particles activate, it is likely not true that AF=1(see for example, Garimella et al.[1],[2]). This will be a source of uncertainty in measurements and should be acknowledged. Also perhaps a short statement of where and how such error would enter into the results should be made.

- (15,17) used *in* each

- (15,19) to what are the authors referring when they say "this subset"

- (15,34) Do not begin sentence with mathematical symbol, "$\Delta \log$...

At this point in the text I had largely run out of the time/energy to make careful editorial remarks. However, the need for careful editing remains throughout the text, I continue below primarily with comments I see as scientifically relevant.

- (16,1) see tilda comment

- (16,6) What is meant by "status of the suspension solution..."? Do they intend to say something like a, 'a description of the suspension...'

- (17,31) *the* Supplemental Information.

- (18,1-5) Have the authors considered the recent comment by Vali (2018) in response to the Polen et al.[3] AMT paper (see the discussion for Vali comment)? If so I think these works should be cited, and furthermore, it seems that $C_{INP}$ should emerge from differential freezing spectra, not simply what is presented in Eq. (4). This links directly to section 4.5.1 and Figure 12. Both of which perhaps should be moved forward to offer a cohesive view of how the active site spectra are generated.

- (19,12-26) The discussion of temperature binning, especially how the moving average is constructed is confusing and needs to be clarified. Typically a moving average reassigns a value for each temperature that is used. Thus some temperature must still be chosen? Depending on the temperature resolution it then seems that a 3-point moving average might be inadequate. More specifics are needed. I understand that perhaps for a 0.5 degree resolution, a 3 point, centered moving average could (generally) be used, with the average for each integer degree then extracted from the moving average and used for the binning. Such a description would be valid for that specific case,

but would not perhaps not make sense for a different T resolution. As it is presented, it is impossible to know what exactly was done for the temperature binning.

- (20, 25) "Complementally" is not a word

- (21, 4-10) Figure 4 is introduced and the next figure introduced is Figure 6? Figures should be numbered and introduced in order of appearance within the text. See the more general comment also regarding Figures 9-11.

- (22,11) ratio of the *log of* individual...$n_{s,geo}$ expressed as $n_{s,avg}$

- (23,4) What is meant by, "across the heterogeneous freezing T"? I suggest giving a range of T, or in someway being more specific.

- (23, 5-6) "each portion of techniques"? do the authors mean, 'each suspension technique'

- (23, 10) *the* two subsets...

- (23,13) indicates *a* fundamental

- (23, 15) The way in which Figure S2 is currently introduced and repeatedly referred to it would seem like it should be part of the main manuscript.

- (§4.3) Initially reading section 4.3 I thought that it would contain notable results from individual instruments. However, reading onward it seems details of measurements from every utilized instrument are included. In my mind if this approach is taken the individual instruments should be reported prior to the collective results present in sections 4.1 and 4.2, such that the collective results build from the individual results. Another choice could be made which would be to simply highlight particularly notable results from individual instruments and relegate the remainder to supplementary material. In the current form, given the primary stated purpose of the paper, the most important message is buried deep in the middle of the paper (Figure 4-5), and was easy to forget by the time I had finished reading to the end.

- (40, 2-4) The first 2 sentences of section 4.5.2 seem to be extraneous, and can be struck.

- (Section 4.5.2) Perhaps this would be better integrated into other parts of the text. It lacks motivation or connection to descriptions of the experiments and ends with an incomplete thought.

- (41, 2) strike"giant and submicron" These are disparate size scales which seem to suggest a full range of size.

- (41, 9-11) "...fibrous structures that may act as the ice nucleation active site..." seems completely speculative. This is not observed and no convincing link between surface structure and IN activity was established. It would be better to stick to concrete conclusions.

- (41, 22) "deviations in T..." What T? Specify.

- (Figure 1, caption) I think it should read 'therefore *not* useful...'

- (Figures 4-8) Mostly very nice figures, but I wonder if the plot areas could be optimized a bit to improve visibility? Shorter hash marks? Begin y-axis with $10^2$? Anything to improve the data visibility would be good.

- (Figure 6, caption) 19 panels/measurement methods? The caption states 20, what do I miss?

- (Figure 10) It is very unclear from the caption and text what is plotted here. Is it in fact a continuous data set, or do the lines represent connected data points? What was done to generate this plot? Can it be related to Figure 9?

- (Figure 12) See previous comment regarding frozen fraction and differential spectra etc.

**Summation:** The Hiranuma manuscript is an impressive effort to summarize and present an enormous amount of work by many research groups. However, in its current state the manuscript lacks focus and does not present a clear and cohesive picture that matches with its stated intent. I suggest that the manuscript should be heavily altered in such a way that a clear research trajectory is

presented. Furthermore, the inclusion of complementary information should be motivated by how it helps grow the intended understanding. The links between things like physical and chemical characterization of particles and the ice nucleation should be made explicit. If clear connections are lacking, perhaps it is better to leave certain things broadly descriptive with detailed supplementary material available. Finally, for ACP the entire scope of the work would benefit from stronger grounding in its atmospheric relevance – beyond the innumerable and complicated issues that abound from such a multifarious measurement comparison.

It is my hope that with the necessary work the manuscript can proceed to full publication, given that this is an impressive and important data set. However, the effort might require a significant distillation of the discussion manuscript.

–

[1] Garimella, S., Rothenberg, D. A., Wolf, M. J., David, R. O., Kanji, Z. A., Wang, C., Rösch, M., and Cziczo, D. J. (2017). Uncertainty in counting ice nucleating particles with continuous flow diffusion chambers. *Atmospheric Chemistry and Physics*, 17(17):10855–10864.

[2] Garimella, S., Rothenberg, D. A., Wolf, M. J., Wang, C., and Cziczo, D. J. (2018). How uncertainty in field measurements of ice nucleating particles influences modeled cloud forcing. *Journal of the Atmospheric Sciences*, 75(1):179–187.

[3] Polen, M., Brubaker, T., Somers, J., and Sullivan, R. C. (2018). Cleaning up our water: reducing interferences from nonhomogeneous freezing of "pure" water in droplet freezing assays of ice-nucleating particles. *Atmospheric Measurement Techniques*, 11(9):5315–5334.

---

## Referee Comment (RC2) · Anonymous Referee #2 · 7 Jan 2019

The authors present laboratory results on heterogeneous ice nucleation triggered by three different types of cellulose (MCC, FC, and NCC). They use 20 different methods to measure the ice nucleation activity (INA) including nine dry dispersion and eleven aqueous suspension techniques. The manuscript is well written and the topic fits well into the journal ACP. However, the authors should carry out some revisions before the paper is published in ACP. In general, I consider this paper very important, since cellulose is so common in the atmosphere and could be a very common ice nucleation particle (INP).

[Figure]

My main concern regarding this paper is that only three types of cellulose have been investigated. However, cellulose is the most common organic compound on Earth and it is the most common polysaccharide. Of course, there are many, many cellulose types and MCC, FC and NCC are only a very few representatives. It comes not clear from the manuscript how and why these three have been chosen. In general, I miss a more elaborated introduction (1.1 background) where the sources of cellulose in the biosphere and finally in the atmosphere are discussed. Also relevant literature should be discussed (regarding marine aerosols, bio-aerosols (fungi, pollen, bacteria, plant fragments, leaf litter etc.)), e.g. the fact that water extractable INPs consist of polysaccharides should be mentioned (Dreischmeier 2017,Pummer 2012). So there are many sources of cellulose but most cellulose is not ice nucleation active. Then it is important to understand what makes the difference in terms of INA. Why are some cellulose samples so much more ice nucleation active than others? The authors might at least try to find an answer on this question in order to enhance the scientific value of the manuscript.

In principle, the physical and chemical properties of cellulose depend a lot on the history of the respective sample: water uptake, swelling, drying, shrinking, are inherently important for the INA. Even a freeze-thawing cycle of the same cellulose-water system could change the INA from one experiment to the other. These are just some points which should be discussed in more detail and might also help to understand the results of the paper. From my point of view, cellulose is not the ideal candidate for an inter-comparison program due to its unstable INA. On the other hand, this study gives good proof that it is not so much the influence of the different instruments which are responsible for the differing results, but much more the cellulose sample, since it properties are not sufficiently constant.

Another important point is the specific surface area of cellulose, since the calculation of the ice active site number inherently depends on it. However, the specific surface area of dry cellulose is not the same as the surface area in aqueous solution after swelling.

Much more area becomes available and also the surface chemistry exhibited to the water interface might be changed. The authors should explain how they include this into their parametrization.

Minor comment Fig. 3, y-axis: "relative intensity (a.u.)"
* * *

---

## Author Comment (AC1) · 16 Feb 2019

**Authors Response to Anonymous Referee #1**

The authors wish to thank Referee #1 for his/her thoughtful comments and useful discussions. Below are our point-by-point responses (in blue texts) to the reviewer's comments. Corresponding modifications are reflected in the manuscript and figures.

**Referee Comment for Summation**
1. The Hiranuma manuscript is an impressive effort to summarize and present an enormous amount of work by many research groups. However, in its current state the manuscript lacks focus and does not present a clear and cohesive picture that matches with its stated intent. I suggest that the manuscript should be heavily altered in such a way that a clear research trajectory is presented.
2. Furthermore, the inclusion of complementary information should be motivated by how it helps grow the intended understanding. The links between things like physical and chemical characterization of particles and the ice nucleation should be made explicit. **If clear connections are lacking, perhaps it is better to leave certain things broadly descriptive with detailed supplementary material available**.
3. Finally, for ACP the entire scope of the work would benefit from stronger grounding in its atmospheric relevance – beyond the innumerable and complicated issues that abound from such a multifarious measurement comparison.
4. It is my hope that with the necessary work the manuscript can proceed to full publication, given that this is an impressive and important data set. However, the effort might require a significant distillation of the discussion manuscript.

**Authors Response**: The authors would like to start with addressing the referee's summation to provide our revision overview. Please see below our four comments for each point the referee raised:
1. **The manuscript has been heavily edited to clarify our study focus. As a result, our previous version of 83 pages manuscript is now shortened to 43 pages (excl. tables).** We revised the organization of the paper, which is now more intuitive, and match with our conclusion that several types of cellulose have the capacity to nucleate ice as efficiently as some mineral dust samples, warranting more studies. Additionally, we also changed the title (to *A comprehensive characterization of ice nucleation by three different types of cellulose particles immersed in water*) to clarify/simplify our study focus.
2. The referee makes a good point. We moved all complementary information (**Sects. 2.2, 2.3, 3.4, 4.4 and 4.5**) to the Supplemental Information (**S.2, S.3, S.5, and S.9**). The authors believe this has substantially improved readability of manuscript, and we are grateful for the referee's suggestion.
3. The authors added statements strengthening atmospheric relevance of cellulose with five new references in **Sect. 1.3** (detailed in our next comment response below). With this, the authors believe the revised focus of our manuscript meets with the aims and scope of ACP (i.e., laboratory study of aerosols and cloud formation with atmospheric implication).
4. The authors thank the referee again for his/her positive input. Please refer to Author Response point 1 above.

**Referee Comment**: From the beginning I am left a bit confused about what is the primary purpose of the manuscript.
1. Immediately in §1.3 Goals the authors say, "The main objective of this study is to examine how different ice nucleation instrument techniques compare when using chemically homogeneous biological material rather than multi mineral systems,..." This type of explicit statement and the general way the manuscript seems to be framed is as a summary of how well ice nucleation tests of a single substance in laboratories around the world can be compared.

2. In this context would AMT not have been a better choice for manuscript submission? For ACP, I would expect the motivations for the research to more clearly focus on the importance of cellulose to the physics and chemistry of the atmosphere.
3. In its current state the single paragraph on page 3 motivating cellulose as important in an atmospheric context is not particularly convincing.

**Authors Response**: The authors re-edited the entire manuscript to clarify the scope of the study. See below our three follow-up comments for each point the referee points out:

1. The authors revised the **Sect. 1.3** and the primary purpose of the revised manuscript as "Due to increasing and diverse awareness of presence of atmospheric cellulose (e.g., *Vlachou et al.*, 2018; *Schütze et al.*, 2017; *Legrand et al.*, 2007; *Yttri et al.*, 2018; *Samake et al.*, 2018) – not as levogulcosan (the pyrolysis product of cellulose), the main objective of this study is to comprehensively examine the immersion freezing efficiency of cellulose that could be important in an atmospheric context.". The authors also added the following sentence with two important citations in the **Sect. 1.3** to address the potential occurrence of cellulosic INPs in the atmosphere – "Besides, the comprehensive ice nucleation data of cellulose materials presented in this work can be used to elucidate the role of airborne biological ice-nucleating aerosols derived from leaf litters and their emissions over natural surfaces (e.g., *Schnell and Vali*, 1976) and harvest regions, which certainly contained populations of plant matter in the air (*Suski et al.*, 2018)."

2. With the revised form, the authors believe the focus of our manuscript meets with the aims and scope of ACP (i.e., laboratory study of aerosols and cloud formation with atmospheric implication). Please refer below to point 3 for the atmospheric relevance of cellulose.

3. The authors added five more references (listed below) to support an importance of cellulose in an atmospheric context in **Sect. 1**. More specifically, the authors revised the **Sect. 1.3** (as stated above) and the second paragraph of **Sect. 1.1** as "In general, airborne cellulose particles are prevalent ($>0.05$ µg m$^{-3}$) throughout the year even at remote and elevated locations as reported in *Sánchez-Ochoa et al.* (2007). More recent study of carbonaceous aerosol composition in Switzerland over two years showed that ambient cellulose represents approximately 36-60% of primary biological organic aerosols, and the ambient cellulose concentration exceeded a few µg m$^{-3}$ (Figs. 6 and 7d of *Vlachou et al.*, 2018).".

Reference:
- Vlachou, A. et al.: Advanced source apportionment of carbonaceous aerosols by coupling offline AMS and radiocarbon size-segregated measurements over a nearly 2-year period, Atmos. Chem. Phys., 18, 6187-6206, https://doi.org/10.5194/acp-18-6187-2018, 2018.
- Schütze, K. et al.: Sub-micrometer refractory carbonaceous particles in the polar stratosphere, Atmos. Chem. Phys., 17, 12475-12493, https://doi.org/10.5194/acp-17-12475-2017, 2017.
- Legrand, M. et al.: Major 20th century changes of carbonaceous aerosol components (EC, WinOC, DOC, HULIS, carboxylic acids, and cellulose) derived from Alpine ice cores, J. Geophys. Res.-Atmos., 112, D23S11, doi:10.1029/2006JD008080, 2007.
- Yttri, K. E. et al.: The EMEP Intensive Measurement Period campaign, 2008–2009: Characterizing the carbonaceous aerosol at nine rural sites in Europe, Atmos. Chem. Phys. Discuss., https://doi.org/10.5194/acp-2018-1151, in review, 2018.
- Samake, A. et al.: Polyols and glucose particulate species as tracers of primary biogenic organic aerosols at 28 french sites, Atmos. Chem. Phys. Discuss., https://doi.org/10.5194/acp-2018-773, in review, 2018.
- Schnell, R. and Vali, G.: Biogenic Ice Nuclei: Part I, Terrestrial and Marine Sources, J. Atmos. Sci., 33, 1554–1564, 1976.
- Suski, K. J., Hill, T. C. J., Levin, E. J. T., Miller, A., DeMott, P. J., and Kreidenweis, S. M.: Agricultural harvesting emissions of ice-nucleating particles, Atmos. Chem. Phys., 18, 13755–13771, https://doi.org/10.5194/acp-18-13755-2018, 2018.

**Referee Comment**: Within x**2. Sample Preparation and Characterization** the authors spend considerable space discussing the use of Laser Ablation mass spectrometry to characterize samples and also discuss ambient ALABAMA measurements. **To me the motivation for this type of characterization is not clear and I am left to surmise that ultimately a demonstration of the laboratory relevance to ambient measurements is the goal.** It is unclear what such a hard ionization treatment of cellulose, which in the case presented results in mass spectra of fragmented materials, can really illuminate. Cellulose has a high molecular weight m=z _ 160, which is the weight of its basic building block levoglucosan. Thus it is unsurprising with laser ablation one generates mass spectra with many fragments, why would one expect

anything different? To me the only thing that clearly emerges from the mass spectrometry is that the examined substances included have high molecular weight, and thus fragment to yield peaks at many lower molecular weights – therefore I am left wondering how Figures 1, 2 (d) and 3 further the discussion. Do the authors intend to assert that there are clear cellulose fingerprints and that ultimately these are present in both Figure 3 panels? If this is the case then why is the choice made to use "average mass spectra"? Why not present some precise exemplar mass spectra, or perhaps use another technique to highlight peaks (lack of peaks) or peak combinations of interest (e.g., PMF)? Without considering the issue deeply, I think averaging these types of spectra will result in a loss of information. Furthermore, given the strong fragmentation I would think any ambient sample that included biomass materials (e.g., biomass burning) would to the eye look similar. **Overall my impression is that the mass spectrometry approach to particle composition taken within this manuscript contains either too little or too much information. Perhaps it would be better left to an entirely different report to present and discuss links between mass spectra of ambient samples and mass spectra of known cellulose samples? If the authors presume to have a strong case linking their characterizations to ambient measurements and therefore might make some statement about the atmospheric budget of cellulose, it is perhaps an important story beyond the scope of this manuscript.** As it stands the link of the mass spectra to ice nucleation is never really revisited in the discussion and conclusions, making its presentation seem to add material without a clear purpose.

**Authors Response**: The authors defer to the referee. We omit our former **Sect. 2.4** *Atmospheric Relevance* because our attempt to demonstrate the laboratory relevance to ambient measurements contains a lot of speculation as the referee pointed out. Since MS data are valuable for other purpose, such as (1) discussion of the difference between dry and wet particle generation and (2) impurities tests, we would like to retain our lab-measured MS data in the **Supplemental Sects. S.2 and S.3**, but we will not address links between ambient cellulose and commercialized cellulose samples (and remove all of speculative parts pointed out). The authors agree that the discussion of *the links* would be better left to an entirely different report as the reviewer suggests.

**Referee Comment**: This section (**2. Sample Preparation and Characterization**) is also somehow representative of the lack of manuscript cohesion. It was discomfiting to begin reading about the sample specifics in **2. Sample Specifications**, including introducing some SEM information, only to 12 pages later come across a section **3.4 Surface Structure Analyses**, were the writing very much gave the impression it should have led section 2. Likewise this is revisited again a further 20 pages later in **4.4 Surface Structure of Cellulose Samples** where Figures 9 – 11 are introduced. These or one of these could potentially have led the entire manuscript as an introduction to the material. Generally, I am uncertain that any part of the material characterization needs to be in the results section. First, it does not match with the stated objectives of the paper that an important result is physical/chemical characterization of cellulose samples. Second, breaking up the discussion of the material of study is one example of how the current manuscript lacks cohesion. That said the analysis in 3.4 is outright confusing, for me primarily due to the introduction of "line structures". Is the statement, "Followed by the background correction, line structures on the particle surfaces were clipped." supposed to mean something? Is this type of image analysis something that is well-known in the SEM lexicon? I am unable to follow the "line structure" analysis, or discern from the cited material whether or not it is simply my ignorance of some standard analysis. Naively, from Figures 9 and 10 I would think that "line structure" simply says something about surface roughness at a length scale concomitant with the measurement wavelength. However, Figures 9 and 10 which seem to relate directly to this section 3.4 are not even introduced until section 4.4.

**Authors Response**: To improve the manuscript cohesion, we removed ≈9 pages of the discussion regarding sample preparation and characterization from the main manuscript. We moved the sample characterization into the supplement as follows:
- **Sect. 2.2** Chemical Composition → Moved to the **Supplemental Sect. S.2**

- **Sect. 2.3** Tests to Investigate Impurities → Moved to the **Supplemental Sect. S.3**
- **Sect. 3.4.** Surface Structure Analyses & **Sect. 4.4.** Surface Structure of Cellulose Samples
    → merged into the **Supplemental Sect. S.5**
- **Sect. 4.5.** Experimental Parameters → Moved to the **Supplemental Sect. S.9**

The authors believe that the revised form of Sect. 2 (Sample Characterization) and Sect. 3 (Methods) improves readability and aligns with the scope (comprehensively examining immersion freezing efficiency of cellulose that could be important in an atmospheric context) and the conclusion (several types of cellulose have the capacity to nucleate ice as efficiently as some mineral dust samples, warranting more studies).

The method for the SEM surface analysis including the background correction is based on the image analysis used in *Adachi et al.* (2007) but has been largely modified for this study with a rigorous input of the first author of the paper. Our surface structure analysis by SEM may be qualitative, but we would like to keep this section in the **Supplemental Sect. S.5**. The most common method for surface structure analysis would be a modern surface physisorption characterization tool. For instance, one can quantitatively determine the pore size distribution and the void volume density of cellulose samples (in addition to BET-SSA) via the standard isothermal physisorption method (at STP) using Krypton as low saturation pressure gasses. Krypton allows us to assess porous but low SSA materials like cellulose (personal communication with the manufacturer). Additionally, $CO_2$ will be used for nanoscopic pore (≈3.5 to 15 Å) distribution analysis, if necessary, due to its lower molecular quadrupole moment (*Rouquerol et al.*, 1989). Nitrogen can be used for BET surface area, micro-pore characterization, and meso-pore characterization (≈20 to 3000 Å). Finally, we will relate the measured quantities to the number of ice-nucleating surface active sites (i.e., $n_s(T)$) to develop the morphology-resolved parameterization, which can be formulated in the atmospheric models. Unfortunately, such a modern tool is not available to us, and we attempted to complement it by using SEM. It is important to note that this standard physisorption method is limited to measuring surface structures ≈3000 Å (≈ 0.3 μm), and may not be suitable to understand large structures on our cellulose materials. Nonetheless, this limitation does not rule out the possibility of a capillary condensation effect (i.e., inverse Kelvin effect) of nano-sized pores on nucleation enhancement (*Marcolli*, 2014 and 2017). Further detailed investigation of the influence of < 0.3 μm ice nucleation active sites using a physisorption tool in the future is necessary. The authors clarify this point in the **Supplemental Sect. S.5.** as follows: "Though looking into the pore size distribution and the void volume density of the samples below this size threshold is beyond the scope of the current study, it is necessary in the future to carry out a more detailed study in characterizing surface structure by applying a modern surface physisorption characterization tool. It is possible that a capillary condensation effect (i.e., inverse Kelvin effect) of nano-sized pores on nucleation enhancement (*Marcolli*, 2014 and 2017)."

Reference:
- Adachi, K., Chung, S. H., Friedrich, H., and Buseck, P. R.: Fractal parameters of individual soot particles determined using electron tomography: Implications for optical properties, J. Geophys. Res., 112, D14202, doi: 10.1029/2006JD008296, 2007.
- Marcolli, C., Deposition nucleation viewed as homogeneous or immersion freezing in pores and cavities. *Atmos. Chem. Phys.* **2014,** *14*, (4), 2071-2104.
- Marcolli, C., Pre-activation of aerosol particles by ice preserved in pores. *Atmos. Chem. Phys.* **2017,** *17*, (3), 1595-1622.

**Referee Comment**: (5, 15) many institutions should be preceded by a 'the' e.g., the Pacific Northwest National Laboratory... double check for readability.

**Authors Response**: Based on inputs from native English speakers, here is how we revised with the need definite articles in red. "…Beyond official INUIT-participating institutes, including Bielefeld University (BU), Goethe University Frankfurt (GUF), Johannes Gutenberg University of Mainz (JGU), Karlsruhe Institute of Technology (KIT), the Max Planck Institute for Chemistry (MPIC), the Leibniz Institute for Tropospheric Research (TROPOS), the Technical University of Darmstadt (TUD) and the Weizmann Institute of Science (WIS), ten associated institutes (five from U.S., three from E.U. and two from Japan) are involved in this study. These associated partners include Carnegie Melon University (CMU), Colorado State University (CSU), North Carolina State University (NC State), the Pacific Northwest National Laboratory (PNNL), West Texas A&M University (WTAMU), the Institute of Atmospheric Sciences and

Climate-National Research Council (ISAC-CNR), the University of Basel, the University of Leeds, the Meteorological Research Institute (MRI) and the National Institute of Polar Research (NIPR)…". These revisons have been incorporated in our **Sect. 1.3**.

**Referee Comment**: (14,33) The discussion of what is activated to droplets versus "activated fraction" is poorly structured. Are the authors using AF to mean droplet activation or freezing?

**Authors Response**: We meant the ratio of measured pristine ice crystal concentrations to the particle concentration as *AF*. On the other hand, *FF* represents the ratio of observed pristine ice crystal number relative to the total droplet number. This discussion is now moved to the **Supplemental Sect. S.4**.

"…most of the dry-dispersion methods measure the concentration of ice crystals and separately determine the particle concentration, assuming that for immersion freezing measurements the conditions chosen in the instrument cause all particles to be activated to droplets. This yields the ratio of measured pristine ice crystal concentrations to the particle concentration, the so-called "activated fraction"(*AF*) as described in *Burkert-Kohn et al*. (2017).".

**Referee Comment** *Cont'd*: Furthermore, even in systems (e.g., CFDCs) when it is assumed all particles activate, it is likely not true that AF=1(see for example, Garimella et al. [1,2]). This will be a source of uncertainty in measurements and should be acknowledged.

**Authors Response**: The reviewer makes an important point. The authors added the following sentence right after the discussion of *AF* and *FF* in the **Supplemental Sect. S.4.**:
"It is important to note that CFDCs may expose particles to different humidities and/or temperatures in chamber geometry; therefore, *AF* = 1 is not achieved because not all particles are activated into the droplets in CFDCs (*Garimella et al*., 2017; 2018).".

Reference:
- Garimella, S., Rothenberg, D. A., Wolf, M. J., David, R. O., Kanji, Z. A., Wang, C., Rösch, M., and Cziczo, D. J.: Uncertainty in counting ice nucleating particles with continuous flow diffusion chambers, Atmos. Chem. Phys., 17, 10855–10864, doi: https://doi.org/10.5194/acp-17-10855-2017, 2017.
- Garimella, S., Rothenberg, D. A., Wolf, M. J., Wang, C., and Cziczo, D. J.: How uncertainty in field measurements of ice nucleating particles influences modeled cloud forcing, Journal of the Atmospheric Sciences, 75, 179–187, 2018.

**Referee Comment** *Cont'd*: Also perhaps a short statement of where and how such error would enter into the results should be made.

**Authors Response**: This point is clarified and followed in **Supplemental Sect. S.4.**: "…simultaneous measurements at the same measurement location were done, and CFDCs yielded lower results by roughly a factor of 3 for conditions where all particles should activate to droplets in the instruments.".

**Referee Comment**: (18,1-5) Have the authors considered the recent comment by Vali (2018) in response to the Polen et al. 3 AMT paper (see the discussion for Vali comment)? If so I think these works should be cited, and furthermore, it seems that CINP should emerge from **differential freezing spectra**, not simply what is presented in Eq. (4). This links directly to section 4.5.1 and Figure 12. Both of which perhaps should be moved forward to offer a cohesive view of how the active site spectra are generated.

**Authors Response**: Yes, we are aware of this important recent discussion regarding alternate methods to analyze droplet freezing assays. Polen and Sullivan and co-authors of this study, and their recent paper in AMT regarding how to reduce and assess background freezing in droplet freezing assays led to Vali's comment where he pointed out the $k(T)$ differential method, as opposed to the commonly used cumulative $K(T)$ (which is then often converted to $n_s$) metric. Both these methods were originally laid out by *Vali* (1971). When Polen and Sullivan attempted to apply the $k(T)$ method to their dataset the resulting spectra

did not look reasonable. Further discussions with Gabor Vali led him to realize that applying $k(T)$ is not as trivial as it first appears. A key detail is the choice of the $\Delta T$ parameter, which is not simply the temperature step between successive droplet images. This led Vali to write a detailed explanation and tutorial on the $k(T)$ analysis, using the *Polen et al.* dataset as an example. This tutorial is currently under review in AMT (*Vali*, 2018). We have added a brief discussion of the $k(T)$ framework in the **Supplemental Sect. S.9.1**. with reference to these two recent papers and Vali's comment in AMT.

However, we elected to not use $k(T)$ for the cellulose datasets presented here. The great utility of $k(T)$ is to determine the number of INPs active in a narrow temperature range, $\Delta T$, as opposed to the cumulative INP or active site concentration returned by $K(T)$ or $n_s$. This is especially useful when the sample's freezing spectrum approaches that of the background freezing spectrum. Then $k(T)$ for the filtered water background can be subtracted from $k(T)$ of the sample to determine the number of INPs in that temperature segment attributable only to the sample. For the cellulose samples studied here, the observed freezing in the wet dispersion methods occurred at several degrees Celsius warmer than where background freezing is observed in the different systems. The reliable frozen fraction range for each instrument and any background correction applied is provided in **Table S3 and S4**. To properly determine the $k(T)$ spectrum for each sample and purified water background from each instrument would require going to the raw data, instead of the temperature binned data that each group provided for use in this manuscript (since the choice of $\Delta T$ must be optimized for the sample and method in the $k(T)$ analysis). This would require considerable effort while not providing significant value to the analysis. We do appreciate the suggestion and will certainly explore using $k(T)$ in future droplet freezing assay analyses.

The following discussion has been added in the **Supplemental Sect. S.9.1**. Please note that the study of $\Delta T$ to understand the $k(T)$ feature could be explored for a detailed quantitative assessment of artifacts including the background INP concentration. In this study, as we address the background correction method of individual techniques in **Tables S3 and 4**, we only provide a brief discussion of $k(T)$.

"In addition, the differential freezing spectra of the water used suspending cellulose samples can be used to assess the background freezing. The concept and importance of the differential freezing spectra is described in *Vali* (2018) and *Polen et al*. (2018), stemmed from the original concept introduced in Vali (1971). Briefly, the differential freezing, $k(T)$, can be formulated as:

$$k(T) = -\frac{1}{V_d \Delta T} ln \left(1 - \frac{\Delta N}{N_u(T)}\right) \tag{S1}$$

in which $k(T)$ is the differential ice nucleus concentration (L$^{-1}$), $V_d$ is the individual droplet volume, $\Delta T$ is an arbitrary temperature step, $\Delta N$ is the number of frozen droplets within aforementioned $\Delta T$, and $N_u(T)$ is the total number of unfrozen droplets at $T$. Note that $\Delta T$ is not the temperature step of the actual measurements, $\Delta T_m$. The study of $\Delta T$ could be explored in the future for a detailed quantitative assessment of artifacts including the background INP concentration. In this study, as we address the background correction method of individual techniques in **Tables S3 and S4**, we briefly report $k(T)$ with $\Delta T$ of 1.0 °C (i.e., $\Delta T$ to be ±0.5 °C around the reported temperature).".

Reference:

- Vali, G.: Revisiting the differential freezing nucleus spectra derived from drop freezing experiments; methods of calculation, applications and confidence limits, Atmos. Meas. Tech. Discuss., https://doi.org/10.5194/amt-2018-309, in review, 2018.
- Polen, M., Brubaker, T., Somers, J., and Sullivan, R. C.: Cleaning up our water: reducing interferences from nonhomogeneous freezing of "pure" water in droplet freezing assays of ice-nucleating particles, Atmos. Meas. Tech., 11, 5315-5334, https://doi.org/10.5194/amt-11-5315-2018, 2018.

**Referee Comment**: (19,12-26) The discussion of temperature binning, especially how the moving average is constructed is confusing and needs to be clarified. Typically a moving average reassigns a value for each temperature that is used. Thus some temperature must still be chosen? Depending on the temperature resolution it then seems that a 3-point moving average might be inadequate. More specifics are needed. I

understand that perhaps for a 0.5 degree resolution, a 3 point, centered moving average could (generally) be used, with the average for each integer degree then extracted from the moving average and used for the binning. Such a description would be valid for that specific case, but would not perhaps not make sense for a different T resolution. As it is presented, it is impossible to know what exactly was done for the temperature binning.

**Authors Response**: The authors also found this section confusing and misleading in part. We now clarified our moving average method in **Sect. 3.3** as "For the former case, the default span for the moving average is 3 (i.e., centered moving average for a 0.5 °C resolution data). If the temperature resolution is finer than 0.5 °C, the number of moving average span is equal to the number of data points in each temperature bin (an even span is reduced by 1).".

Referring back to Eq. (2), as an example, given an array of 100 droplets and a specified _T of 0.1 _C intervals, if the first 2 droplets freeze within one measurement interval, _T = 0.1 _C, _N = 2, and N(T) = 98. Using this metric, each freezing event in the interval _T is the result of at least one active INP, but given a small _T and a large N the interval can be approximately attributed to a single active INP.

**Referee Comment**: (23, 15) The way in which Figure S2 is currently introduced and repeatedly referred to it would seem like it should be part of the main manuscript.

**Authors Response**: **Figure S2** (now **Fig. S7**) is a graphical representation of **Table 8** (now **Table 4**). In our revised manuscript, **Fig. S7** is now mentioned only once in **Sect. 4.2**. The authors would like to stick with the tabulated data and keep the figure in the Supplemental Information (**Sect. S.6**).

**Referee Comment**: (x4.3) Initially reading section 4.3 I thought that it would contain notable results from individual instruments. However, reading onward it seems details of measurements from every utilized instrument are included. In my mind if this approach is taken the individual instruments should be reported prior to the collective results present in sections 4.1 and 4.2, such that the collective results build from the individual results. Another choice could be made which would be to simply highlight particularly notable results from individual instruments and relegate the **remainder to supplementary material**. In the current form, given the primary stated purpose of the paper, the most important message is buried deep in the middle of the paper (Figure 4-5), and was easy to forget by the time I had finished reading to the end.

**Authors Response**: Presenting the collective results (**Sects. 4.1 and 4.2**) followed by detailed results of individual techniques (**Sect. 4.3**) is our intension to constrain one of our main messages – three types of cellulose have the capacity to nucleate ice without an exception. With the current (edited) format, the authors believe this subsection fits in in our main manuscript. For this reason, the authors would like to keep all individual results as is. Although all individual results are important, seven notable results from a subset of techniques are first listed (**Sects. 4.3.1-4.3.7**) as stated in the beginning of the section. In addition, we have deleted "Since the primary focus of this study is on the methods inter-comparison,…" to be consistent with our revised study objective; i.e., comprehensively examining the immersion freezing efficiency of cellulose that could be important in an atmospheric context.

**Referee Comment**: (Section 4.5.2) Perhaps this would be better integrated into other parts of the text. It lacks motivation or connection to descriptions of the experiments and ends with an incomplete thought.

**Authors Response**: The authors agree with the reviewer. Sect. 4.5.2 is now moved to the **Supplemental Sect. S.9**.

**Referee Comment**: (Figures 4-8) Mostly very nice figures, but I wonder if the plot areas could be optimized a bit to improve visibility? Shorter hash marks? Begin y-axis with 102? Anything to improve the data visibility would be good.

**Authors Response**: Figures 4, 6, 7 and 8 (now, Figs. 1, 3, 4 and 5, respectively) now have the lower end y-axis limit value of 2 x 10². Figure 5 (now, Fig. 2) now have the lower end y-axis limit value of 0.65.

**Referee Comment**: (Figure 10) It is very unclear from the caption and text what is plotted here. Is it in fact a continuous data set, or do the lines represent connected data points? What was done to generate this plot? Can it be related to Figure 9?

**Authors Response**: It is the plot of discrete data points (non-continuous). We re-plotted the figure (**Fig. S5**) with measured abundance data points.

[Figure]

Yes - **Figure 10** (now **Fig. S5**) is related to **Fig. 9** (now **Fig. S4**) as **Fig. S4** represents a snapshot picture of one of 123 particles we examined to generate a plot of compiled surface abundance of line structures per unit area ($\mu m^2$) as a function of size. For clarity, the authors modified the text in **Sect S.5.** as "**Figure S5** shows the surface density of these submicron structures on MCC as well as FC (i.e., a compilation of 61 MCC and 62 FC particles).". The figure caption is also modified as "**Figure S5**. A compiled surface abundance of line structures scaled to the particle surface area as a function of line structure length for MCC and FC particles (61 MCC and 62 FC particles). An example of surface image analysis used for the plot is shown in **Fig. S4**. Peaks with smaller than 0.2 μm include noise and are excluded.".

*Other Specific Suggestions:*
**Authors Response**: The authors would once again like to thank the referee for his/her effort to provide valuable comments for such a lengthy manuscript. Our point-by-point responses to the reviewer's technical comments are addressed below.
**Referee Comment**: (3,27) remove "indeed"
**Authors Response**: Removed.
**Referee Comment**: (3,32) to study heterogeneous ice... (strike "the")
**Authors Response**: Struck.
**Referee Comment**: (4,1) "various yet meticulous" seems like a misuse of yet
**Authors Response**: Corrected to "…between meticulous groups…".

**Referee Comment**: (4,10) "remarked the importance" – fragment

**Authors Response**: Changed to "…*Burkert-Kohn et al.* (2017) conducted the inter-comparison…".

**Referee Comment**: (4,28) What to the authors mean by "concurrent study"?

**Authors Response**: Replaced with "this study"

**Referee Comment**: (5,1) Should simply be 'in 2015', strike "year"

**Authors Response**: Struck.

**Referee Comment**: (5,2) the sensitivity, also suggest ending becomes, '... ice nucleation instruments with respect to immersion...'

**Authors Response**: Corrected as suggested.

**Referee Comment**: (5,8-9) strike "alphabetical order according to the abbreviations" Order is not relevant.

**Authors Response**: Struck.

**Referee Comment**: (5, 15-20) This is a log run-on sentence. Break apart and/or change.

**Authors Response**: It is now changed and sub-divided into two sentences as "In this study, we have used three cellulose samples: micro-crystalline cellulose (MCC, Aldrich, 435236), fibrous cellulose (FC, Sigma, C6288) and nano-crystaline cellulose (NCC, Melodea, WS1) as atmospheric surrogates for non-proteinaceous biological particles. These samples were shared with all collaborators, and immersion freezing experiments were conducted individually at each institution to obtain immersion freezing data as a function of multi-experimental parameters (see **Sect. 3.1**).".

**Referee Comment**: (5,21) awkward use of "towards"

**Authors Response**: Corrected – towards accessing ➔ to access

**Referee Comment**: (5, 28) "using" should be used

**Authors Response**: Corrected. Thank you.

**Referee Comment**: (7, 10) The "electron micrograph-assessed size of...." What does this mean?

**Authors Response**: Corrected. This part now reads "…the size of bulk materials measured by electron microscopy...".

**Referee Comment**: (8,2) insert and before "droplet residuals"

**Authors Response**: We now added numbers (1) – (3) for clarity – "These measurements correspond to *SSA* of (1) mechanically aerosolized particles (<10 μm in diameter) in the Aerosol Interaction and Dynamics in the Atmosphere (AIDA) chamber, (2) droplet residuals obtained after evaporating water content of 5 μL droplet of 0.03 wt% aqueous suspension and (3) bulk samples, respectively.".

**Referee Comment**: (8,22) extra )

**Authors Response**: Corrected. Thank you. Now in the **Supplemental Sect. S.2**.

**Referee Comment**: (8,34) Use of ˜ . Here is should likely be _; see above comment.

**Authors Response**: Corrected. Now in the **Supplemental Sect. S.2**.

**Referee Comment**: (10,13) "or/and" is typically 'and/or'

**Authors Response**: Corrected. Now in the **Supplemental Sect. S.2**.

**Referee Comment**: (10,27) perhaps use 'in more detail than what is reported by' in place of "in addition to what"

**Authors Response**: Corrected as suggested. Now in the **Supplemental Sect. S.3**.

**Referee Comment**: (11,10) the U.S.

**Authors Response**: Corrected. Thank you. Now in the **Supplemental Sect. S.3**.

**Referee Comment**: (11,29) suggest: With this methodology, a total of 5637 particles () were analyzed and impurity inclusions of less than 0.25% were identified.

**Authors Response**: Corrected as suggested. Now in the **Supplemental Sect. S.3**.

**Referee Comment**: (11,25) are known to have negligible...

**Authors Response**: Corrected as suggested. Now in the **Supplemental Sect. S.3**.

**Referee Comment**: (11,26) strike "for" and "as"

**Authors Response**: Struck. Now in the **Supplemental Sect. S.3**.

**Referee Comment**: (11,33) sodium, which possibly ...

**Authors Response**: Corrected as suggested. Now in the **Supplemental Sect. S.3**.

**Referee Comment**: (12,1) strike "up and"

**Authors Response**: Struck. Now in the **Supplemental Sect. S.3**.

**Referee Comment**: (12,5) should this be ≤3%

**Authors Response**: Corrected. Now in the **Supplemental Sect. S.3**.

**Referee Comment**: (12,6) "wall" should be walls, strike "our"

**Authors Response**: Corrected. Now in the **Supplemental Sect. S.3**.

**Referee Comment**: (12,7) AIDA expansion experiments...

**Authors Response**: Corrected. Now in the **Supplemental Sect. S.3**.

**Referee Comment**: (12,8) change to: impurities negligibly impact the ice nucleation activity of cellulose at heterogeneous...

**Authors Response**: Changed as suggested. Now in the **Supplemental Sect. S.3**.

**Referee Comment**: (12,12) Should the > be a <? If it is correct then it seems an upper bound should be provided.

**Authors Response**: Corrected to <23 cm$^{-3}$. Now in the **Supplemental Sect. S.3**.

**Referee Comment**: (14,16) differential mobility analyzer should likely be plural, as I presume each partner was using their own unit.

**Authors Response**: Corrected. Now in the **Supplemental Sect. S.4**.

**Referee Comment**: (14,20) see tilde comment

**Authors Response**: Corrected. Now in the **Supplemental Sect. S.4**.

**Referee Comment**: (14,24) Units are missing for droplet size in parenthesis.

**Authors Response**: these numbers are ratios (unit-less).

**Referee Comment**: (15,17) used in each

**Authors Response**: Corrected. Now in the **Supplemental Sect. S.4**.

**Referee Comment**: (15,19) to what are the authors referring when they say "this subset"

**Authors Response**: Corrected – the aqueous suspension  subset. Now in the **Supplemental Sect. S.4**.

**Referee Comment**: (15,34) Do not begin sentence with mathematical symbol, "_log...

**Authors Response**: Corrected – The $\Delta\log(n_{s,geo})/\Delta T$ value… Now in the **Supplemental Sect. S.4**.

**Referee Comment**: (16,1) see tilda comment

**Authors Response**: Corrected. Now in the **Supplemental Sect. S.4**.

**Referee Comment**: (16,6) What is meant by "status of the suspension solution..."? Do they intend to say something like a, 'a description of the suspension...'

**Authors Response**: Corrected as suggested. Now in the **Supplemental Sect. S.4**.

**Referee Comment**: (17,31) the Supplemental Information.

**Authors Response**: Corrected to "…as described in the **Supplemental Sect. S.1**".

**Referee Comment**: (20, 25) "Complementally" is not a word

**Authors Response**: Corrected – Complementally, we → We also. Now in the **Supplemental Sect. S.5**.

**Referee Comment**: (21, 4-10) Figure 4 is introduced and the next figure introduced is Figure 6? Figures should be numbered and introduced in order of appearance within the text. See the more general comment also regarding Figures 9-11.

**Authors Response**: All corrected. Now, the figure number order is sequential.

**Referee Comment**: (22,11) ratio of the log of individual...ns;geo expressed as ns;avg

**Authors Response**: Corrected. The sentence now reads "Next, **Fig. 2** depicts the $n_{s,geo}(T)$ diversity in $\log(n_{s,ind.})/\log(n_{s,avg})$, which represents the ratio of the log of individual measurements ($n_{s,ind}$) to the log average of $n_{s,geo}(T)$ expressed as $n_{s,avg}$ at given temperatures.".

**Referee Comment**: (23,4) What is meant by, "across the heterogeneous freezing T"? I suggest giving a range of T, or in someway being more specific.

**Authors Response**: Clarified as "similar ice nucleation above examined temperatures (>-36 °C).".

**Referee Comment**: (23, 5-6) "each portion of techniques"? do the authors mean, 'each suspension technique'

**Authors Response**: The authors mean "each technique (i.e., DD and AS)".

**Referee Comment**: (23, 10) the two subsets...

**Authors Response**: Corrected. Thank you.

**Referee Comment**: (23,13) indicates a fundamental

**Authors Response**: Corrected. Thank you.

**Referee Comment**: (40, 2-4) The first 2 sentences of section 4.5.2 seem to be extraneous, and can be struck.

**Authors Response**: The second sentence is now struck. Correction can be seen in the **Supplemental Sect. S.9.2**.

**Referee Comment**: (41, 2) strike "giant and submicron" These are disparate size scales which seem to suggest a full range of size.

**Authors Response**: Struck.

**Referee Comment**: (41, 9-11) "...fibrous structures that may act as the ice nucleation active site..." seems completely speculative. This is not observed and no convincing link between surface structure and IN activity was established. It would be better to stick to concrete conclusions.

**Authors Response**: We agree. This sentence is now omitted.

**Referee Comment**: (41, 22) "deviations in T..." What T? Specify.

**Authors Response**: Specified; -36 °C < $T$ < -4 °C.

**Referee Comment**: (Figure 1, caption) I think it should read 'therefore not useful...'

**Authors Response**: The authors modified the caption as "Laboratory reference mass spectra of dry dispersed cellulose particles with ALABAMA. a) Fibrous cellulose (FC), b) Microcrystalline cellulose (MCC), left: anions, right: cations. These mass spectra represent between 60 and 75% of the particles (FC: 1585 out of 2071; MCC: 193 out of 329).". It is now **Fig. S2**.

**Referee Comment**: (Figure 6, caption) 19 panels/measurement methods? The caption states 20, what do I miss?

**Authors Response**: The results of two versions of FRIDGE (deposition and immersion mode techniques) are co-plotted in **panel e**. Thus, we present 20-1 panels. For clarity, we now added the following sentence in the caption of **Fig. 3** – "Both aqueous suspension and dry dispersion results of FRIDGE are presented in **panel e**.".

**Referee Comment**: (Figure 12) See previous comment regarding frozen fraction and differential spectra etc.

**Authors Response**: Please refer to our previous response above.

**Note**: Dr. Romy Ullrich has been added as an author for her extensive contribution to the database work.

---

## Author Comment (AC2) · 16 Feb 2019

**Authors Response to Anonymous Referee #2**

The authors wish to thank Referee #2 for his/her thoughtful comments and useful discussions. Below are our point-by-point responses (in blue texts) to the reviewer's comments. Corresponding modifications are reflected in the manuscript and figures.

**Referee Comment**: My main concern regarding this paper is that only three types of cellulose have been investigated. However, cellulose is the most common organic compound on Earth and it is the most common polysaccharide. Of course, there are many, many cellulose types and MCC, FC and NCC are only a very few representatives. It comes not clear from the manuscript how and why these three have been chosen.

**Authors Response**: The reason we choose three cellulose types was because these have diverse surface structures (**Table 1**), and their INP properties differ by orders of magnitude. Further, such a family of cellulose types allows us to probe the sensitivity of ice nucleation experimental techniques towards detecting non-proteinaceous biological materials. This point is now clarified in the end of **Sect. 1.3.**: "The motivation of using multiple types of cellulose was to (1) examine the immersion freezing abilities of both predominantly supermicron (MCC and FC) and submicron (NCC) cellulose particles to assess a wide size range of chemically uniform biological particles and (2) look into diverse surface structure (**Table 1**)".

**Referee Comment**: In general, I miss a more elaborated introduction (1.1 background) where the sources of cellulose in the biosphere and finally in the atmosphere are discussed.

**Authors Response**: We added the following introductory sentences regarding general cellulose source in **Sect. 1.1.**: "Cellulose is a linear polymer of 1–4 linked β-d-anhydroglucopyranose molecules, derived from plant fragments, leaf litter, wood fiber, non-wood fiber and/or even microbes (*Quiroz-Castañeda & Folch-Mallol*, 2013; *Thakur and Thakur*, 2014; *Chawla et al.*, 2009). The composition and structure of cellulose-containing bio fiber depends on the source and several different factors, summarized in *Khalil et al.* (2012) and *Dittenber and GangaRao* (2012).".

Reference:
- Quiroz-Castañeda, R. E. and Folch-Mallol, J. L.: Hydrolysis of Biomass Mediated by Cellulases for the Production of Sugars. In: Sustainable Degradation of Lignocellulosic Biomass - Techniques, Applications and Commercialization, edited by: Chandel. A., ISBN: 978-953-51-1119-1, InTech, doi: 10.5772/53719, 2013.
- Chawla, P. R., Bajaj, I. B., Survase, S. A., Singhal, R.S.: Microbial cellulose: Fermentative production and applications, Food Technology and Biotechnology, 47, 107–124, 2009.
- Dittenber, D. B., and GangaRao, H. V. S. Critical review of recent publications on use of natural composites in infrastructure, Composites Part A: Applied Science and Manufacturing, 43, 1419–1429, doi: https://doi.org/10.1016/j.compositesa.2011.11.019, 2012.
- Khalil, H. P. S. A., Bhat, A. H., and Yusra, A. F. I.: Green composites from sustainable cellulose nanofibrils: A review, Carbohydrate Polymers, 87, 963–979, doi: https://doi.org/10.1016/j.carbpol.2011.08.078, 2012.
- Thakur, V. K., and Thakur, M. K.: Processing and characterization of natural cellulose fibers/thermoset polymer composites, Carbohydr. Polym., 109, 102–117, doi: https://doi.org/10.1016/j.carbpol.2014.03.039, 2014.

**Referee Comment**: Also relevant literature should be discussed (regarding marine aerosols, bio-aerosols (fungi, pollen, bacteria, plant fragments, leaf litter etc.)), e.g. the fact that water extractable INPs consist of polysaccharides should be mentioned (Dreischmeier 2017, Pummer 2012).

**Authors Response**: These polysaccharides are not discussed since these pollen release INM polysaccharides are fundamentally different from cellulose. The authors note that cellulose is a specific allomorph of polysaccharides (i.e., polymer containing D-glucose residues linked by β-1,4-glycosidic bonds). *Dreischemeier et al.* (2017) addresses boreal pollen INM saccharide vs. "*other polysaccharides such as cellulose.*". For given specific reason, the authors would like to omit any extensive discussion of INM in general in the current manuscript. Nevertheless, the authors now address the importance of more comprehensive study of plant constituents, including INM polysaccharides, with suggested citations in our conclusion section: "…it is important to further conduct comprehensive study on ice nucleation activities

of other important plant structural materials, such as cellulose polymorphs, lignin materials, lipids, carbohydrates and other macromolecule saccharides (e.g., *Pummer et al.*, 2012; *Dreischmeier et al.*, 2017), as well as natural plant debris in simulated super-cooled clouds of the lower and middle troposphere.".

Reference:
- Dreischmeier, K., Budke, C., Wiehemeier, L., Kottke, T., and Koop, T.: Boreal pollen contain ice-nucleating as well as ice-binding 'anti-freeze' polysaccharides, Sci. Rep., 7, 41890.
- Pummer, B. G., Bauer, H., Bernardi, J., Bleicher, S., and Grothe, H.: Suspendable macromolecules are responsible for ice nucleation activity of birch and conifer pollen, Atmos. Chem. Phys., 12, 2541-2550, https://doi.org/10.5194/acp-12-2541-2012, 2012.

**Referee Comment**: In principle, the physical and chemical properties of cellulose depend a lot on the history of the respective sample: water uptake, swelling, drying, shrinking, are inherently important for the INA.

1. Even a freeze-thawing cycle of the same cellulose-water system could change the INA from one experiment to the other. These are just some points which should be discussed in more detail and might also help to understand the results of the paper.
2. From my point of view, cellulose is not the ideal candidate for an intercomparison program due to its unstable INA.
3. On the other hand, this study gives good proof that it is not so much the influence of the different instruments which are responsible for the differing results, but much more the cellulose sample, since it properties are not sufficiently constant.
4. Another important point is the specific surface area of cellulose, since the calculation of the ice active site number inherently depends on it. However, the specific surface area of dry cellulose is not the same as the surface area in aqueous solution after swelling. Much more area becomes available and also the surface chemistry exhibited to the water interface might be changed. The authors should explain how they include this into their parametrization.

**Authors Response**: The reviewer makes good points. See below our four comments:

1. For clarity, all of our analyses were done only when cellulose materials were dry or newly wet-generated by purpose to minimize the bias from these potential artifacts and not to refrain from comparing wet and dry. Nonetheless, it is possible that "water uptake, swelling, drying, shrinking" processes may affect (therefore, cellulose may not be stable). The impact of freeze-thawing cycle as well as pre-activation (*Wagner et al.*, 2016) should be carefully looked into. The authors clarify this point in the **Supplemental Sect. S.10.** as follows: "Though looking into the stability of the samples is beyond the scope of the current study, it is necessary in the future to carry out a more detailed study in characterizing the saturation level and temperature dependence of specific adsorption-desorption processes at atmospherically relevant heterogeneous freezing temperature range of cellulose at <-4 °C (*this study*) by applying a modern surface physisorption characterization tool. It is possible that the freeze-thawing processes affect stability of cellulose materials due to water uptake, swelling, drying and/or shrinking. It is also desired to carefully look into pre-activation (e.g., *Wagner et al.*, 2016).".

   Reference:
   - Wagner, R., Kiselev, A., Möhler, O., Saathoff, H., and Steinke, I.: Pre-activation of ice-nucleating particles by the pore condensation and freezing mechanism, Atmos. Chem. Phys., 16, 2025-2042, https://doi.org/10.5194/acp-16-2025-2016, 2016.

2. Our thought is now clearly addressed in the conclusion section as well as in the **Supplemental Sect. S.10.** as follows:
   - "…These diversities suggest the complex surface structure and compositional heterogeneity may play a substantial role to explain the diversity. This also implies that the cellulose system might not be suitable as a calibrant at this stage unless we completely understand the complex properties of cellulose materials."

- "...The observed discrepancy may be due to non-uniform active site density for different sizes and/or the alteration in physico-chemical properties of cellulose by liquid-suspending it. Unless otherwise defined, the cellulose system may not be an ideal calibrant at this moment."

3. The authors agree. As addressed above, stability of the sample is different issue.
4. The reviewer is right – swelling may alter the specific surface area of cellulose. To rigorously address this point, it is necessary to carry out a more detailed study in characterizing and quantifying surface properties of cellulose materials by applying a modern surface physisorption characterization tool in the future (currently not available). Here we outline the necessary two steps of potential future studies:

- Step 1: Quantitatively determine the BET-SSA, the pore size distribution and the void volume density of cellulose samples via the standard isothermal physisorption method (at STP) using Krypton as low saturation pressure gasses. Krypton allows us to assess porous but low SSA materials like cellulose (personal communication with the manufacturer). Additionally, $CO_2$ will be used for nanoscopic pore ($\approx$3.5 to 15 Å) distribution analysis, if necessary, due to its lower molecular quadrupole moment (*Rouquerol et al*., 1989). Nitrogen can be used for BET surface area, micro-pore characterization, and meso-pore characterization ($\approx$20 to 3000 Å). Finally, we will relate the measured quantities to the number of ice-nucleating surface active sites (i.e., $n_s(T)$) to develop the morphology-resolved parameterization, which can be formulated in the atmospheric models.

- Step 2: Assess the saturation level and temperature dependence of specific adsorption-desorption processes at atmospherically relevant heterogeneous freezing temperature range of cellulose at <-4 °C. An advanced physisorption characterization tool (e.g., Micromeritics, 3Flex) enables the cryogenic-physisorption of $H_2O$ and $N_2$ for temperature above -28 °C and pressure below 100 mmHg (personal communication with the manufacturer). These ranges are relevant to atmospheric mixed-phase clouds, where immersion freezing dominates the ice nucleation process (*Hande and Hoose*, 2017). Assessing the effect of variabilities in thermodynamic conditions will allow us to define the relationship between material porosity and reactivity, separately and in conjunction with Step 1.

    Reference:
    - Rouquerol, J.; Rouquerol, F.; Grillet, Y., ENERGETICAL ASPECTS OF N2 AND AR ADSORPTION - SPECIFIC ADSORPTION, TWO-DIMENSIONAL PHASE-CHANGES AND ADSORPTION IN MICROPORES. *Pure and Applied Chemistry* **1989,** *61*, (11), 1933-1936.
    - Hande, L. B.; Hoose, C., Partitioning the primary ice formation modes in large eddy simulations of mixed-phase clouds. *Atmospheric Chemistry and Physics* **2017,** *17*, (22), 14105-14118.

**Referee Comment**: So there are many sources of cellulose but most cellulose is not ice nucleation active. Then it is important to understand what makes the difference in terms of INA. Why are some cellulose samples so much more ice nucleation active than others? The authors might at least try to find an answer on this question in order to enhance the scientific value of the manuscript.

**Authors Response**: The statement of "most cellulose is not ice nucleation active" seems speculative. As described above, the authors offer logical steps and a potential approach to find an ultimate answer for the question raised (i.e., why are some cellulose samples so much more ice nucleation active than others). This point is addressed in the **Supplemental Sect. S.10.** ("...it is necessary in the future to carry out a more detailed study in characterizing the saturation level and temperature dependence of specific adsorption-desorption processes at atmospherically relevant heterogeneous freezing temperature range of cellulose at <-4 °C (*this study*) by applying a modern surface physisorption characterization tool."). Indeed, these points warrant some follow up studies.

The authors note that our knowledge of whether the laboratory results of a few cellulose materials can be representatively scaled up to the total plant fiber content in the atmosphere to assess the overall role of non-proteinaceous bio-INPs in clouds and the climate system is still limited. Luckily, there has been

another on-going AIDA study to investigate if other important plant constituents, such as cellulose polymorphs, lignin materials, lipids and carbohydrates, as well as natural plant debris can act as bio-INPs in simulated super-cooled clouds of the lower and middle troposphere. Preliminary scientific results have been presented in three conferences as of 2015 (*Hiranuma et al.*, 2015; *Steinke et al.*, 2017 and 2018; Note both corresponding authors participate in all of these plant fiber INP studies). Overall, our findings support the view that MCC may be a good proxy for inferring ice nucleating properties of natural plant debris. Our detailed outcomes will be presented in another paper (currently in preparation).

To clarify this important point, the authors added the following sentence in the end of the conclusion section: "Our knowledge of non-proteinaceous biological INPs is still limited. Thus, it is important to further conduct comprehensive studies on the ice nucleation activity of other important plant structural materials, such as cellulose polymorphs, lignin materials, lipids, carbohydrates and other macromolecule saccharides (e.g., *Pummer et al.*, 2012; *Dreischmeier et al.*, 2017; *Suski et al.*, 2018), as well as natural plant debris in simulated supercooled clouds of the lower and middle troposphere. Such additional studies are especially important for assessing the overall role of non-proteinaceous bio-INPs in clouds and the climate system.".

Reference:

- Hiranuma, N., Hoose, C., Järvinen, E., Kiselev, A., Möhler, O., Schnaiter, M., Ulrich, R., Cziczo, D.J., Zawadowicz, M., Felgitsch, L., Grothe, H., Kulkarni, G., Reicher, N., Rudich, Y., and Tobo, Y: Ice nucleation by plant structural materials and its potential contribution to glaciation in clouds, AGU Fall Meeting, San Francisco, CA, USA, Dec., 2015.
- Steinke, I., Funk, R., Hiranuma, N., Möhler, O., and Zhang, K.: Immersion freezing properties of complex biological aerosols derived from plants, INUIT Final Conference and 2nd Atmospheric Ice Nucleation Conference, Grasellenbach, Germany, Feb.-Mar., 2018.
- Steinke, I., Funk, R., Hiranuma, N., Möhler, O., Shen, X.: From macromolecules to plant related aerosols – investigating the ice nucleation properties of complex biological particles, 1st Atmospheric IN Conference, Leeds, UK., Jan., 2017.
- Pummer, B. G., Bauer, H., Bernardi, J., Bleicher, S., and Grothe, H.: Suspendable macromolecules are responsible for ice nucleation activity of birch and conifer pollen, Atmos. Chem. Phys., 12, 2541–2550, https://doi.org/10.5194/acp-12-2541-2012, 2012.
- Dreischmeier, K., Budke, C., Wiehemeier, L., Kottke, T., and Koop, T.: Boreal pollen contain ice-nucleating as well as ice-binding 'anti-freeze' polysaccharides, Sci. Rep., 7, 41890, 2017.
- Suski, K. J., Hill, T. C. J., Levin, E. J. T., Miller, A., DeMott, P. J., and Kreidenweis, S. M.: Agricultural harvesting emissions of ice-nucleating particles, Atmos. Chem. Phys., 18, 13755-13771, https://doi.org/10.5194/acp-18-13755-2018, 2018.

**Referee Comment**: Minor comment Fig. 3, y-axis: "relative intensity (a.u.)"

**Authors Response**: We omit Fig. 3 concerning the comment provided by Referee #1 (and upon the agreement with the relevant data providers).

**Note**: Dr. Romy Ullrich has been added as an author for her extensive contribution to the database work.

---

## Author Response (AR2)

**Authors Response to Co-Editor**

The authors wish to thank Referee #1 for her thoughtful comments and useful discussions. Below are our point-by-point responses (in blue texts) to the editor's comments. Corresponding modifications are reflected in the manuscript and SI.

**Editor Comment**: p. 7, l. 11 – 19: This sentence would be clearer if split into two. I think mostly the insert 'not as levoglucosan' (note 'levoglucosan' is misspelled!) is interrupting the flow.

**Authors Response**: Corrected. It now reads "To date, there has been an increasing and diverse awareness of presence of atmospheric cellulose (e.g., *Vlachou et al.*, 2018; *Schütze et al.*, 2017; *Legrand et al.*, 2007; *Yttri et al.*, 2018; *Samake et al.*, 2018) – not as levoglucosan (the pyrolysis product of cellulose). Thus, the main objective of this study is to comprehensively examine the immersion freezing efficiency of cellulose that could be important in an atmospheric context."

**Editor Comment**: p. 17, l. 19: constrains --> constraints

**Authors Response**: Corrected. Thank you.

**Editor Comment**: p. 22, l. 31: reports --> report

**Authors Response**: Corrected.

**Editor Comment**: p. 22, 34: colder temperature --> lower temperatures (only materials or gases etc can be 'cold' whereas temperature is a value that is high or low, respectively)

**Authors Response**: We agree. Corrected.

**Editor Comment**: p. 24, l.22: 'See Supplemental Information for details' can be omitted.

**Authors Response**: Omitted.

**Editor Comment**: p. 26, l. 31: should 'activates' by 'activated'?

**Authors Response**: Yes. Thank you.

**Editor Comment**: p. 28, l. 35: Is 'attach' the right verb here? Do you mean 'agree'?

**Authors Response**: Yes. Corrected.

**Editor Comment**: p. 32, l. 10: Which table are you referring to here? 4S2?

**Authors Response**: SI Table S2.

**Editor Comment**: p. 32, l. 22: form --> from

**Authors Response**: Corrected.

**Editor Comment**: p. 32, l. 24: week T-dependent --> weak T-dependence of

**Authors Response**: Corrected.

**Editor Comment**: All subsections in Section 4.3: It would help if you spell out at the beginning of each section (or in Table 2) the different instruments as you did for INKA in Section 4.3.10.

**Authors Response**: As the Editor suggested, we now spelled out all instrument abbreviations at the beginning of each sub-section.

**Editor Comment**: p. 35, l. 22: closest to the results…

**Authors Response**: Corrected.

**Editor Comment**: p. 36, l. 14: warm temperature --> high temperatures

**Authors Response**: Corrected.

**Editor Comment**: SI p14, l. 12: There seems to be a verb missing (e.g., 'occurs' or similar)

**Authors Response**: Thank you for catching this. It now reads "It is possible that a capillary condensation of nano-sized pores (i.e., inverse Kelvin effect) occurs, enhancing ice nucleation (*Marcolli*, 2014 and 2017)."

**Editor Comment**: SI, p. 21, l. 17: indicate --> indicates

**Authors Response**: Corrected.

[revised manuscript text omitted]
 the ratio of measured pristine ice crystal concentrations to the particle concentrationa value, the so--called "activated fraction"(*AF*) as described inin e.g., *Burkert-Kohn et al*. (2017). Others look at the entirety of all droplets and check how many of these are frozen, determining a "frozen fraction" (*FF*), the latter being done e.g., for LACIS (*Burkert-Kohn et al*., 2017), but generally also for all aqueous suspension methods. Likewise, the uncertainties in *RH*$_w$ and *S*$_w$ are also small (<5%). It is important to note that CFDCs may expose particles to different humidities and/or temperatures in chamber geometry; therefore, *AF* = 1 is not achieved because not all particles are activated into the droplets in CFDCs (*Garimella et al*., 2017; 2018). 
[revised manuscript text omitted]

In addition, the differential freezing spectra of the water used suspending cellulose samples can be used to assess the background freezing. The concept and importance of the differential freezing spectra is described in *Vali* (2018) and *Polen et al*. (2018), stemmed from the original concept introduced in Vali (1971). Briefly, the differential freezing, *k*(*T*), can be

35  formulated as:

$$k(T) = -\frac{1}{V_d \Delta T} ln\left(1 - \frac{\Delta N}{N_u(T)}\right) \qquad\qquad\qquad\qquad\qquad\text{(S1)}$$

in which $k(T)$ is the differential ice nucleus concentration (L$^{-1}$), $V_d$ is the individual droplet
volume, $\Delta T$ is an arbitrary temperature step, $\Delta N$ is the number of frozen droplets within
aforementioned $\Delta T$, and $N_u(T)$ is the total number of unfrozen droplets at $T$. Note that $\Delta T$ is
not the temperature step of the actual measurements, $\Delta T_m$. The study of $\Delta T$ could be explored
in the future for a detailed quantitative assessment of artifacts including the background INP
concentration. In this study, as we address the background correction method of individual
techniques in **Tables S3 and S4**, we elect not to report $k(T)$.

**S.9.2. Nominal Experimental Parameters**

[revised manuscript text omitted]

3) **Interfacial effect characterization:** Since the cellulose is a strong desiccant and absorbs a lot of water from the droplet, pre-exposure to humidified condition may create partially immersed solid-liquid interfacial condition. An effect is viable. For instance, supermicron-sized particles (MCC and FC) partially immersed but half exposed to air may create the interfacial condition preferable for ice formation. This quasi-contact (perhaps also condensation) freezing process may be analogous to the dry dispersion techniques (with different induction time). The future study to visually inspect this mechanism by means of microscopy (*Kiselev et al.*, 2017) and verify it as an atmospherically representative process is an imperative task. Though looking into the stability of the samples is beyond the scope of the current study, it is necessary in the future to carry out a more detailed study in characterizing the saturation level and temperature dependence of specific adsorption-desorption processes at atmospherically relevant heterogeneous freezing temperature range of cellulose at <-4 °C (*this study*) by applying a modern surface physisorption characterization tool. It is possible that the freeze-thawing processes affect stability of cellulose materials due to water uptake, swelling, drying and/or shrinking. It is also desired to carefully look into pre-activation (e.g., *Wagner et al.*, 2016).

[revised manuscript text omitted]